# Insight into selectivity of photocatalytic methane oxidation to formaldehyde on tungsten trioxide

Yingying Fan[1,6], Yuheng Jiang [2,3,6], Haiting Lin[1,6], Jianan Li[1], Yuanjiang Xie[1], Anyi Chen[1], Siyang Li[2], Dongxue Han [1,4] ✉, Li Niu [1,5] ✉ & Zhiyong Tang [2] ✉

Tungsten trioxide ($WO_3$) has been recognized as the most promising photocatalyst for highly selective oxidation of methane ($CH_4$) to formaldehyde (HCHO), but the origin of catalytic activity and the reaction manner remain controversial. Here, we take {001} and {110} facets dominated $WO_3$ as the model photocatalysts. Distinctly, {001} facet can readily achieve 100% selectivity of HCHO via the active site mechanism whereas {110} facet hardly guarantees a high selectivity of HCHO along with many intermediate products via the radical way. In situ diffuse reflectance infrared Fourier transform spectroscopy, electron paramagnetic resonance and theoretical calculations confirm that the competitive chemical adsorption between $CH_4$ and $H_2O$ and the different $CH_4$ activation routes on $WO_3$ surface are responsible for diverse $CH_4$ oxidation pathways. The microscopic mechanism elucidation provides the guidance for designing high performance photocatalysts for selective $CH_4$ oxidation.

Photocatalytic $CH_4$ oxidation using semiconductor and solar light enables green synthesis of one-carbon (C1) oxygenates like methanol ($CH_3OH$) and HCHO to supply the key feedstock for chemicals production[1,2]. Compared with $CH_3OH$, the photocatalytic $CH_4$ oxidation to HCHO is more scientifically challenging. This is because, in contrast to one step conversion of $CH_4$ to $CH_3OH$, the preparation of HCHO from $CH_4$ oxidation generally needs to undergo multiple intermediates conversion[1,3,4], and moreover HCHO is easily overoxidized to carbon dioxide ($CO_2$)[3].

To date, numerous semiconductor photocatalysts for $CH_4$ oxidation to HCHO have been examined, such as titanium dioxide[5,6], zinc oxide[1,7] and $WO_3$[8–10]. Amongst, $WO_3$ is the only catalyst reported to be capable of generating HCHO with unity selectivity[9,10]. Unfortunately, due to the complex reaction process, diverse mechanisms on $CH_4$ oxidation to HCHO in $WO_3$ system have been proposed (Fig. 1)[8,9,11–14], which in turn provide the confused guidance for the photocatalyst design. Generally, we classify the reaction mechanisms in Fig. 1 into two pathways. Mechanisms 1–5 represent the radical processes of HCHO formation through $CH_4 \rightarrow CH_3OOH \rightarrow CH_3OH \rightarrow HCHO$[12,15]. Owing to the excessive existence of intermediates ($CH_3OOH$, $CH_3OH$), the radical reaction processes are not conducive to the highly selective production of HCHO. Alternatively, mechanism 6 involves the active site, where $CH_4$ is oxidized by lattice-O of $WO_3$ to directly make HCHO[9]. Apparently, the selectivity of HCHO in mechanism 6 approaches 100%. However, the crucial factors that drive photocatalysts, not limited to $WO_3$, following the desirable reaction mechanism remain largely unexplored.

[1]Center for Advanced Analytical Science, Guangzhou Key Laboratory of Sensing Materials and Devices, Guangdong Engineering Technology Research Center for Photoelectric Sensing Materials and Devices, c/o School of Chemistry and Chemical Engineering, Guangzhou University, Guangzhou 510006, P. R. China. [2]Chinese Academy of Science (CAS) Key Laboratory of Nanosystem and Hierarchy Fabrication, CAS Center for Excellence in Nanoscience, National Center for Nanoscience and Technology, 100190 Beijing, PR China. [3]Center for Nanochemistry, Peking University, 100871 Beijing, PR China. [4]Guangdong Provincial Key Laboratory of Psychoactive Substances Monitoring and Safety, Anti-Drug Technology Center of Guangdong Province, Guangzhou 510230, PR China. [5]School of Chemical Engineering and Technology, Sun Yat-sen University, Zhuhai 519082, P. R. China. [6]These authors contributed equally: Yingying Fan, Yuheng Jiang, Haiting Lin. ✉e-mail: dxhan@gzhu.edu.cn; lniu@gzhu.edu.cn; zytang@nanoctr.cn

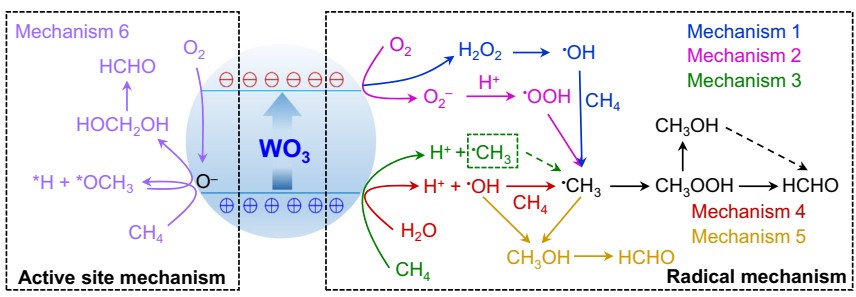

**Fig. 1 | Multiple reaction pathways of photocatalytic CH$_4$ oxidation to HCHO.** Reported oxidation of CH$_4$ to HCHO on WO$_3$ photocatalyst following six types of reaction mechanisms.

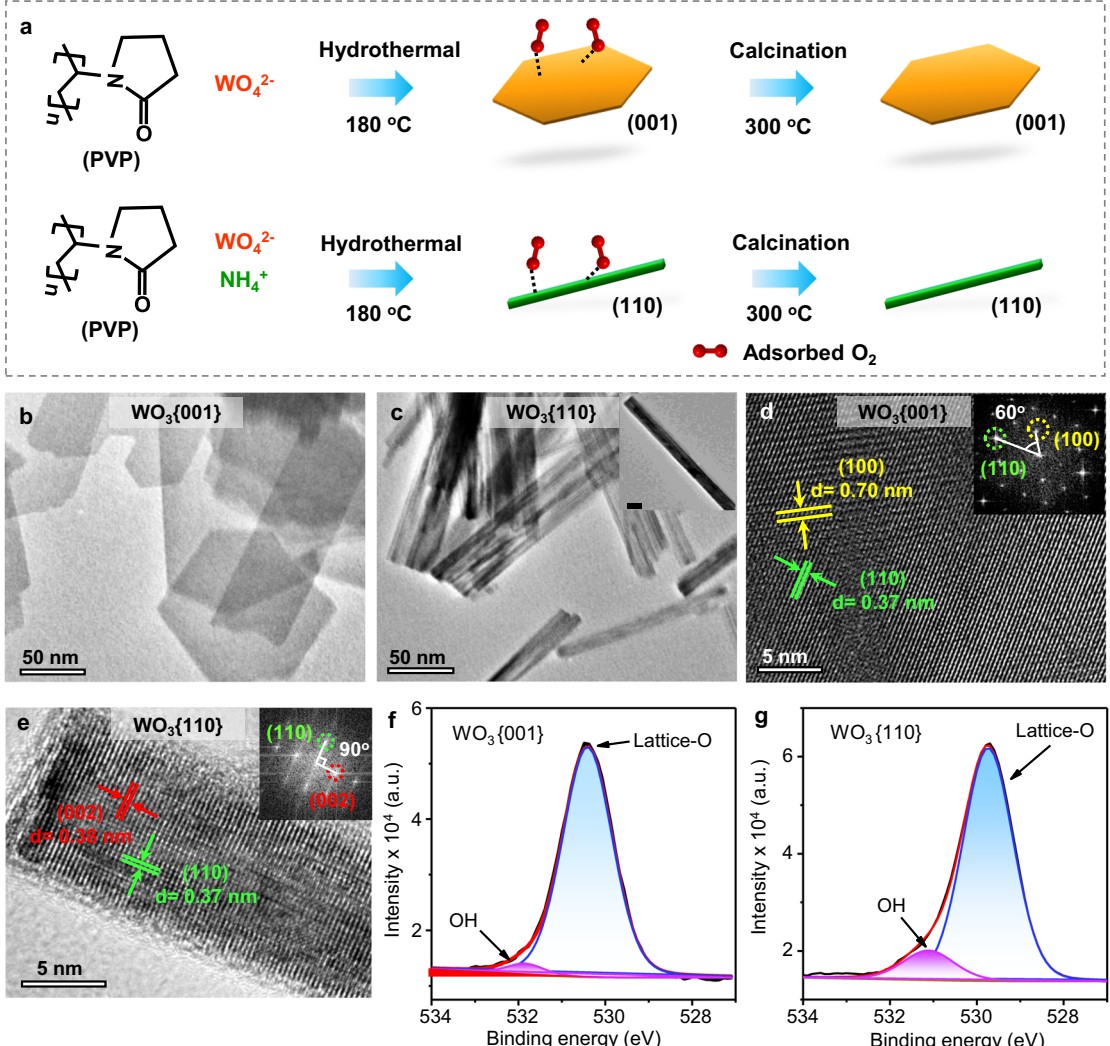

**Fig. 2 | Synthesis and characterization of the photocatalysts. a** Scheme of preparation process of WO$_3${001} and WO$_3${110}. **b** TEM images of WO$_3${001} and (**c**) WO$_3${110}. The insert is single WO$_3${110} nanowire with the scale bar as 10 nm.

**d** HRTEM images of WO$_3${001} and (**e**) WO$_3${110} with the inset FFT images. **f** High-resolution O1s XPS spectra of WO$_3${001} and (**g**) WO$_3${110} after 300 °C calcination.

In this work, we aim at understanding the origin of activity and selectivity of WO$_3$ photocatalysts upon CH$_4$ oxidation to HCHO. To simplify the investigation, all the reactions are performed under the identical reaction condition using WO$_3$ with the same crystal structure as photocatalysts. Such prerequisite guarantees that the distinct reaction performance only correlates with the surface coordination environment of WO$_3$. Therefore, we intentionally select the WO$_3$ samples with same crystal structure but enclosed by different facets as the candidates to inspect the surface effect.

## Results and discussion

### Synthesis and characterization of catalysts

The WO$_3$ photocatalysts enclosed by {001} and {110}, named as WO$_3${001} and WO$_3${110}, were synthesized by 180 °C hydrothermal treatment of Na$_2$WO$_4$·2H$_2$O followed by calcination at 300 °C (Fig. 2a). It is noted that the hydrothermal process with polyvinylpyrrolidone (PVP) as capping agent or ammonium ion (NH$_4^+$) as the directing agent facilitates the formation of {001} and {110} facets of WO$_3$, respectively. Furthermore, characterization results of temperature-programmed

desorption of $O_2$ ($O_2$-TPD), X-ray photoelectron spectroscopy (XPS) and Fourier-transform infrared spectroscopy (FTIR) confirm that the calcination at 300 °C results in full removal of the chemical adsorbed $O_2$ and the capping PVP or $NH_4^+$ from the catalyst surface (Supplementary Figs. 1–3). As shown in Fig. 2b, c, as-prepared $WO_3\{001\}$ and $WO_3\{110\}$ are of nanosheet and nanowire morphology, respectively. Their dominant surfaces are scrutinized by high-resolution transmission electron microscope (HRTEM) imaging with fast Fourier transformations (FFT). Figure 2d presents the clear lattice fringes of (110) (d = 0.37 nm) and (100) (d = 0.70 nm), suggesting that the electron beam transmits perpendicular to the exposed {001} crystal surface of $WO_3$ nanosheet (Supplementary Fig. 4). Likewise, perpendicularly to {110} surface (Fig. 2e), lattice fringes of (110) (d = 0.37 nm) and (002) (d = 0.38 nm) are discerned in parallel with the long and short sides of $WO_3$ nanowire. To further get the information on the surface composition, the high-resolution O1s XPS spectra of $WO_3\{001\}$ (Fig. 2f) and $WO_3\{110\}$ (Fig. 2g) are recorded, in which only lattice-O (ca. 530.4 eV)[16–18] and OH species (ca. 531.8 eV)[19,20] are distinguished without adsorbed $O_2$. The OH species results from the surface hydroxide[17,18,21], and the absence of chemisorbed $O_2$ is consistent with the $O_2$-TPD test (Supplementary Fig. 2b)[2,22]. Finally, the X-ray diffraction (XRD) patterns indicate that both $WO_3\{001\}$ and $WO_3\{110\}$ possess the hexagonal crystal structure (Supplementary Fig. 5).

## Photocatalytic CH$_4$ oxidation performance

Following the previous reports[1,3,4], we carried out the photocatalytic $CH_4$ oxidation with aqueous solution in a high-pressure reactor, where the total pressure of $CH_4$ and $O_2$ was maintained at 20 bar and the volume of $H_2O$ was fixed at 5 mL. Reaction temperature was controlled at 25 °C or 50 °C using water circulation to investigate the temperature effect. Before the photocatalytic experiments, the nitrogen adsorption−desorption isotherm curves were utilized to calculate the multipoint Brunauer–Emmett–Teller specific surface area of $WO_3\{001\}$ (Supplementary Fig. 6a–c) and $WO_3\{110\}$ (Supplementary Fig. 6d–f), being 15.70 m$^2$ g$^{-1}$ and 16.63 m$^2$ g$^{-1}$, respectively. We notice that the ratio of the specific surface area (0.94) of $WO_3\{001\}/WO_3\{110\}$ is similar to that of the geometric surface area (-1, Supplementary Figs. 7–9). Thus, we take the specific surface area of $WO_3\{001\}$ and $WO_3\{110\}$ for catalytic activity comparison. The $WO_3\{001\}$ and $WO_3\{110\}$ samples were first subjected to pure $CH_4$ atmosphere to reveal their intrinsic oxidation property. Only HCHO without other liquid products is produced on $WO_3\{001\}$ at both 25 °C (Fig. 3a, Supplementary Fig. 10) and 50 °C (Supplementary Fig. 11), indicating that $CH_4$ is directly oxidized to HCHO not through other intermediates. The absence of $CO_2$ is possibly ascribed to the low concentration of HCHO, which is not enough to be overoxidized (Supplementary Fig. 12). With the reaction time prolonging, the productivity of HCHO does not increase after 5 h (0.72 μmol m$^{-2}$). Since $H_2O$ and $WO_3\{001\}$ are sole oxygen sources in

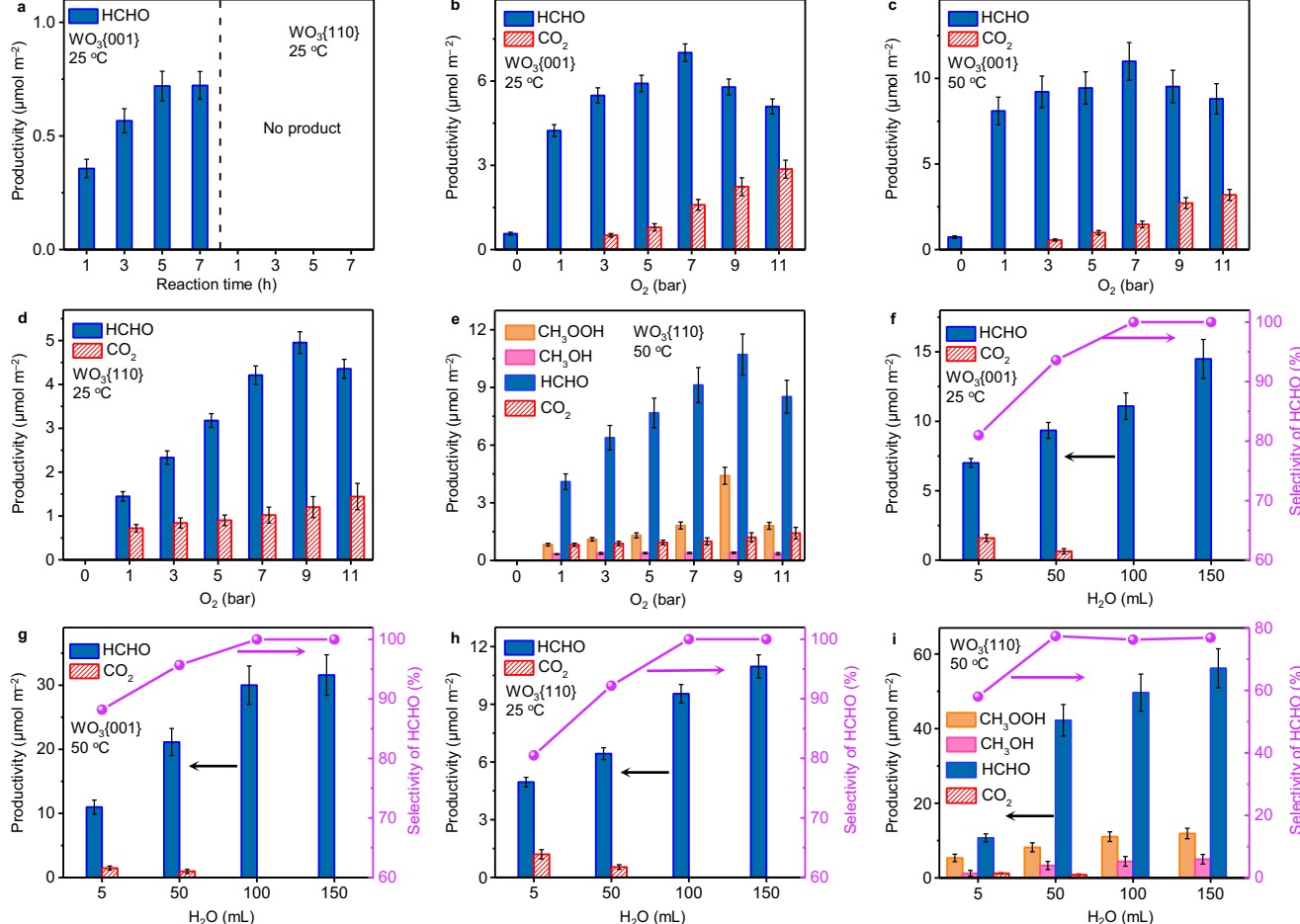

**Fig. 3 | Photocatalytic oxidation of CH$_4$ under different conditions.**
**a** Photocatalytic $CH_4$ oxidation performance on $WO_3\{001\}$ and $WO_3\{110\}$ in pure $CH_4$ atmospheres with reaction time prolonging at 25 °C. **b** Photocatalytic $CH_4$ oxidation performance on $WO_3\{001\}$ at 25 °C and (**c**) 50 °C with variation of $O_2$ amounts. **d** Photocatalytic $CH_4$ oxidation performance on $WO_3\{110\}$ at 25 °C and (**e**) 50 °C with variation of $O_2$ amount. **f** Photocatalytic $CH_4$ oxidation performance on $WO_3\{001\}$ at 25 °C and (**g**) 50 °C with variation of $H_2O$ amount. **h** Photocatalytic $CH_4$ oxidation performance on $WO_3\{110\}$ at 25 °C and (**i**) 50 °C with variation of $H_2O$ amount. Reaction condition: (**a–e**) 10 mg catalyst, 3 h reaction time, Xenon light 150 mW cm$^{-2}$, 5 mL $H_2O$, pressures of $CH_4 + O_2 = 20$ bar; (**f–i**) 10 mg catalyst, 3 h reaction time, Xenon light 150 mW cm$^{-2}$, 7 bar $O_2 + 13$ bar $CH_4$ ($WO_3\{001\}$), 9 bar $O_2 + 11$ bar $CH_4$ ($WO_3\{110\}$). Error bars indicate standard deviations.

pure $CH_4$ atmosphere, the O-atom of HCHO must originate from one of them. If the abundant $H_2O$ is the oxygen source, the HCHO formation from $CH_4$ oxidation will not stop at 5 h. Thus, we speculate that the finite surface lattice-O from $WO_3${001} provides the O-atom of HCHO and limits it production in $CH_4$ atmosphere as previously reported[9,10]. It is worth mentioning that during the $CH_4$ oxidation on $WO_3${001}, no $H_2$ product is detected (Supplementary Fig. 13). As for $WO_3${110} system, no product is found in $CH_4$ atmosphere at both 25 °C (Fig. 3a, Supplementary Figs. 14, 15) and 50 °C (Supplementary Fig. 11). Thus, we deduce that the lattice-O of $WO_3${110} cannot oxidize $CH_4$ to HCHO.

The effect of $O_2$ on $CH_4$ oxidation over $WO_3${001} and $WO_3${110} was revealed in the mixed $CH_4 + O_2$ atmosphere. Compared to the reaction of $WO_3${001} in pure $CH_4$ atmosphere, the participation of 7 bar $O_2$ brings ca. ninefold (7.05 μmol m$^{-2}$ at 25 °C, Fig. 3b and Supplementary Fig. 16) and 15-fold (11.04 μmol m$^{-2}$ at 50 °C, Fig. 3c and Supplementary Fig. 17) enhancement in the yield of HCHO. Benefitting from the rise of reaction temperature, the total yield is also increased by 1.45 times at 7 bar $O_2$ from 8.68 μmol m$^{-2}$ (25 °C) to 12.57 μmol m$^{-2}$ (50 °C). The increased HCHO production is likely caused by the sustainable renewal of lattice-O with the added $O_2$ for $CH_4$ oxidation[9,10], and the elevation of reaction temperature promotes this process. The descended HCHO production after 7 bar $O_2$ may be attributed to the reduction in partial pressure of $CH_4$ and the overoxidation to $CO_2$ (Supplementary Figs. 18–20). As comparison, with the addition of $O_2$, HCHO is also produced in $WO_3${110} system with the highest productivity of 4.96 μmol m$^{-2}$ (Fig. 3d, Supplementary Figs. 21, 22) at 25 °C. To check if any other intermediates are formed, the reaction temperature is elevated to 50 °C and many types of liquid products including $CH_3OOH$, $CH_3OH$ and HCHO are discerned (Fig. 3e, Supplementary Figs. 23–25). At 9 bar of $O_2$, the yield of HCHO reaches a maximum value of 10.71 μmol m$^{-2}$ (50 °C). The maximum total yield on $WO_3${110} is improved by 2.73 times by reaction temperature increasing from 25 °C to 50 °C, which is considerably higher than $WO_3${001}. Evidently, the HCHO formation on $WO_3${110} experiences the process of $CH_4 \rightarrow CH_3OOH \rightarrow CH_3OH \rightarrow HCHO$ or $CH_4 \rightarrow CH_3OOH \rightarrow HCHO$. The descended liquid products on $WO_3${110} upon $O_2$ pressure of larger than 9 bar are also assigned to the reduction in partial pressure of $CH_4$ and the overoxidation to $CO_2$. The distinct effect of reaction temperature on $CH_4$ oxidation performance between $WO_3${001} and $WO_3${110} is attributed to their different kinetic properties of Arrhenius and non-Arrhenius behaviors, respectively, which are discussed detailedly in the reaction kinetics analysis part in supplementary information. The higher activation energy of $CH_4$ oxidation on $WO_3${110} than $WO_3${001} leads to lower catalytic performance at 25 °C. However, the non-Arrhenius behavior on $WO_3${110} makes its reaction rate highly depend on the reaction temperature, thus the maximum productivity over $WO_3${110} surpasses $WO_3${001} at 50 °C. Besides, the promoted reaction rate of $WO_3${110} at 50 °C also accelerates the formation of intermediates, contributing to the appearance of $CH_3OOH$ and $CH_3OH$ signals.

The solvent volume has considerable influence on the photocatalytic selectivity and activity. With $H_2O$ volume increasing from 5 to 150 mL at the fixed pressure of 7 bar $O_2$ and 13 bar $CH_4$ in $WO_3${001} system, the productivity of HCHO increases to 14.49 μmol m$^{-2}$ at 25 °C (Fig. 3f, Supplementary Fig. 26) or 31.59 μmol m$^{-2}$ at 50 °C (Fig. 3g, Supplementary Fig. 27), possibly resulting from the improved dissolution of $CH_4$ in $H_2O$ solvent[7]. Besides, the $CO_2$ signal diminishes gradually and eventually disappears in 150 mL $H_2O$ (Supplementary Fig. 28, 25 °C and Supplementary Fig. 29, 50 °C), leading to the 100% selectivity of HCHO product. The disappearance of $CO_2$ signal is attributed to the reduced concentration of HCHO as previously reported (Supplementary Figs. 26, 27)[3,7]. While for $WO_3${110}, all the yields of $CH_3OOH$, $CH_3OH$ and HCHO grows with $H_2O$ volume increasing. Despite the selectivity of HCHO approaching 100% at 25 °C in 150 mL $H_2O$ (Fig. 3h, Supplementary Figs. 30, 31), it is only 73.73%

when the reaction temperature rises to 50 °C (Fig. 3i, Supplementary Figs. 32–35). This result correlates with the non-Arrhenius dependence over $WO_3${110} involved with radical mechanism. Besides, the HCHO selectivity enhancement with $H_2O$ volume increasing from 5 to 150 mL for both $WO_3${001} and $WO_3${110} in our work does not involve the change of reaction mechanism, which is described in supplementary information (Supplementary Figs. 36, 37). Both cyclic test and long-term reaction for $CH_4$ oxidation on $WO_3${001} and $WO_3${110} reveal their excellent photocatalytic stability (Supplementary Figs. 38–45). Additional verification experiments were also accomplished. The measured quantum efficiency values of both $WO_3${001} and $WO_3${110} follow the their diffuse reflectance spectra (Supplementary Figs. 46–48 and Supplementary Table 1), implying that the oxidation of $CH_4$ to HCHO on $WO_3${001} and $WO_3${110} involves photocatalytic reaction process. The contrast experiments in the absence of light, catalyst or $CH_4$ do no acquire the products (Supplementary Table 2).

## Mechanism investigation

In situ diffuse reflectance infrared Fourier transform spectroscopies (DRIFTS) performed in pure $CH_4$ atmosphere with or without $H_2O$ addition are used to investigate the active site mechanism on $WO_3${001} for $CH_4$ oxidation. Figure 4a, b shows that in pure $CH_4$ atmosphere without $H_2O$ addition, no peak is observed prior to light irradiation (0 min, black curves) on both $WO_3${001} and $WO_3${110}. Once the light turns on, a series of peaks emerge on $WO_3${001} (Fig. 4a). The peaks at 917, 1363 and 2830 cm$^{-1}$ are attributed to the stretching vibration of C-H bond in the adsorbed *$OCH_3$ species[2], while the ones at 1149 and 1478 cm$^{-1}$ are assigned to the vibration of adsorbed *$CH_2$ species[16,17]. Both *$OCH_3$ and *$CH_2$ are believed as the crucial intermediates upon the direct $CH_4$ oxidation to HCHO[9]. The peaks at 1251 and 1810 cm$^{-1}$ are ascribed to the adsorbed HCHO* and C = O* species, further verifying the HCHO formation[18,19]. Note that the adsorbed HCOO* at 1381 and 1594 cm$^{-1}$ might be the intermediate for overoxidation to $CO_2$[2]. Noteworthily, the consumption of lattice-O is evidenced by continuous descending of the W-O peak at 989 cm$^{-1}$ below zero baseline[20,21]. Clearly, $CH_4$ is steadily oxidized to HCHO by lattice-O of $WO_3${001}. As comparison, no rise of carboxyl peaks on $WO_3${110} is found along with irradiation time whereas a mass of lattice-O is lost (Fig. 4b). This result discloses that HCHO cannot be generated through $CH_4$ oxidation by lattice-O of $WO_3${110}, which is consistent with the catalytic experiments (Fig. 3a). Besides, no new peak is observed in both $WO_3${001}and $WO_3${110} systems after addition of $H_2O$ (Supplementary Fig. 49) with $CH_4$ atmosphere, indicating that $H_2O$ molecule does not involve in $CH_4$ oxidation. The causes of lattice-O consumption on $WO_3${001} and $WO_3${110} as well as their quantitative comparison are explained in detail in the theoretical calculation section and the ·OH radical analysis section see below. Moreover, the O1s XPS spectra of $WO_3${001} (Fig. 4c) and $WO_3${110} (Fig. 4d) after reaction in $CH_4$ atmosphere were recorded to quantitatively measure the change of lattice-O. The intensity of lattice-O peaks is reduced by 25% ($WO_3${001}) and 35% ($WO_3${110}) after reaction in $CH_4$ atmosphere (Supplementary Table 3), which is in accordance with the DRIFTS results. And the C = O peak (533.46 eV) is merely detected on $WO_3${001} but not on $WO_3${110}, presenting the HCHO production. Finally, the appearance of adsorbed $O_2$ peak is attributed to surface oxygen vacancy, which is formed by the removal of surface lattice-O.

To explore the role of $O_2$ in $CH_4$ oxidation on $WO_3${001} and $WO_3${110}, in situ DRIFTS experiments were also conducted in the mixed $CH_4 + O_2$ atmosphere without (Fig. 5a, b) or with $H_2O$ addition (Supplementary Fig. 50a and 50b). Similar signals on $WO_3${001} emerge as that in $CH_4$ atmosphere, indicating that there is no new surface reaction pathway (Fig. 5a and Supplementary Fig. 50a). Exceptionally, the W-O peak raises above the zero baseline (insets), signifying that the consumed lattice-O of $WO_3${001} is replenished adequately in $O_2$ atmosphere. The regeneration of lattice-O is also confirmed by the O1s

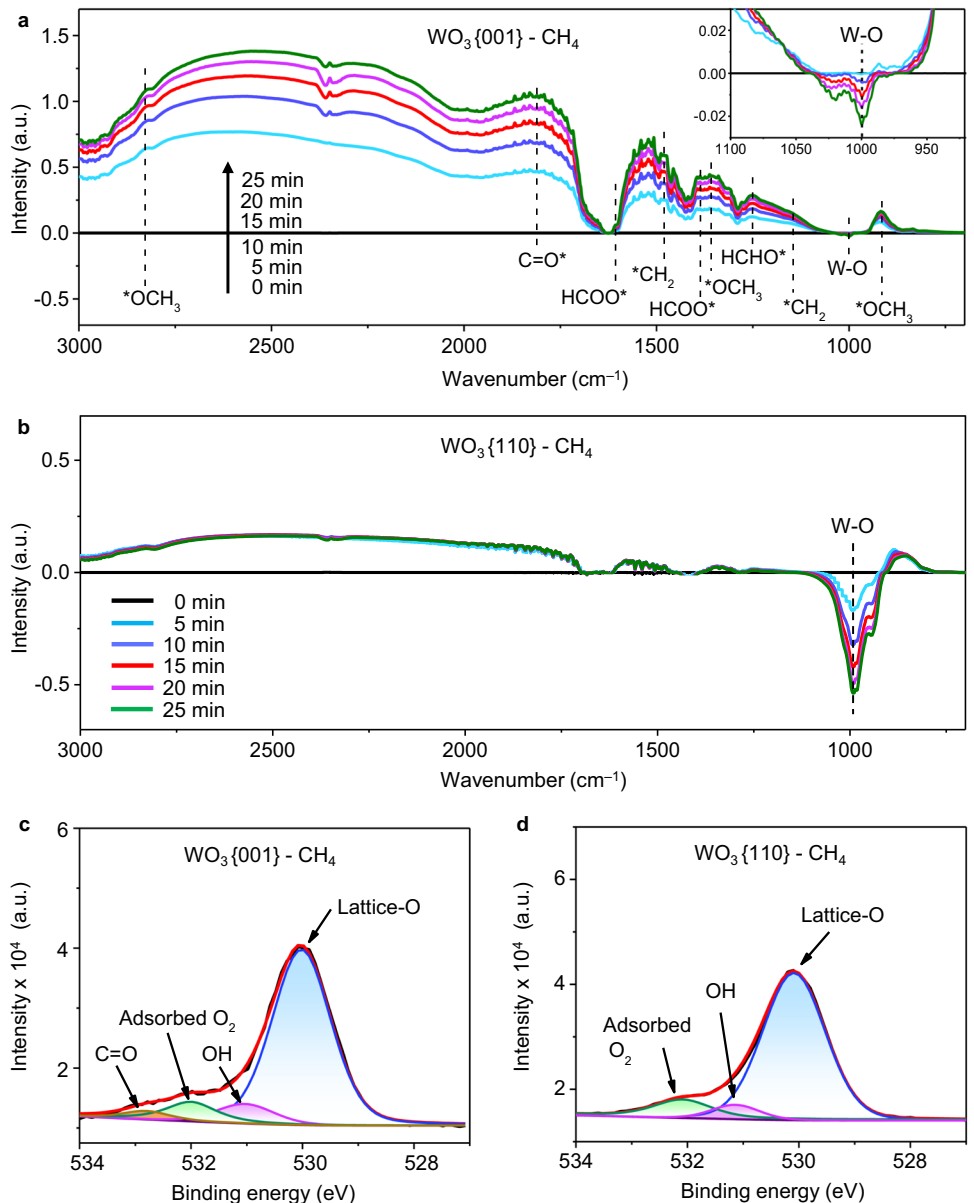

**Fig. 4 | Investigation on the role of lattice-O in CH$_4$ oxidation.** In situ DRIFTS spectra of (**a**) WO$_3${001} and (**b**) WO$_3${110} in CH$_4$ atmosphere under different light irradiation time without H$_2$O addition. Here, * denotes an adsorption site on surface. The inset is the magnified W-O peak. **c** High-resolution O1$s$ XPS spectra of WO$_3${001} and (**d**) WO$_3${110} after reaction in CH$_4$ atmosphere.

XPS of WO$_3${001} (Supplementary Fig. 51a). Such timely supplement of lattice-O guarantees the sustaining oxidation of CH$_4$ to HCHO. For WO$_3${110} sample, similar signals are also found between CH$_4$ + O$_2$ atmosphere (Fig. 5b) and pure CH$_4$ atmosphere (Fig. 4b) in absence of H$_2$O. The negative W-O signal indicates that the lost lattice-O in WO$_3${110} could not be totally repaired in O$_2$ atmosphere, which is also validated by 13.9% decrease in O1$s$ XPS peak intensity (Supplementary Fig. 51b and Supplementary Table 4). After H$_2$O addition in CH$_4$ + O$_2$ atmosphere, signals of *OCH$_3$, C = O*, *CH$_2$ and HCHO* appear on WO$_3${110} (Supplementary Fig. 50b). This result indicates that the photocatalytic CH$_4$ oxidation reaction over WO$_3${110} is implemented through a radical process, and the addition of H$_2$O enables the radical reaction pathway happening.

To elucidate the different CH$_4$ oxidation process on WO$_3${001} and WO$_3${110}, density functional theory (DFT) calculations were performed to examine their abilities responsible for CH$_4$ and H$_2$O adsorption as well as activation. Bridging O (O$_b$), terminal O (O$_t$) and W

atoms are taken as the adsorption sites of CH$_4$, respectively. It turns out that both WO$_3${001} (Fig. 6a, c) and WO$_3${110} (Fig. 6b, d) prefer CH$_4$ adsorption on O$_b$ sites (O$_b$-*CH$_4$) instead of O$_t$ (O$_t$-*CH$_4$) and W sites (W-*CH$_4$). To be specific for WO$_3${001}, through CH$_4$ activation, a *CH$_3$ group is formed and firmly adsorbed on its O$_b$ site to generating *OCH$_3$ group. Such species is confirmed by the rising *OCH$_3$ signals in in situ DRIFTS spectra of WO$_3${001} in CH$_4$ or CH$_4$ + O$_2$ atmospheres under light irradiation (Figs. 4a, 5a, Supplementary Figs. 49a, 50a). The positive energy of O$_b$ + CH$_3$(g) + *H ($\Delta E$ = 1.21 eV) means that the *CH$_3$ group is hardly desorbed from WO$_3${001} surface. This deduction is proved by the EPR test of WO$_3${001} in CH$_4$ or CH$_4$ + O$_2$ atmosphere, where no ·CH$_3$ radical is observed (Supplementary Fig. 52a). Alternatively, the ·OH radical is derived from the adsorption of H$_2$O molecules at O$_b$ site (O$_b$-*H$_2$O, Fig. 6a and Supplementary Fig. 53a) with subsequent oxidation, which is not involved in CH$_4$ oxidation as proved by in situ DRIFTS spectra before (Figs. 4a and 5a) and after H$_2$O addition (Supplementary Figs. 49a and 50a). Thus, the CH$_4$ oxidation

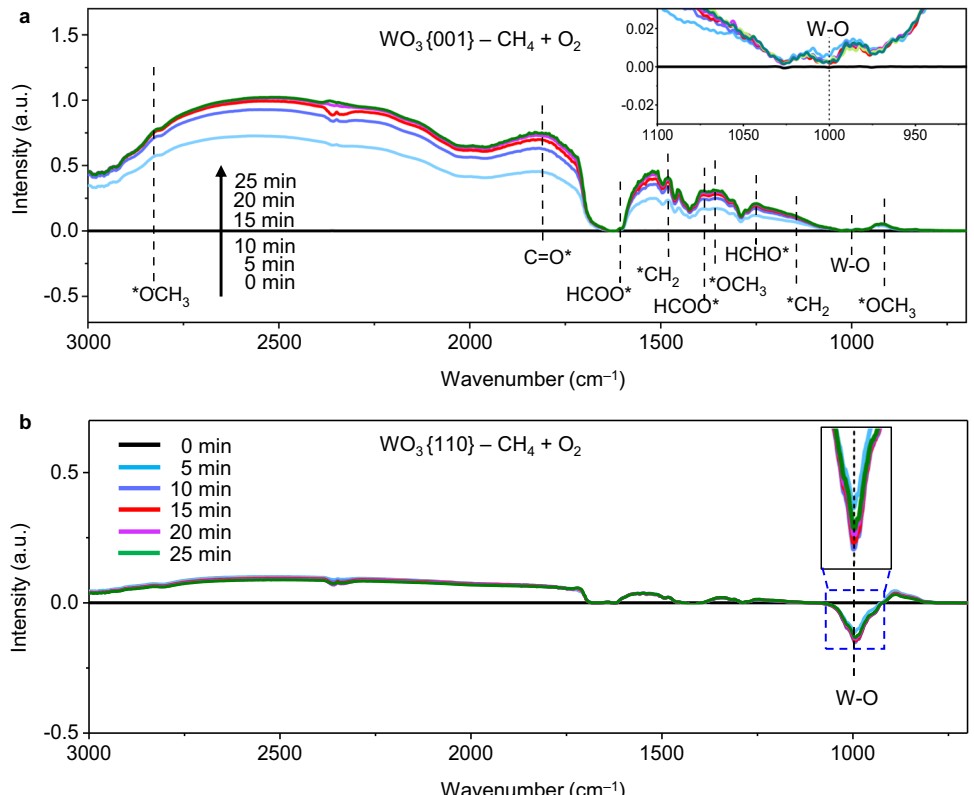

**Fig. 5 | Investigation on the role of $O_2$ in $CH_4$ oxidation.** In situ DRIFTS spectra of (**a**) $WO_3${001} and (**b**) $WO_3${110} in the mixed $CH_4 + O_2$ atmosphere under different light irradiation time without $H_2O$ addition. Here, * denotes an adsorption site on surface. The insets highlight the magnified W-O peak.

process of $WO_3${001} for HCHO generation goes through an active site mechanism rather than a radical mechanism. To explore whether the $H_2O$ oxidation on $WO_3${001} surface affects the $CH_4$ oxidation mechanism through $O_b$ site consumption, the ·OH formation mechanism is investigated. According to previous reports, the formation of ·OH radical via $H_2O$ oxidation can be divided into two ways: one is that both $O_b$ and photohole participate in $H_2O$ oxidation to generate one oxygen vacancy and two ·OH radicals ($O_b + h^+ + H_2O \rightarrow V_o + 2$·OH)[23-25]; the other is simply hole oxidizing $H_2O$ to produce one $H^+$ cation and one ·OH radical without $O_b$ consumption ($h^+ + H_2O \rightarrow H^+ + $·OH)[23,26]. As shown in Supplementary Fig. 54a and 55a, the positive energy of $V_o + 2OH(g)$ ($\Delta E = 0.36$ eV) on $WO_3${001} reveals that the ·OH radical formation is not through the process of $O_b + h^+ + H_2O \rightarrow V_o + 2$·OH. While the negative energy of *$H + OH(g)$ ($\Delta E = -0.91$ eV, Supplementary Figs. 54a and 55b) confirms that the ·OH radical is generated through $h^+ + H_2O \rightarrow H^+ + $·OH process without $O_b$ consumption. Therefore, the $H_2O$ oxidation on $WO_3${001} surface does not change the $CH_4$ oxidation process as no $O_b$ is consumed.

As for $WO_3${110}, Fig. 6b uncovers that after $CH_4$ activation on $O_b$ site of ($O_b$-*$H + CH_3(g)$), the adsorbed *$CH_3$ group is hardly formed and easily desorbed to form $CH_3(g)$ leaving $O_b$-H group. Thus, the active site mechanism of $CH_4$ oxidation by lattice-O does not occur on $WO_3${110}. Due to the higher surface energy of $WO_3${110} (4.13 J m$^{-2}$)[22,27], the $O_b$-H is also easily desorbed to form oxygen vacancy, which is similar to the $H_2O$ oxidation to form oxygen vacancy and ·OH radical discussed later. This is consistent with in situ DRIFTS results in $CH_4$ atmosphere without $H_2O$ addition, where no $CH_4$ oxidation is discerned only with the detected loss of W-O signal (Fig. 4b). Actually, in an aqueous environment, due to the large difference of energy between the adsorbed $H_2O$ and $CH_4$ molecules on the surface of $WO_3${110} (Fig. 6b and Supplementary Fig. 53b), $O_b$ preferentially adsorbs $H_2O$ molecules ($O_b$-*$H_2O$) to block the adsorption of $CH_4$ ($O_b$-*$CH_4$). And as displayed in Supplementary Figs. 54b and 55c, the

adsorbed $H_2O$ molecule is easily oxidized to ·OH radical through $O_b + h^+ + H_2O \rightarrow V_o + 2$·OH ($\Delta E = -1.24$ eV, $V_o + 2OH(g)$). During this process, $O_b$ is consumed along with oxygen vacancy generation, further excluding the possibility of $CH_4$ oxidation at $O_b$ active site. This is proved by the signal of W-O consumption in absence of $CH_4$ activation in in situ DRIFTS spectra of $WO_3${110} in $CH_4$ atmospheres with $H_2O$ addition (Supplementary Fig. 49b). Thereby, only ·OH radical without ·$CH_3$ radical is observed in pure $CH_4$ atmosphere for $WO_3${110} aqueous system (Supplementary Fig. 52b). Altogether, the $CH_4$ oxidation process on $WO_3${110} for HCHO generation follows a radical mechanism instead of an active site mechanism. We note that the large difference in the energy for $H_2O$ molecules adsorption on surfaces of $WO_3${001} and $WO_3${110} stems from the number of hydrogen bonds formed. On $WO_3${001} surface, $H_2O$ molecule is adsorbed through one hydrogen bond, while on $WO_3${110} surface, two hydrogen bonds are formed after $H_2O$ adsorption. Therefore, $WO_3${110} has a higher adsorption capacity for $H_2O$ molecules than $WO_3${001}.

The reactive radical species in $WO_3${110} system is monitored by EPR spectroscopy. In pure $O_2$ atmosphere, three signals at g = 2.027, 2.017 and 2.003 appear on $WO_3${110} surface upon photoirradiation (Fig. 7a). These three signals of orthorhombic symmetry are the characteristic hallmarks of surface-dwelling $O_2^-$ anions[2,5], which are stabilized at the W sites (Eqs. (1)–(4)). The surface-dwelling $O_2^-$ anion is capable of breaking $CH_4$ molecule to form ·$CH_3$ radical and ·OOH radical (Eq. (5)) with the regeneration of oxygen vacancy (Eq. (6))[5,28], which are proved by the 5,5-dimethyl-1-pyrroline N-oxide (DMPO)-·$CH_3$ (Supplementary Fig. 52b) and DMPO-·OOH (Fig. 7b and Supplementary Fig. 56) signals in EPR spectra[29], respectively. Finally, as-formed ·$CH_3$ radical combines with ·OOH radical to produce $CH_3OOH$ (Eq. (7)) that is unstable and decomposed to $CH_3OH$ and HCHO (Eqs. (8), (9))[7]. Although ·OH radical was also detected, it was mainly been quenched and not involved in oxygenates production, which has been explained in supplementary information (Supplementary Figs. 57–68).

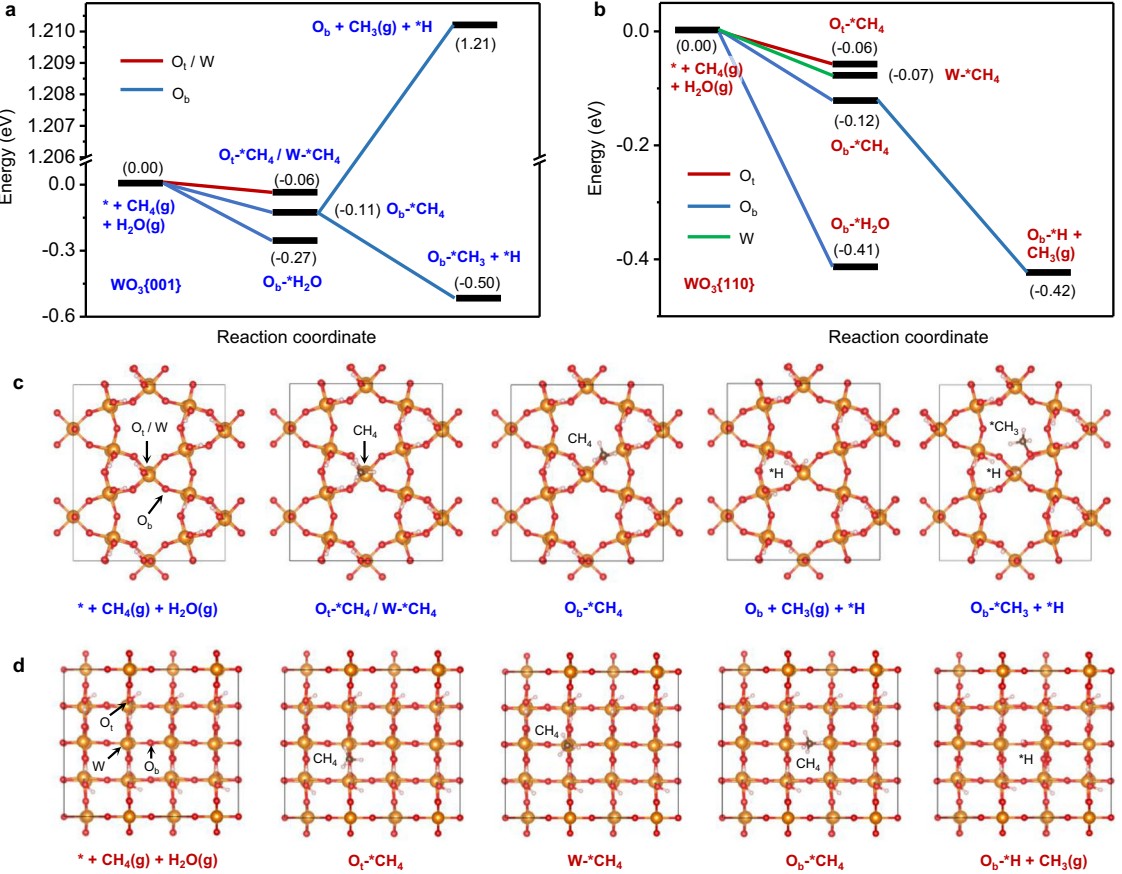

**Fig. 6 | Elucidation of different CH$_4$ oxidation pathways based on DFT calculation.** Energy diagrams of CH$_4$ and H$_2$O adsorption as well as CH$_4$ activation on the surface of (**a**) WO$_3${001} and (**b**) WO$_3${110} at the active sites of O$_b$ (blue line), O$_t$ (red line) and W (green line). Atomic configurations for the corresponding steps in the simulation of (**c**) WO$_3${001} and (**d**) WO$_3${110} (red – O, orange – W, gray – C, white – H).

As previous reports[5,30,31], CH$_3$OOH can be spontaneously decomposed into HCHO, whereas the conversion of CH$_3$OOH into CH$_3$OH is an electron reduction process. Thus, Eq. (9) becomes the major pathway of HCHO formation. This is the reason why HCHO is the main product in WO$_3${110} system. Altogether, the rich oxygen vacancies in WO$_3${110} facilitate the formation of O$_2^-$ anion, then promoting HCHO generation via the radical way. On the contrary, neither O$_2^-$ anion (Supplementary Fig. 69a) nor ·CH$_3$ (Supplementary Fig. 69b) and ·OOH (Supplementary Fig. 69c) radicals are observed in the WO$_3${001} system, excluding involvement of the radical process in HCHO formation. The redox potential energy for the intermediate formation over WO$_3${110} is provided in supplementary information (Supplementary Note 3).

$$W^{6+} - O^{2-} + h\nu \rightarrow W^{5+} - O^- \quad (1)$$

$$W^{5+} - O^- + H_2O \rightarrow W^{5+} \cdots OH^- + {}^{\bullet}OH \quad (2)$$

$$W^{5+} \cdots OH \rightarrow W^{5+} - V_o + {}^{\bullet}OH \quad (3)$$

$$W^{5+} - V_o + O_2 \rightarrow W^{6+} - O_2^- \quad (4)$$

$$W^{6+} - O_2^- + CH_4 + h^+ \rightarrow W^{6+} - V_o + {}^{\bullet}OOH + {}^{\bullet}CH_3 \quad (5)$$

$$W^{6+} - V_o + e^- \rightarrow W^{5+} - V_o \quad (6)$$

$${}^{\bullet}CH_3 + {}^{\bullet}OOH \rightarrow CH_3OOH \quad (7)$$

$$CH_3OOH + 2W^{5+} \cdots OH + 2e^- \rightarrow 2W^{5+} - O^- + CH_3OH + H_2O \quad (8)$$

$$CH_3OOH \rightarrow HCHO + H_2O \quad (9)$$

The energy band potential is also responsible for the distinct CH$_4$ oxidation mechanism between WO$_3${001} and WO$_3${110}. The energy band structure of both WO$_3${001} (Fig. 7c and Supplementary Fig. 70) and WO$_3${110} (Fig. 7c and Supplementary Fig. 71) is established with the valence band energy of 2.86 V and 2.46 V *vs* normal hydrogen electrode (NHE) and the conduction band energy of 0.08 V and −0.06 V *vs* NHE, respectively. The formation potential of O$_2^-$ anion from O$_2$ reduction is reported to be −0.046 V *vs* NHE[23], which is lower than −0.06 V of WO$_3${110} but higher than 0.08 V of WO$_3${001} (Fig. 7c). Therefore, WO$_3${110} rather than WO$_3${001} favors the formation of O$_2^-$ anion, leading to the generation of HCHO through the radical way. Alternatively, it is known that the top of valence band of WO$_3$ is mainly composed of O2$p$ orbitals[32,33], and the obvious photocurrent under irradiation (Supplementary Fig. 72) on both WO$_3${001} and WO$_3${110} manifests that their lattice-O is activated through loosing electron. The more positive valence band of WO$_3${001} than WO$_3${110} could contribute to the preferential oxidation of CH$_4$ to HCHO by lattice-O.

Armed with the above results, we attain the insight into the photocatalytic mechanisms of WO$_3${001} (Fig. 8a) and WO$_3${110} (Fig. 8b) toward CH$_4$ oxidation. In both cases of WO$_3${001} and WO$_3${110}, the

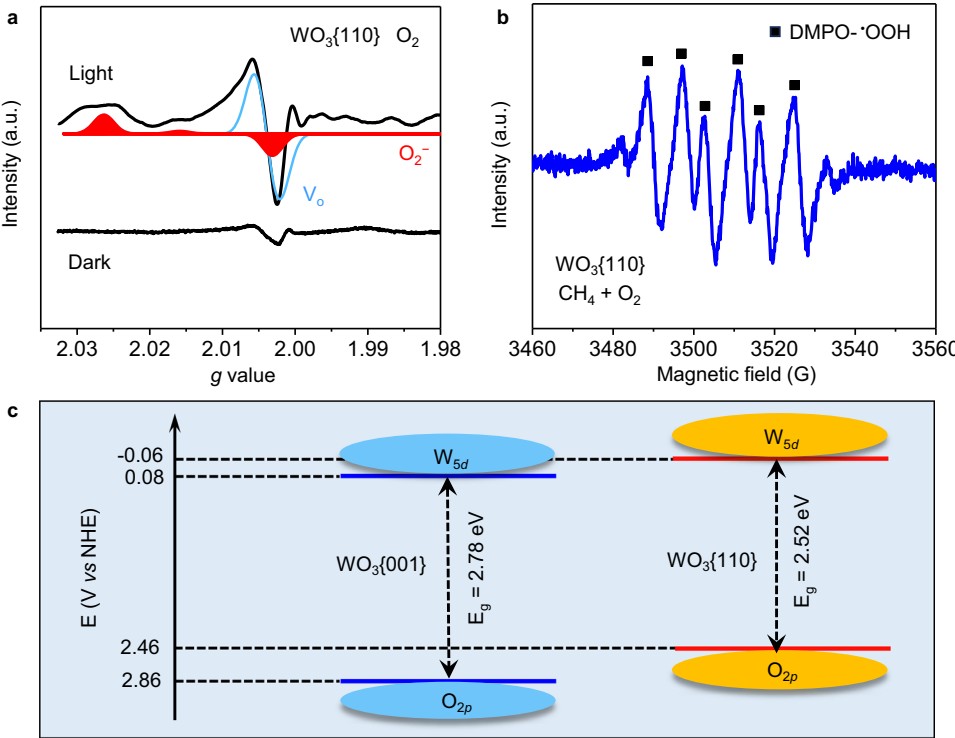

**Fig. 7 | Determination of reactive radical species. a** EPR spectra of WO$_3${110} in O$_2$ atmosphere at 77 K temperature. The WO$_3${110} is the recycled sample after CH$_4$ oxidation reaction without O$_2$. **b** EPR spectrum of WO$_3${110} under light irradiation for 80 s with CH$_4$ and O$_2$ dissolved in methanol. DMPO is added to the reaction mixture as the radical trapping agent. The WO$_3${110} is the recycled sample after CH$_4$ oxidation reaction without O$_2$. **c** Band energy diagrams of WO$_3${001} and WO$_3${110}.

**Fig. 8 | Proposed reaction mechanism. a** The schematic illustration of the proposed mechanism for photocatalytic oxidation of CH$_4$ on WO$_3${001} and (**b**) WO$_3${110}. V$_o$ is the oxygen vacancy.

CH$_4$ molecules are preferentially attached to the lattice-O$_b$, which are disclosed by the DFT calculations (Fig. 6a, b). It is known that the valence band maximum of WO$_3$ is mainly composed of O2$p$ orbitals, meanwhile the conduction band minimum is mainly constituted by W5$d$ orbitals[9,34]. Under light irradiation, the photoelectron from the valence band of WO$_3$ is excited to the conduction band, that is, from O2$p$ orbitals to W5$d$ orbitals. This excitation makes the valence state of O$_b$ change from O$^{2-}$ to O$^-$ and W atom from W$^{6+}$ to W$^{5+}$. Therein, the O$^-$ is the photohole (h$^+$) and the W$^{5+}$ is photoelectron (e$^-$)[34–36]. After the

CH$_4$ adsorption at the O$_b$ site, the O$^-$ (h$^+$) of WO$_3${001} is able to insert into the C-H bond of CH$_4$ molecule and sequentially forms the *OCH$_3$ and *CH$_2$ species, as shown in in situ DRIFTS spectra of WO$_3${001} in CH$_4$ and CH$_4$ + O$_2$ atmospheres (Figs. 4a, 5a). Meanwhile the left H atom from CH$_4$ is abstracted by adjacent -OH on W site via hydrogen atom transfer (HAT) process, as revealed by DFT results (O$_b$-*CH$_3$ + *H, Fig. 6a, c). Finally, the *OCH$_2$ species is desorbed to form HCHO molecule. During this process, the O$^-$ (h$^+$) is consumed and e$^-$ makes the W partially reduced (W$^{5+}$), which is inspected by the high-resolution

W4$f$ XPS spectra (Supplementary Fig. 73). The consumed $O_b$ atom becomes an oxygen vacancy that is conducive to the adsorption of $O_2$ molecule, as observed by the XPS spectra after photocatalytic reaction in pure $CH_4$ atmosphere (Fig. 4c). The adsorbed $O_2$ molecule is then reduced by photoelectrons ($e^-$) from $W^{5+}$ to fix the depleted $O_b$ atom, which is confirmed by the XPS spectra after $CH_4$ oxidation in $CH_4 + O_2$ atmosphere (Supplementary Fig. 51a, Supplementary Table 4) and in situ DRIFTS spectra of $WO_3${001} in $CH_4 + O_2$ atmosphere (Fig. 5a and Supplementary Fig. 50a). As a result, the $e^-$ is consumed. The above process involves the HCHO formation and the utilization route of photogenerated electron-hole pairs of $WO_3${001}.

As for $WO_3${110} (Fig. 8b), though the adsorbed $CH_4$ molecule may be activated by lattice-$O_b$, the adsorbed $^*CH_3$ group is hardly formed as revealed by the DFT calculation (Fig. 6b, $O_b$-$^*H + CH_3$(g)). Moreover, the large adsorption energy of $H_2O$ (Fig. 6b, $O_b$-$^*H_2O$) at the $O_b$ site further inhibits the above $CH_4$ activation process. Upon $H_2O$ oxidation at the $O_b$ site, the oxygen vacancy is formed by releasing $^\cdot OH$ radical (Fig. 4b, Supplementary Figs. 52b and 54b). With $O_2$ addition, the adsorbed $O_2$ molecule at the site of oxygen vacancy can be reduced to repair the left lattice-O atom or generate $O_2^-$ anion by the photoelectron from the conduction band of $WO_3${110}. Compared with the four-electron oxygen reduction to repair lattice-O, the one-electron oxygen reduction in $O_2^-$ generation pathway has lower kinetic energy barrier. Therefore, for $WO_3${110}, the consumed lattice-O of $WO_3${110} is only partially repaired, and $O_2$ is mainly involved in the formation of $O_2^-$ anion. The $O_2^-$ anion activates $CH_4$ molecule to produce $^\cdot CH_3$ (Supplementary Fig. 52b) and $^\cdot OOH$ (Fig. 7b) radicals. Through combination between $^\cdot CH_3$ and $^\cdot OOH$ radicals, $CH_3OOH$ is generated followed by decomposition to $CH_3OH$ and HCHO. Also, the difference in the reaction mechanism between this work and the previous works[3,37,38] is discussed in detail (Supplementary Note 2 and Supplementary Fig. 74).

To trace the carbon and oxygen sources of HCHO product, isotope tests were carried out. Employing $^{13}CH_4$ as reactant (Supplementary Fig. 75), NMR experiments show that only the peak of $HO^{13}CH_2OH$ at 81.9 ppm is detected on $WO_3${001} ($CH_4$ atmosphere or the mixed $CH_4$ and $O_2$ atmosphere at 25 °C) and $WO_3${110} (the mixed $CH_4$ and $O_2$ atmosphere at 25 °C). Note that $HOCH_2OH$ is the diol structure of HCHO in aqueous solution, verifying that the C-atom in HCHO comes from $CH_4$. The carbon source of HCHO in both $WO_3${001} and $WO_3${110} (Fig. 9a and Supplementary Fig. 76) systems was also inspected by gas chromatograph-mass spectrometer (GC-MS). The $H^{13}CHO$ peaks ($m/z = 31$) via $^{13}CH_4$ oxidation proves that the C-atom of HCHO is from $CH_4$. The origin of O-atoms in HCHO was traced by isotope labeling experiments with $^{18}O_2$ and $H_2^{18}O$. In both $WO_3${001} (Fig. 9b, Supplementary Fig. 77) and $WO_3${110} (Fig. 9c, Supplementary Fig. 78) systems, $HCH^{18}O$ peaks ($m/z = 32$) are found taking $^{18}O_2$ as reactant while $H_2^{18}O$ makes no difference (Fig. 9d, Supplementary Fig. 79 and Fig. 9c, Supplementary Fig. 78), indicating that the O-atom of HCHO originates from $O_2$. After the $CH_4$ oxidation reaction in $^{18}O_2$ atmosphere or taking $H_2^{18}O$ as solvent, the $WO_3${001} photocatalyst was recycled and put into another $CH_4$ oxidation system without $O_2$ addition and with $H_2O$ as solvent. Taking the recycled $WO_3${001} from $^{18}O_2$ atmosphere as photocatalyst, the clear signal of $HCH^{18}O$ confirms that the lattice-O atom of $WO_3${001} participates in the formation of

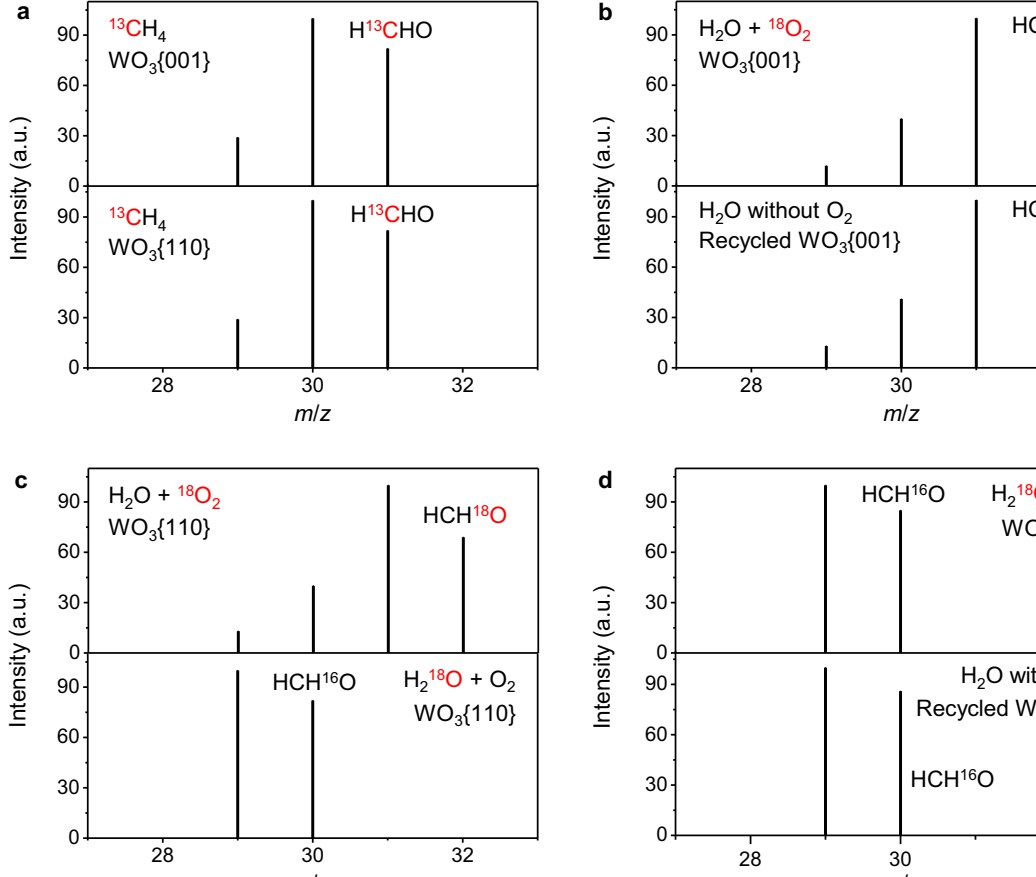

**Fig. 9 | Trace of HCHO elements. a** GC-MS spectra of HCHO obtained in $WO_3${001} and $WO_3${110} system using $^{13}CH_4$ as carbon isotope. **b** GC-MS spectra of HCHO obtained in $WO_3${001} system using $^{18}O_2$ as oxygen isotope and the recycled $WO_3${001} as photocatalyst without $O_2$ addition. **c** GC-MS spectra of HCHO obtained in $WO_3${110} system using $^{18}O_2$ or $H_2^{18}O$ as oxygen isotope. **d** GC-MS spectra of HCHO obtained in $WO_3${001} system using $H_2^{18}O$ as oxygen isotope and the recycled $WO_3${001} as photocatalyst without $O_2$ addition.

HCHO and can be supplemented by $O_2$. On the contrary, only HCHO without $^{18}O$ labelling is found taking the recycled $WO_3\{001\}$ from $H_2^{18}O$ solvent as photocatalyst, uncovering that the consumed lattice-O of $WO_3\{001\}$ cannot be repaired by $H_2O$. Additionally, we note that no product is found in pure $CH_4$ atmosphere using recycled $WO_3\{110\}$ as photocatalyst without $O_2$ addition, which is reasonable considering that the consumed lattice-O of $WO_3\{110\}$ is hardly repaired. Based on the oxygen isotope experiments, we conclude that the O-atoms of HCHO products on $WO_3\{001\}$ and $WO_3\{110\}$ are both from $O_2$ rather than $H_2O$. Besides, for $WO_3\{001\}$ system, $O_2$ is involved in the formation of HCHO by repairing the lattice-O. The different oxygen source analysis of $CH_3OH$ between this work and the previous work[38] is provided in supplementary information (Supplementary Note 2). The missing peak signal ($m/z = 28$) of HCHO is analyzed in supplementary information (Supplementary Figs. 80–85).

In summary, we in-depth explore the HCHO formation mechanism from photocatalytic $CH_4$ oxidation in $WO_3$ system. The high oxidation potential, satisfied adsorption of activated $CH_4$ molecule and low surface energy of $WO_3\{001\}$ confer the lattice-O to directly oxidize $CH_4$ to HCHO without intermediates and ensure the 100% selectivity of HCHO through the active site oxidation mechanism. While, the $WO_3\{110\}$ with preferential activation of $H_2O$ and rich oxygen vacancy conforms to the free radical oxidation mechanism, possibly giving rise to a low selectivity of HCHO. This work not only provides pivotal insight into competitive catalytic pathways involved with the active site mechanism and the radical mechanism, but also opens the avenue towards optimizing the performance of important photocatalytic reactions, including but not limited to $CH_4$ oxidation.

## Methods

### Photocatalyst preparation
Photocatalysts $WO_3\{001\}$ and $WO_3\{110\}$ were prepared through simple hydrothermal methods with subsequent calcination treatments.

For $WO_3\{001\}$, $Na_2WO_4 \cdot 2H_2O$ (2.7 g) and PVP (0.4 g) were dissolved in 50 mL water, then $CH_3COOH$ solution (8 mL) was added with continuous stirring for 30 min. After that, the suspension was transferred to a 100 mL Teflon-lined stainless-steel autoclave and treated under hydrothermal condition at 180 °C for 12 h. The obtained powers were washed with deionized water until the pH = 7. After drying at 80 °C overnight, the samples were calcined at 300 °C for 3 h with a heating rate of 3 °C min$^{-1}$. Finally, the desired $WO_3\{001\}$ was obtained.

For $WO_3\{110\}$, $Na_2WO_4 \cdot 2H_2O$ (2.7 g), PVP (0.4 g) and $CH_3COONH_4$ (0.4 g) were dissolved in 50 mL water, then $CH_3COOH$ solution (8 mL) was added with continuous stirring for 30 min. After that, the suspension was transferred to a 100 mL Teflon-lined stainless-steel autoclave and treated under hydrothermal condition at 180 °C for 12 h. The obtained powers were washed with deionized water until the pH = 7. After drying at 80 °C overnight, the samples were calcined at 300 °C for 3 h with a heating rate of 3 °C min$^{-1}$. Finally, the desired $WO_3\{110\}$ was obtained.

### Characterization
TEM and HRTEM were carried out using an FEI Tecnai G2 F20 electron microscope that was operated at 200 kV. The crystal structures were characterized through XRD patterns, which were obtained using a D/MAX-TTRIII (CBO) and Xeuss small-/wide-angle X-ray scattering (SAXS/WAXS) system with Cu Kα radiation (λ = 1.542 Å) operating at 50 kV and 300 mA. XPS experiments were carried out using an X-ray photoelectron spectrometer (EscaLab 250Xi, Thermo Scientific) and the spectra were calibrated with the C 1s peak at 284.8 eV. UV-visible diffuse reflectance spectra (UV-Vis DRS) taking $BaSO_4$ as the internal reference sample were recorded using a Hitachi U-3010 UV-visible spectrometer. A Mott-Schottky plot was obtained from samples in 1 M $Na_2SO_4$ solution at a frequency of 1500 Hz, prepared using a CHI 760E electrochemical workstation. The electron paramagnetic resonance (EPR) spectra were recorded at 9.43 GHz using a Bruker EMX spectrometer. In the case of the EPR test of $O_2^-$ anion: 25 mg photocatalysts were loaded into a quartz tube and gas of $O_2$ or mixed $O_2$ and $CH_4$ was introduced for 20 min. Then, the EPR tests were carried out with or without Xenon light irradiation at a liquid nitrogen temperature (77 K). For the EPR test of oxygen vacancy: 25 mg photocatalysts were loaded into a quartz tube and tested at room temperature without light irradiation. For the EPR test of $\cdot CH_3$ radical: 25 mg photocatalysts were loaded into a quartz tube with 5,5-dimethyl-1-pyrroline-1-oxide (DMPO) as the radical trapping agent in aqueous solution, then the test was carried out in the mixed $O_2$ and $CH_4$ atmosphere at room temperature with or without light irradiation, respectively. In situ DRIFTS tests were conducted by Thermo Scientific Nicolet IS52 in $CH_4$ / $CH_4 + O_2$ atmosphere, with or without $H_2O$ addition, with or without light irradiation. Atomic force microscopy (AFM) was performed on Bruker Dimension Icon.

### Photocatalytic oxidation of $CH_4$
The photocatalytic oxidation of $CH_4$ was performed in a stainless-steel autoclave with a quartz glass window on the top. All the photocatalysis experiments were carried out at room temperature along with 25 °C or 50 °C cooling water and a fixed pressure of 20 bar. In a typical experiment, the photocatalyst sample (10 mg) was weighted and added in the center of the reactor with specified amounts of deionized water. 20 bar $CH_4$ was inflated into reactor for the anaerobic reaction. Different ratio of $CH_4$ and $O_2$ with a total pressure of 20 bar was mixed and added in reactor for the aerobic reaction. A Xenon lamp (excitation wavelengths 300-700 nm, irradiation intensity of 150 mW cm$^{-2}$, CEAULIOHT) or a light-emitting diode monochromatic light source (Perfectlight) was used to initiate the photocatalytic reactions. For $^{18}O$- and D-isotope tests, to prevent the exchange of $\cdot OH$ between HCHO and $H_2O$ in aqueous solution, $H_2O$, $H_2^{18}O$ and $D_2O$ was added in the form of steam, respectively, with HCHO as the gaseous product for GC-MS tests.

### Product analysis
Analysis of the oxygenated liquid product was carried out using NMR spectroscopy. The $^1H$ NMR and $^{13}C$ NMR spectra were recorded using a Bruker AVANCE III HD 400 MHz NMR spectrometer. The amount of HCHO was quantified using the acetylacetone colour-development method. Gaseous products were qualitatively and quantitatively determined by gas chromatography (GC) tests with flame ionization detector (FID) and thermal conductivity detector (TCD). Test condition of GC: inlet temperature 100 °C, nitrogen as carrier gas with 0.1 MPa, column temperature of 60 °C, FID temperature of 100 °C, TCD bridge current of 60 mA.

The GC-mass spectrum (GC-MS) was performed on SHIMADZU with the SH-PolarWax column. Test condition: inlet temperature 180 °C, splitless inlet, helium as carrier gas, linear speed of 25.5 cm s$^{-1}$, column temperature of 40 °C with 120 °C pretreatment, GC-MS ion source temperature of 200 °C.

### Reporting summary
Further information on research design is available in the Nature Portfolio Reporting Summary linked to this article.

## Data availability
All data supporting the findings of this study are available within the paper, supplementary information files or are available from the corresponding authors upon request.

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

## Acknowledgements

This research was funded by the National Natural Science Foundation of China (22205043, Y.F.; 92056204, 21890381, 21721002, Z.T.; 22172040, D.H.; 21974031, L.N.), the Strategic Priority Research Program of Chinese Academy of Sciences (XDB36000000, Z.T.), National Key Basic Research Program of China (2016YFA0200700, Z.T.; 2022YFD2100304, D.H.), Frontier Science Key Project of Chinese Academy of Sciences (QYZDJ-SSW-SLH038, Z.T.), the Guangdong Science and Technology Department (2021A1515110705, Y.F.; 2021A1515010180, D.H.; 2019B010933001, 2022A156, L.N.), Guangzhou Municipal Science and Technology Project (202201020178, 202201010598, Y.F.; 202102010449, D.H.; 202201000002, L.N.), the Department of Guangdong Provincial Public Security (GZQC20-PZ11-FD084, D.H.), Key Discipline of Materials Science and Engineering, Bureau of Education of Guangzhou (20225546, L.N.), On-campus Scientific Research Project of Guangzhou University (RQ2021010, Y.F.), the Innovation Training Program for College Students of Guangzhou University (XJ202311078038, A.C.). The authors would like to thank Liu Jia from Shiyanjia Lab (www. shiyanjia.com) for the XPS analysis.

## Author contributions

Y.F., Y.J. and Z.T. conceived and designed the experimental scheme. Y.F., Y.J. and H.L. performed the key experiments, processed experimental data and wrote the paper. H.L., J.L., Y.X. and A.C. assisted to carry out the TEM, XRD, XPS, EPR and in situ DRIFTS characterization. S.L. performed the theoretical calculation. D.H., L.N. and Z.T. supervised the project. All the authors discussed the results and commented on the manuscript.

## Competing interests

The authors declare no competing interests.
