## [Peer Review File · Nature Communications]

Insight into selectivity of photocatalytic methane oxidation to formaldehyde on tungsten trioxideReviewers' comments:

Reviewer #1 (Remarks to the Author):

The study reported photocatalytic CH₄ oxidation to oxygenates over WO₃ {110} and WO₃ {001} and found that a ~100% HCHO selectivity was obtained over WO₃ {001} while a much lower selectivity was observed over WO₃ {110}. However, the manuscript is not written well and results presented in this manuscript cannot convincingly support their conclusions, and more importantly, some results/conclusions are confusing. The DFT calculations is not rigorous. I think that this manuscript does not meet the stringent criteria for publication on Nature Communications. Thus, I am very sorry to have to recommend rejection.

1. Scheme 1. The authors showed that methane oxidation could occur via mechanism 2, in which O₂ is reduced to O₂⁻. However, according to large amounts of reported work, WO₃ is unable to reduce O₂ to •O₂⁻ due to the relatively low conduction band potential, even in aqueous solution (Chem. Rev. 2017, 117, 11302–11336, page 11308, Catal. Sci. Technol. 2014, 4, 3850-3860). What is difference between this work and the reported studies? It is plausible that O₂ is directly reduced to H₂O₂ or •OOH on WO₃, instead of O₂⁻.

2. Photocatalytic CH₄ oxidation in the absence of O₂. First, the productivity of HCHO under this reaction condition was extremely low ($0.72 \times 15.70 \times 0.01 = 0.11304 \mu\text{mol}$). Such low productivity makes the conclusion questionable, because molecular oxygen absorbed on the surface of WO₃ probably participate into the reaction, which should not be neglected. On the other hand, if no O₂ participate into the reaction, what is the reduction product (HCHO is the oxidation product from CH₄)? Considering that only CH₄ or CH₄+H₂O participate into the reaction, the byproduct is probably H₂. If so, the author should detect the amount of H₂ produced during the reaction?

3. Photocatalytic CH₄ oxidation in the presence of O₂. At the reaction temperature of 25 degree, the highest productivity of HCHO over WO₃ {110} and WO₃ {001} is 1.1 μmol and 0.8 μmol , respectively, and the HCHO selectivities are likely quite similar (although the authors did not mention their selectivities). Again, differences in activity and selectivity are quite low, making the conclusion unconvincing. At 50 degree, both CH₃OOH and CH₃OH were detected only over WO₃ {110}, and the I notice that the productivity of CH₃OOH and CH₃OH is around 0.1-0.3 μmol . Can 1H NMR detect such low amount of CH₃OOH and CH₃OH? How about the limits of 1H NMR for detecting CH₃OOH and CH₃OH? When the water volume is 150 mL at 50 degree, the selectivities of HCHO over WO₃ {110} and WO₃ {001} are all 100%, and the amount of HCHO of WO₃ {110} is even higher than that of WO₃ {001}, which contradicts their conclusions in Abstract “{001} facet achieves 100% selectivity of HCHO in liquid product via the active site mechanism while {110} facet endows a low selectivity of HCHO...”. It is very confusing.

4. In situ DRIFTS spectra. In situ DRIFTS spectra were conducted without the addition of water, while the photocatalytic CH₄ conversion reaction was performed in water. Therefore, the data are not so meaningful.

5. DFT. The calculation and simulation parts are unconvincing and not rigorous enough. The WO₃ {110} surface used is not reasonable, since there are obvious O dangling bonds, making the surface unstable. The comparison of H₂O adsorption energy and -OCH₃ formation energy is also not reasonable, because the formation of OCH₃ group involve many steps such as the CH₄ adsorption, C-H bond dissociation and -OCH₃ formation, and the energy of one of them is probably higher than that of H₂O adsorption. Therefore, only comparing H₂O adsorption energy and the final energy of -OCH₃ formation is not meaningful. In addition, there is only one configuration of H₂O adsorption, other configuration should be considered such as such as the adsorption of O atom in H₂O at the W site.

6. Mechanism. As above discussion, if the DFT is inaccurate, the mechanism presented in this manuscript may be wrong. The valence band energy of WO₃ {001} is much higher than that of WO₃ {110}, thus, thermodynamically, strong driving force is obtained over WO₃ {001} to oxidize water to •OH radical, instead of merely oxidizing methane. In addition, the ¹⁸O isotope experiments are not rigorous, because the produced H₂¹⁸O is possibly from HCHO, but is definitely from the direct reduction of ¹⁸O₂ to water. Also, even using the recycled WO₃ {001}, the produced H₂¹⁸O is probably from the direct reduction of lattice ¹⁸O to water, instead of from HCHO.

Reviewer #2 (Remarks to the Author):

The manuscript revealed insights into formaldehyde selectivity on WO₃ for photocatalytic methane oxidation. The research direction belongs to a current budding topic and provides new revelations which can assist in better understanding the field. There are however some issues that should be addressed as stated below:

1. For Figure 1, FFT is incorrectly indexed.

2. For Figure 4, the caption “potential energy” is not terminologically correct. Based on the text, this should be more accurately depicted as adsorption energy.

3. Quantum efficiency is in the range ~10-4% which are relatively lower than reported literature values (J. Am. Chem. Soc. 2019, 141, 20507-20515; Nat. Commun. 2021, 12, 1-20) or past works by the authors (Nat. Sustain. 2021, 4, 509-515; Appl. Catal. B 2021, 283, 119661), even using a similar WO₃ material.

4. For Figure 3a-b and Figure 5a-b, DRIFT studies are rarely interpreted on the basis of negative axes/baseline. Particularly, findings pertaining to the generation of oxygen vacancies raises few questions and require clarifications, such as

- The mechanism behind the generation of oxygen vacancies is not explicitly discussed. Is it the same for WO₃{001} and WO₃{110}?
- Since, authors claim that oxygen source originates from lattice-O from WO₃{001}, shouldn't this directly connote that there is a higher lattice loss in WO₃{001} than in WO₃{110} after the reaction under pure CH₄ atmosphere? But this is not the case as revealed from the DRIFT/XPS results.
- To further confirm whether WO₃{110} Ov fails to be replenished in a mixed CH₄ + O₂ atmosphere, quantitative XPS data should also be provided as per Table S3.

5. Although the utilization routes for photogenerated electron-hole pairs are clearer for WO₃{110} which proceed through the radical process (e⁻ for superoxide generation and h⁺ for hydroxyl ions), it is unclear for WO₃{001} which proceed through active site routes. Where do the e⁻ and h⁺ pairs flow to for WO₃{001}?

6. Any reasons as to why ¹³C NMR was used for ¹³C isotope labeling experiments and not GCMS like in the case with ¹⁸O₂?

7. There should be discussion on the role of H₂O in the reaction system. Is it possible that H₂O can act as an O source as well? This possibility was neglected in the entire work.

8. There are several grammatical and typological errors which are present in the manuscript which include and are not limited to

- Figure 1 caption should provide descriptions for inset
- In Figure S3, 'calculation' should be 'calcination'
- On Page S15, 'carbohydrate' should be carbon oxygenates

Response to Reviewers

Dear Reviewers:

Thank you for your great effort and the reviewers' kind comments concerning on our manuscript entitled "Insight into selectivity of photocatalytic methane oxidation to formaldehyde on tungsten trioxide" (NCOMMS-22-40994). All the comments raised by you and reviewers are valuable and very helpful for us to improve the quality of the manuscript, as well as the important guiding significance to our researches. We have studied all the comments carefully and made comprehensive corrections that we hope will meet your approval. All the revised portions are marked in "red" in the revised manuscript and the revised supplementary information. The point-to-point responses to your and reviewer's comments are listed as following:

R1
R2

Referee #1

General comment: The study reported photocatalytic CH₄ oxidation to oxygenates over WO₃{110} and WO₃{001} and found that a ~100% HCHO selectivity was obtained over WO₃{001} while a much lower selectivity was observed over WO₃{110}. However, the manuscript is not written well and results presented in this manuscript cannot convincingly support their conclusions, and more importantly, some results/conclusions are confusing. The DFT calculations is not rigorous. I think that this manuscript does not meet the stringent criteria for publication on Nature Communications. Thus, I am very sorry to have to recommend rejection.

Our response: Thanks a lot for reviewer's careful reading and critical comments. According to the reviewer's kind comments, many verification experiments, spectroscopy characterizations and rigorous theoretical calculations have been carried out to support our conclusions. Besides, the logic and language of our manuscript have also been carefully polished to offer straightforward information for the readers. We expect that our revisions will meet your approval. Meanwhile, we suppose that this work can offer the insight into the different reaction mechanisms on CH₄ oxidation to HCHO over WO₃ system.

Question 1: Scheme 1. The authors showed that methane oxidation could occur via mechanism 2, in which O₂ is reduced to O₂^{•-}. However, according to large amounts of reported work, WO₃ is unable to reduce O₂ to O₂^{•-} due to the relatively low conduction band potential, even in aqueous solution (Chem. Rev. 2017, 117, 11302–11336, page 11308, Catal. Sci. Technol. 2014, 4, 3850-3860). What is difference between this work and the reported studies? It is plausible that O₂ is directly reduced to H₂O₂ or [•]OOH on WO₃, instead of O₂^{•-}.

Our response: We are grateful for the reviewer's insightful comment. Mechanism 2 in below Scheme 1 is summarized and deduced based on the previous reports (*Lin, R. et al. J. Am. Chem. Soc. 140, 9078–9082 (2018); Feng, N. et al. Nat. Commun. 12, 4652 (2021); Zeng, Y. et al. Appl. Catal. B: Environ. 306, 120919 (2022); Song, S. et al. Nat. Catal. 4, 1032–1042 (2021)*). The explanation is as follows. In the previous works, the signal of O₂^{•-} has been detected in WO₃ system (Figure R1a), and the protonated O₂^{•-} ([•]OOH) in WO₃ system combines with [•]CH₃ to form CH₃OOH (Figure R1b). Actually, the O₂^{•-} and [•]OOH radicals are the dissolved state of O₂^{•-} ions in neutral (or alkaline)

and acidic solutions (Figure R2, R3a and R3b) as the single reduction state of O_2 , respectively. Thus, it is reasonable that CH_4 oxidation to HCHO follows mechanism 2 in Scheme 1, in which O_2 is reduced to O_2^- on WO_3 surface (Scheme 1).

Scheme 1. Reported oxidation of CH_4 to HCHO on WO_3 photocatalyst following six types of reaction mechanisms.

Figure R1. (a) Detection of $\cdot O_2^-$ in $WO_3\{110\}$ and $WO_3\{001\}$ system (*Lin, R. et al. J. Am. Chem. Soc. 140, 9078–9082 (2018)*). (b) Signal of protonated $\cdot O_2^-$ ($\cdot OOH$), and $\cdot OOH$ combines with $\cdot CH_3$ to form CH_3OOH (*Zeng, Y. et al. Appl. Catal. B: Environ. 306, 120919 (2022)*).

Figure R2. Existing form of $\cdot\text{O}_2^-$. The single electron reduced O_2 exists in the form of $\cdot\text{O}_2^-$ in alkaline and neutral solutions or combines with H^+ to $\cdot\text{OOH}$ in acidic solution (Nosaka, Y. et al. Chem. Rev. 117, 11302–11336 (2017)).

Figure R3. (a) Single electron reduction of O_2 to $\text{O}_2^{\cdot-}$ ($\text{O}_2^{\cdot-}$) on the catalyst surface, (b) namely $\text{O}_2^{\cdot-}$ (or $\cdot\text{O}_2^-$) (Feng N., et al. Nat. Commun. 12, 4652 (2021); Song S., et al. Nat. Catal. 4, 1032–1042 (2021)).

There has been some controversy on whether $\cdot\text{O}_2^-$ radicals may be formed in WO_3 system, since the band potential of WO_3 would be changed by varying crystal structure, surface structure, defect type and particle size. Exactly as the reviewer stated, many papers have reported that the photoelectrons from conduction band of WO_3 are unable to reduce O_2 to $\cdot\text{O}_2^-$. Whereas, other reports have discovered that $\cdot\text{O}_2^-$ can be formed in WO_3 systems with specific crystal surface modification and defect introduction (Lin, R. et al. J. Am. Chem. Soc. 140, 9078–9082 (2018), Farhadian, M. et al. J. Energy Chem., 24, 171–177 (2015), Iliev, V. et al. J. Photochem. Photobiol. A: Chem., 159, 281–287 (2003), Noda, H. et al. Bull. Chem. Soc. Jpn., 66, 3542 (1993), Shiraishi, Y. et al. Catal. Sci. Technol. 2, 400–405 (2012), Tang, H. et al. ACS Sustain. Chem. Eng., 9, 15001506 (2021), Bao, K. et al. Catal. Today, 340, 311–317 (2020), Shang, J. et al. Nanotechnology, 31, 125603 (2020), Singh, S. et al. Mater. Sci. Forum, 855, 105–126 (2016)). Furthermore, as-formed $\cdot\text{OOH}$ radical (the protonated $\cdot\text{O}_2^-$ radical) in WO_3 system is explored to combine with $\cdot\text{CH}_3$ to form CH_3OOH (Figure R1b). So, even though the papers of Nosaka, Y. et al. Chem. Rev. 117, 11302–11336 (2017) and Tomita, O. Catal. Sci. Technol. 4, 3850–3860 (2014) propose that $\cdot\text{O}_2^-$ could not be generated in their mentioned WO_3 system, we cannot exclude the possibility of $\cdot\text{O}_2^-$ radicals generation in all types of WO_3 systems. To be specific, in both $\text{WO}_3\{110\}$ and $\text{WO}_3\{001\}$ systems, the signals of $\cdot\text{O}_2^-$ radical have been detected (Figure R1a). In our work, $\cdot\text{O}_2^-$ radical signal is only found in $\text{WO}_3\{110\}$ rather than $\text{WO}_3\{001\}$ system. This difference originates from the fact that the $\text{WO}_3\{001\}$ sample in our experiment is calcinated to form the distinct band structure.

As for the question of “It is plausible that O_2 is directly reduced to H_2O_2 or $\cdot\text{OOH}$ on WO_3 , instead of O_2^- ”, we would like to maintain the deduction that the $\cdot\text{OOH}$ radical in our work results from O_2^- that is one electron reduction product of O_2 on photocatalyst’s surface. The reason is as follows. In $\text{WO}_3\{110\}$ system, under O_2 condition, a distinct $\cdot\text{OOH}$ signal with sixfold peaks is detected (Figure R4a). To confirm that the $\cdot\text{OOH}$ radical in our reaction system is from the protonation of $\cdot\text{O}_2^-$ rather than the decomposition of H_2O_2 , we have tested the pH value of solution after the CH_4 oxidation reaction in O_2 atmosphere using $\text{WO}_3\{110\}$ as photocatalyst. The pH value of solution is tested to be 4.33 (Figure R4b). According to the previous reports (Nosaka, Y. et al. Chem. Rev. 117, 11302–11336 (2017), Bard, A. et al. Standard Potentials in Aqueous Solution. Macel Dekker: New York (1985)), the $\cdot\text{OOH}$ radical

generated in acid solution ($\text{pH} < 4.8$) results from the protonation of $\cdot\text{O}_2^-$ radical (Figure R2). And, the $\cdot\text{O}_2^-$ radical is the dissolved state of O_2^- ion in neutral (or alkaline) solution (Figure R3a and R3b). Both $\cdot\text{OOH}$ and $\cdot\text{O}_2^-$ originate from O_2^- anion, which is single electron reduction product of O_2 on the photocatalysts surfaces. The occurrence of single electron reduction of O_2 is attributed to its triplet state ($^3\Sigma_g^-$), which possesses two unpaired electrons on each of the two antibonding π orbitals (π_x^* and π_y^*) at the same energy level (Figure R5). This state of O_2 ensures its reactivity as a strong electron scavenger when adsorbed on the surface of catalyst, resulting in the formation of O_2^- anion. Noteworthily, the decomposition of H_2O_2 to produce OOH^- (the alkaline form of $\cdot\text{OOH}$ radical) can only occur in alkaline solutions ($\text{pK}_a = 11.7$, Figure R6). Furthermore, we cannot assume that the OOH^- anion plays the same role in CH_4 oxidation as the $\cdot\text{OOH}$ radical since no relevant work has been reported. In brief, we suggest that single electron reduction of O_2 on the $\text{WO}_3\{110\}$ surfaces produces O_2^- anions, then giving rise to $\cdot\text{OOH}$ radicals in acid solution. Besides, the signals of O_2^- anion (Figure R7a) and $\cdot\text{OOH}$ radical (Figure R4a) both have been detected in our $\text{WO}_3\{110\}$ system with the suitable energy band (Figure R7b). So, we are confident that $\cdot\text{OOH}$ radical in our reaction system is produced by O_2 reduction via O_2^- anion (Figure R5).

Figure R4. (a) EPR spectrum of $\text{WO}_3\{110\}$ under light irradiation for 80 s with CH_4 and O_2 dissolved in methanol. DMPO is added to the reaction mixture as the radical trapping agent. The $\text{WO}_3\{110\}$ is the recycled sample after CH_4 oxidation reaction without O_2 . (b) The pH value of solution after CH_4 oxidation reaction in O_2 atmosphere with $\text{WO}_3\{110\}$ as photocatalyst.

Figure R5. O₂ reduction process to form •OOH radical via O₂^{•-}.

Chem. Rev. 117, 11302–11336 (2017)

Among ROS, the chemical structures of •O₂⁻, H₂O₂, and •OH alter with pH of aqueous solution due to the rapid acid–base equilibrium as shown in eqs 1–3.³⁴

Nat. Commun. 9, 4060 (2018)

Fig. 3 Schematic illustration of water (left) and H₂O₂ (right) photo-oxidation reaction paths in alkaline solution. The former follows steps (1) to (5), whereas the latter follows steps (6) to (8) combined with step (1). In addition, the red arrow denotes the recombination reaction (2)

Figure R6. Decomposition of H₂O₂ to produce ⁻OOH (the alkaline form of •OOH radical), which only occurs in alkaline solutions (Nosaka, Y. et al. Chem. Rev. 117, 11302–11336 (2017); Avital, Y. et al. Nat. Commun. 9, 4060 (2018)).

Figure R7. (a) EPR spectra of $\text{WO}_3\{110\}$ in O_2 atmosphere at 77 K temperature. The $\text{WO}_3\{110\}$ is the recycled sample after CH_4 oxidation reaction without O_2 . (b) Band energy diagrams of $\text{WO}_3\{001\}$ and $\text{WO}_3\{110\}$.

Our revision: The generation origin of $\cdot\text{OOH}$ radical has been added in the revised supplementary information (please see the red marked parts on page S50). Figure R4a, R7a and R7b have been added as Figure 6b, 6a and 6c in the revised manuscript. Figure R4b has been added as Figure S44 in the revised supplementary information.

Question 2: Photocatalytic CH_4 oxidation in the absence of O_2 . First, the productivity of HCHO under this reaction condition was extremely low ($0.72 \times 15.70 \times 0.01 = 0.11304 \mu\text{mol}$). Such low productivity makes the conclusion questionable, because molecular oxygen absorbed on the surface of WO_3 probably participate into the reaction, which should not be neglected. On the other hand, if no O_2 participate into the reaction, what is the reduction product (HCHO is the oxidation product from CH_4)? Considering that only CH_4 or $\text{CH}_4 + \text{H}_2\text{O}$ participate into the reaction, the byproduct is probably H_2 . If so, the author should detect the amount of H_2 produced during the reaction?

Our response: We are grateful for the reviewer's insightful and critical comments. We totally agree that the productivity of HCHO on pristine WO_3 especially in absence of O_2 is quite low. This is also the reason why the currently reported papers on photocatalytic CH_4 oxidation rarely use pristine semiconductors, and most adopt the precious metal-supported semiconductors as photocatalysts. Though the CH_4 oxidation

efficiency may be improved by using precious metal-supported semiconductors as photocatalyst, the addition of precious metal nanoparticles would increase the variety of active site types. Therefore, the CH₄ oxidation pathways become more complicated, resulting in generation of varied products and decrease of the target product selectivity. Obviously, using pristine semiconductor as photocatalyst can simplify the CH₄ oxidation pathways and guarantee the selectivity of target products, which benefit understanding of photocatalytic mechanism.

We would like to politely point out that despite the yield of HCHO is low on pristine WO₃ in absence of O₂, the corresponding concentration is within a detectable range since the solvent volume is only 5 mL (note that the productivity of 0.72 μmol m⁻² indicated by the reviewer was obtained in 5 mL of reaction solution.). As shown in Figure R8a, the signals of HCHO are clear and demonstrate the increase trend along with the reaction time. More importantly, to ensure the accuracy of the data, we have repeated the experiment three times, which show good reproductivity (Figure R8b and R8c) and is reflected by small error bars (Figure 2a in the revised manuscript). This means that even though the HCHO yield is low in absence of O₂, the result is correct and repeatable. Also, similar or even lower concentrations of HCHO products have been reported in other CH₄ oxidation works (*Mao, J. et al. Nat. Catal. DOI: 10.1038/s41929-023-01030-2 (2023)*, *Tang, J. et al. Nat. Commun. 13, 2930 (2022)*, *Ye, J. et al. Nat. Commun. 12, 4652 (2021)*, *Zeng, Y. et al. Appl. Catal. B Environ. 306, 120919 (2022)*, *Wei, W. et al. Cell Rep. Phys. Sci. 3, 100909 (2022)*, *Hu, Y. et al. J. Phys. Chem. Lett. 12, 7459–7465 (2021)*, *Ye, J. et al. ACS Catal. 10, 14318–14326 (2020)*, *Khodakov, A. et al. Cell Rep. Phys. Sci. 4, 101277 (2023)*).

Figure R8. UV-visible absorption spectra of HCHO product on $\text{WO}_3\{001\}$ in CH_4 atmosphere with different reaction times for (a) first cycle, (b) second cycle and (c) third cycle.

We would like to politely point out that no adsorbed O_2 molecules exist on the surfaces of $\text{WO}_3\{001\}$ and $\text{WO}_3\{110\}$. This is because we have pretreated the $\text{WO}_3\{001\}$ and $\text{WO}_3\{110\}$ with calcination to remove the adsorbed O_2 . As shown in $\text{O}1s$ XPS spectra, both $\text{WO}_3\{001\}$ (Figure R9a) and $\text{WO}_3\{110\}$ (Figure R9b) possess the characteristic peak of surface adsorbed O_2 before calcination. To obtain the surface free of O_2 adsorption, the temperature-programmed desorption of O_2 experiments (O_2 -TPD) has been conducted. As displayed in Figure R10a, the peaks ranging from 100°C to 300°C in the spectra are ascribed to the desorption of chemisorbed O_2 molecules (*Song, S. et al. Nat. Catal. 7, 617–632 (2021), Iwamoto, M. et al. J. Phys. Chem. 82, 2564–2570 (1978)*), indicating that a temperature of 300°C is able to fully remove the adsorbed O_2 molecules from catalyst surface. After calcination at 300°C , the adsorbed O_2 is successfully removed (Figure R10b, R11a and R11b). Therein, the OH species results from the surface hydroxide (*Li, L. et al. Nat. Commun. 6, 5881 (2015), Sun, S. et al. J. Am. Chem. Soc. 140, 6474–6482 (2018), Zhang, N. et al. ACS Appl. Mater. Inter. 8, 10367–10374 (2016)*). It is noticed that calcination is a common method to remove adsorbed O_2 in the previous reports (*Chen, Z. et al. Adv. Energy Mater. 12,*

2103670 (2022); Ye, S. et al. *ACS Catal.* 11, 6104-6112 (2021)). Thus, we conclude that in our photocatalytic reaction with pure CH₄ atmosphere, no adsorbed O₂ molecules exist on the surface of WO₃{001} and WO₃{110}.

Figure R9. High-resolution O1s XPS spectra of (a) WO₃{001} and (b) WO₃{110} before calcination.

Figure R10. O₂-TPD spectra of WO₃{001} and WO₃{110} (a) before and (b) after calcination treatment at 300°C.

Figure R11. High-resolution O1s XPS spectra of (a) WO₃{001} and (b) WO₃{110} after 300°C calcination.

In the photocatalytic reaction of WO₃{001} system with pure CH₄ atmosphere, HCHO is the only oxidation product of CH₄ by lattice-O and no reduction product (such

as H₂) is found. Note that since no oxygenates generated in WO₃{110} system with pure CH₄ atmosphere, here WO₃{110} system is not discussed. In WO₃{001} system without O₂ participation, we infer that the unused electrons are trapped in WO₃{001} crystal, which is uncovered by high-resolution W4f XPS spectra (Figure R12). Compared to pristine WO₃{001} (FWHM = 1.0), the full width at half maximum (FWHM = 1.1) of W4f of WO₃{001} after reaction in pure CH₄ atmosphere becomes broader. This means that the peak of W4f after reaction is split. Indeed, besides the W₆₊ 4f_{5/2} and W₆₊ 4f_{7/2} peaks, the peaks of W₅₊ 4f_{5/2} and W₅₊ 4f_{7/2} are separated from broad W4f peak, indicating that excess electrons emerge and reduce W⁶⁺ to W⁵⁺ during CH₄ oxidation without O₂. The presence of excess electrons is also indirectly reflected from experimental phenomena. As shown in Figure R13, for the photocatalytic CH₄ oxidation without O₂ by WO₃{001}, along with the reaction time prolonging, the productivity of HCHO becomes also unchanged after 5 h (0.72 μmol m⁻²). We speculate that as the reaction proceeds, more and more residual photoelectrons are generated and rapidly recombine with photoholes. Without the separated photoholes, the lattice-O cannot be activated, thus preventing CH₄ oxidation to HCHO.

Figure R12. High-resolution W4f XPS spectra of WO₃{001} before and after photocatalytic CH₄ oxidation without O₂ addition.

Figure R13. Photocatalytic CH₄ oxidation performance on WO₃{001} and WO₃{110} in pure CH₄ atmosphere with reaction time prolonging at 25°C.

There is no H₂ generated in WO₃{001} system after CH₄ oxidation reaction in absence of O₂. Seen from the GC spectrum with thermal conductivity detector (TCD), taking pure H₂ as reference, the H₂ signal should be at 1.1 min. While, as shown in Figure R14a and the magnified image in Figure R14b, no H₂ signal is detected for the gas product over WO₃{001} after reaction for 7 h in CH₄ atmosphere without O₂ addition. This result proves that no H₂ is generated.

Figure R14. (a) GC signals of pure H₂ and gas product of WO₃{001} after reaction in pure CH₄ atmosphere for 7 h under light irradiation. (b) Magnified image of gas product from Figure R14a.

Our revision: The analyses of unused electrons trapped in WO₃{001} crystal and possible H₂ product have been added in the revised manuscript (please see the red marked parts on pages 16 and 6) and the revised supplementary information (please see

the red marked parts on page S71 and S18). Figure R11a, R11b and R13 have been added as Figure 1f, 1g and 2a in the revised manuscript. Figure R8a, R9, R10, R12 and R14 have been added as Figure S10a, S1, S2, S61 and S13 in the revised supplementary information.

Question 3: Photocatalytic CH₄ oxidation in the presence of O₂. At the reaction temperature of 25 degree, the highest productivity of HCHO over WO₃ {110} and WO₃ {001} is 1.1 μmol and 0.8 μmol, respectively, and the HCHO selectivities are likely quite similar (although the authors did not mention their selectivities). Again, differences in activity and selectivity are quite low, making the conclusion unconvincing. At 50 degree, both CH₃OOH and CH₃OH were detected only over WO₃ {110}, and the I notice that the productivity of CH₃OOH and CH₃OH is around 0.1-0.3 μmol. Can ¹H NMR detect such low amount of CH₃OOH and CH₃OH? How about the limits of ¹H NMR for detecting CH₃OOH and CH₃OH? When the water volume is 150 mL at 50 degree, the selectivities of HCHO over WO₃ {110} and WO₃ {001} are all 100%, and the amount of HCHO of WO₃ {110} is even higher than that of WO₃ {001}, which contradicts their conclusions in Abstract “{001} facet achieves 100% selectivity of HCHO in liquid product via the active site mechanism while {110} facet endows a low selectivity of HCHO...”. It is very confusing.

Our response: We are grateful for the reviewer’s careful reading and critical comments. We would like to politely point out that despite the small difference in the activity and selectivity of WO₃{001} and WO₃{110} detected at 25°C, the large difference in the activity and selectivity has been discerned at 50°C with different O₂ addition. Furthermore, the similar photocatalytic performance at 25°C does not mean the same reaction mechanism, since it might be affected by various reaction conditions (*Fan, Y. et al. Nat. Sustain. 4, 509–515 (2021)*; *Song, H. et al. J. Am. Chem. Soc. 141, 20507–20515 (2019)*; *Jiang, Y. et al. J. Am. Chem. Soc. 144, 15977–15987 (2022)*). Nevertheless, the large difference in the activity and selectivity at 50°C with different O₂ addition drives us to investigate possible difference in the photocatalytic mechanism over WO₃{001} and WO₃{110} systems.

Figure R15. Calibration curve for quantification of CH₃OH by ¹H NMR.

Figure R16. ¹H NMR spectra of the products generated over WO₃{110} in the atmosphere of 1 bar O₂ + 19 bar CH₄ at 50°C, which are screenshots from MestReNova software. Reaction condition: 10 mg catalyst, 1 bar O₂ + 19 bar CH₄, 5 mL H₂O volume, 3 h reaction time, Xenon light 150 mW cm⁻².

The detection limit of a Bruker AV III, Ascend 500 HD NMR spectrometer (Guangzhou University) for CH₃OOH and CH₃OH is around 1 $\mu\text{mol L}^{-1}$. In our work, the oxygenate with the lowest yield is CH₃OH (0.06 μmol) with a concentration of 11.04 $\mu\text{mol L}^{-1}$, which is produced in WO₃{110} system in the atmosphere of 1 bar O₂ + 19 bar CH₄ at 50°C. The corresponding reaction condition is as follows: 10 mg of WO₃{110} was dispersed in 5 mL of H₂O solution; then 1 bar O₂ and 19 bar CH₄ were applied for 3 h reaction at 50°C under 150 mW cm⁻² Xenon light irradiation; finally, the ¹H NMR was employed to detect the concentration of CH₃OOH and CH₃OH products. Below are the corresponding experimental data with calculation process.

Calibration curve for quantification of CH₃OH by ¹H NMR. Different concentrations of CH₃OH aqueous solutions (1 $\mu\text{mol L}^{-1}$, 10 $\mu\text{mol L}^{-1}$, 50 $\mu\text{mol L}^{-1}$, 100 $\mu\text{mol L}^{-1}$ and 200 $\mu\text{mol L}^{-1}$) were first prepared. Then, 500 μL above CH₃OH solution was mixed with 100 μL 0.7 mmol L⁻¹ DMSO/D₂O solution, which was taken as internal standard for ¹H NMR test. The standard curve was drawn with the peak area ratio of CH₃OH/DMSO as the vertical coordinate and the concentration of CH₃OH as the horizontal coordinate (Figure R15). Then, the standard curve equation was obtained as follows.

$$y = 3.64888x - 0.00028 \quad (\text{Equation 1})$$

where x was the concentration of CH₃OH and y was the peak area ratio of CH₃OH/DMSO. Since CH₃OOH and CH₃OH possess the same number of H atom, the same concentration of CH₃OOH and CH₃OH should have the same ¹H NMR peak area. Thus equation 1 is also applied to CH₃OOH measurement.

Measurement and calculation of the lowest CH₃OH yield in WO₃{110} system. After the photocatalytic CH₄ oxidation in 1 bar O₂ + 19 bar CH₄ atmosphere using WO₃{110} as photocatalyst, 500 μL production solution was taken out and mixed with 100 μL 0.7 mmol L⁻¹ DMSO/D₂O solution. After ¹H NMR test, the peak area ratio of CH₃OH/DMSO was measured to be 0.04 (Figure R16). Taken y = 0.04 into Equation 1, the CH₃OH concentration was calculated to be 11.04 $\mu\text{mol L}^{-1}$, which was converted to 0.33 $\mu\text{mol m}^{-2}$ (specific surface area is 16.63 m² g⁻¹). Reaction condition: 10 mg catalyst, 1 bar O₂ + 19 bar CH₄ atmosphere, 5 mL H₂O volume, 3 h reaction time, Xenon light 150 mW cm⁻².

When the H₂O volume increases to 150 mL at 50°C, despite the selectivity of HCHO over both WO₃{001} and WO₃{110} approaches 100%, the difference in the

product composition still exists. For $\text{WO}_3\{001\}$ system, owing to the active site reaction mechanism, CH_4 is directly oxidized to HCHO ($\text{CH}_4 \rightarrow \text{HCHO}$) without any intermediates and HCHO is the sole liquid product. Excessive solvent addition like 150 mL H_2O would completely avoid the overoxidation of HCHO , thus the HCHO selectivity is 100%. As comparison, for $\text{WO}_3\{110\}$ system, CH_3OOH intermediate gives birth to HCHO as well as CH_3OH byproduct ($\text{CH}_4 \rightarrow \text{CH}_3\text{OOH} \rightarrow \text{HCHO}$ or CH_3OH); since the Gibbs free energy (ΔG) of $\text{CH}_3\text{OOH} \rightarrow \text{HCHO}$ ($\Delta G = -145.74 \text{ kJ mol}^{-1}$) is much larger than that of $\text{CH}_3\text{OOH} \rightarrow \text{CH}_3\text{OH}$ ($\Delta G = 17.48 \text{ kJ mol}^{-1}$), HCHO is the major product. In 150 mL aqueous solution, the absence signals of CH_3OOH and CH_3OH in $\text{WO}_3\{110\}$ system is likely caused by their low concentrations rather than the actual disappearance. It is known that the ^1H NMR signal of CH_3OOH and CH_3OH especially at low concentration depends heavily on the intensity of water suppression peak. Therefore, by optimizing the parameters for strong water suppression, we have re-detected the signals of CH_3OOH and CH_3OH in 100 mL and 150 mL H_2O (Figure R17 and R19i). The selectivity of HCHO is estimated to be 73.73% in 150 mL H_2O . As comparison, for $\text{WO}_3\{110\}$ at 25°C (Figure R18a and R19d) and $\text{WO}_3\{001\}$ at both 25°C (Figure R18b and R19b) and 50°C (Figure R18c and R19c), the signals of CH_3OOH and CH_3OH cannot be discerned even with strong water suppression.

Figure R17. Signals of CH_3OOH and CH_3OH detected in (a) 100 mL and (b) 150 mL H_2O over $\text{WO}_3\{110\}$ at 50°C by optimizing the parameters for strong water suppression.

Figure R18. (a) ^1H NMR spectrum of product on $\text{WO}_3\{110\}$ with 9 bar O_2 and 11 bar CH_4 for 3 h reaction time at 25°C . (b) ^1H NMR spectrum of product on $\text{WO}_3\{001\}$ with 7 bar O_2 and 13 bar CH_4 for 3 h reaction time at 25°C . (c) ^1H NMR spectrum of product on $\text{WO}_3\{001\}$ with 7 bar O_2 and 13 bar CH_4 for 3 h reaction time at 50°C .

Figure R19. (a) Photocatalytic CH_4 oxidation performance on $\text{WO}_3\{001\}$ and $\text{WO}_3\{110\}$ in pure CH_4 atmospheres with reaction time prolonging at 25°C . (b) Photocatalytic CH_4 oxidation performance on $\text{WO}_3\{001\}$ at 25°C and (c) 50°C with variation of O_2 amounts. (d) Photocatalytic CH_4 oxidation performance on $\text{WO}_3\{110\}$ at 25°C and (e) 50°C with variation of O_2 amount. (f) Photocatalytic CH_4 oxidation performance on $\text{WO}_3\{001\}$ at 25°C and (g) 50°C with variation of H_2O amount. (h) Photocatalytic CH_4 oxidation performance on $\text{WO}_3\{110\}$ at 25°C and (i) 50°C with variation of H_2O amount. Reaction condition: (a-e) 10 mg catalyst, 3 h reaction time, Xenon light 150 mW cm^{-2} ,

5 mL water, pressures of CH₄ + O₂ = 20 bar; (f-i) 10 mg catalyst, 3 h reaction time, Xenon light 150 mW cm⁻², 7 bar O₂ + 13 bar CH₄ (WO₃{001}), 9 bar O₂ + 11 bar CH₄ (WO₃{110}).

The effect of reaction temperature on CH₄ oxidation productivity and selectivity in WO₃{001} and WO₃{110} systems can be attributed to their different kinetic properties, which originate from their distinct reaction mechanisms. According to the previous reports, the CH₄ oxidation rate coefficient (k) of WO₃{001} with active site mechanism is of an Arrhenius behavior (Equation 2, *Weaver, J. et al. Science* 356, 299–303 (2017); *Xiong, Y. et al. Nat. Commun.* 13, 2806 (2022); *Lampert, J. et al. Appl. Catal. B: Environ.* 14, 211-233 (1997); *Myshakin, E. et al. J. Phys. Chem. A* 113, 10 (2019); *Dicks, A. et al. J. Power Sources* 86, 523-530 (2000); *Zheng, H. et al. Biochemistry* 45, 6 (2006); *Brazeau, B. et al. Biochemistry* 39, 44 (2000); *Anderson, R. et al. Ind. Eng. Chem.* 53, 10 (1961); *Arutyunov, V. et al. Methanol Chapter 6, 129-172 (2018); Miller, J. et al. Appl. Catal. A: General* 495, 54-62 (2015); *Sabia, P. et al. Fuel* 91, 238-245 (2012)), whereas the k^* of WO₃{110} with radical mechanism should exhibit a non-Arrhenius dependence (Equation 3, *Zavitsas, A. et al. J. Am. Chem. Soc.* 120, 6578-6586 (1998); *Bonard, A. et al. J. Phys. Chem. A* 106, 4384-4389 (2002); *Pilgrim, J. et al. J. Phys. Chem. A* 101, 1873-1880 (1997); *Chen, C. et al. Can. J. Chem.* 54, 3175-3184 (1976); *Boukhalfa, N. et al. Procedia Engineering* 148, 1130–1136 (2016)). For both Arrhenius and non-Arrhenius behaviors, the reaction rate constant increases with the reaction temperature:

$$k = A \exp(-E_a/RT) \quad \text{(Equation 2)}$$

$$k^* = A^*T^n \exp(-E_a^*/RT) \text{ or } k^* = A^*T^6 \exp(-E_a^*/RT) \quad \text{(Equation 3)}$$

where k and k^* represent the rate constant, A and A^* are the preexponential factor, T is the reaction temperature, E_a and E_a^* are the activation energy, R is the molar gas constant, and n is as high as 6.

Based on the productivity of CH₄ oxidation within 3 h reaction time, the k values of WO₃{001} are calculated to be 2.89 $\mu\text{mol m}^{-2} \text{h}^{-1}$ (25°C) and 4.19 $\mu\text{mol m}^{-2} \text{h}^{-1}$ (50°C), so the E_a is calculated to be 11.89 kJ mol⁻¹ (Equation 2). Similarly, the k^* values of WO₃{110} are 2.06 $\mu\text{mol m}^{-2} \text{h}^{-1}$ (25°C) and 5.63 $\mu\text{mol m}^{-2} \text{h}^{-1}$ (50°C) respectively, and the corresponding E_a^* is calculated to be 16.71 kJ mol⁻¹. The higher value of E_a^* than E_a gives rise to the lower CH₄ oxidation performance of WO₃{110} (6.19 $\mu\text{mol m}^{-2}$ with 9 bar O₂, Figure R19d) than WO₃{001} (8.68 $\mu\text{mol m}^{-2}$ with 7 bar O₂, Figure R19b) at 25°C. However, non-Arrhenius behavior of k^* highly depends on T , thus the

maximum productivity over $\text{WO}_3\{110\}$ reaches $16.88 \mu\text{mol m}^{-2}$ at 50°C (Figure R19c) obviously higher than $\text{WO}_3\{001\}$ with $12.57 \mu\text{mol m}^{-2}$ (Figure R19e). The promoted reaction rate k^* of $\text{WO}_3\{110\}$ at 50°C also accelerates the formation of intermediate products, contributing to the appearance of CH_3OOH and CH_3OH signals.

In order to express more accurately, the sentence in Abstract has been revised to “ $\{001\}$ facet can readily achieve 100% selectivity of HCHO via the active site mechanism whereas $\{110\}$ facet hardly guarantees a high selectivity of HCHO along with many intermediate products via the radical way”.

Our revision: The explanation of experimental phenomena on $\text{WO}_3\{001\}$ and $\text{WO}_3\{110\}$ has been added in the revised manuscript (please see the red marked parts on page 6-8) and revised supplementary information (please see the red marked parts on page S4-S5). Figure R19 have been added as Figure 2 in the revised manuscript. Figure R17, R18a, R18b and R18c have been added as Figure S32, S20b, S16b and S17b in the revised supplementary information. The revised sentence in Abstract has been added in the revised manuscript (please see the red marked parts on page 1).

Question 4: In situ DRIFTS spectra. In situ DRIFTS spectra were conducted without the addition of water, while the photocatalytic CH_4 conversion reaction was performed in water. Therefore, the data are not so meaningful.

Our response: We are grateful for the reviewer’s careful reading and critical comment. Following the reviewer’s kind suggestion, *in situ* DRIFTS spectra were retested with addition of H_2O . Therein, 50 mg photocatalyst was pressed and fixed to the test bench. The CH_4 or $\text{CH}_4 + \text{O}_2$ gas with or without water vapor was flowed to create the desirable reaction atmosphere. Then *in situ* DRIFTS tests were conducted by Thermo Scientific Nicolet IS50 with or without light irradiation. Compared with the spectra in CH_4 or $\text{CH}_4 + \text{O}_2$ atmosphere, no new peak is observed in $\text{WO}_3\{001\}$ system after addition of H_2O (Figure R20 and R21), indicating that H_2O molecule is not involved in CH_4 oxidation process. It is worth mentioning that H_2O is also a key species for determining the catalytic performance in $\text{WO}_3\{001\}$ system, which can desorb the oxygenates from the photocatalyst surface and prevent the overoxidation. The peaks at 917 , 1363 and 2830 cm^{-1} on $\text{WO}_3\{001\}$ are attributed to the stretching vibration of C-H bond in the absorbed $^*\text{OCH}_3$ species, while the ones at 1149 and 1478 cm^{-1} are assigned to the vibration of the absorbed $^*\text{CH}_2$ species. Both $^*\text{OCH}_3$ and $^*\text{CH}_2$ are believed as the

crucial intermediates upon the direct CH₄ oxidation to HCHO. The peaks at 1251 and 1810 cm⁻¹ are ascribed to the adsorbed HCHO* and C=O* species, further verifying the formation of HCHO. Note that the adsorbed HCOO* at 1381 and 1594 cm⁻¹ might be the intermediate for overoxidation to CO₂. Noteworthily, the consumption of lattice-O is evidenced by continuous descending of the W-O peak at 989 cm⁻¹ below zero baseline in pure CH₄ atmosphere (insets in Figure R20). Clearly, CH₄ is steadily oxidized to HCHO by lattice-O of WO₃{001}. After O₂ addition, the W-O peak raises above the zero baseline (insets in Figure R21), signifying that the consumed lattice-O of WO₃{001} is replenished adequately in O₂ atmosphere. For WO₃{110}, the addition of H₂O brings no obvious change of DRIFTS spectra in pure CH₄ atmosphere (Figure R22). The adsorbed intermediates are not discerned upon light irradiation, indicating that no CH₄ oxidation product is formed in pure CH₄ atmosphere despite H₂O addition. This result is consistent with the catalytic performance of WO₃{110} in pure CH₄ atmosphere (Figure R13). It is worth noting that the loss of W-O bonds in pure CH₄ atmosphere in absence of H₂O is attributed to lattice-O participated CH₄ activation but without adsorbed *OCH₃ formation, which is verified by DFT results (Figure R25b and R25d in below response to Question 5). As comparison, in the system containing CH₄ and H₂O, the loss of W-O bonds is ascribed to the H₂O oxidation by lattice-O, which is revealed by DFT results (Figure R25b and R26b in below response to Question 5). In CH₄ + O₂ atmosphere, compared to the H₂O free condition (Figure R23a), signals of *OCH₃, C=O*, *CH₂ and HCHO* groups appear and considerably grow after H₂O addition (Figure R23b). This observation demonstrates that the photocatalytic CH₄ oxidation reaction on WO₃{110} is implemented through a radical process, and the addition of H₂O enables such a radical reaction pathway happening.

Figure R20. (a) *In situ* DRIFTS spectra of WO₃{001} test in CH₄ atmosphere without or (b) with H₂O addition under different light irradiation time. Here, * denotes an adsorption site on surface. The inset is the magnified W-O peak.

Figure R21. (a) *In situ* DRIFTS spectra of $\text{WO}_3\{001\}$ test in $\text{CH}_4 + \text{O}_2$ atmosphere without or (b) with H_2O addition under different light irradiation time. Here, * denotes an adsorption site on surface. The inset is the magnified $\text{W}-\text{O}$ peak.

Figure R22. (a) *In situ* DRIFTS spectra of $\text{WO}_3\{110\}$ test in CH_4 atmosphere without or (b) with H_2O addition under different light irradiation time. Here, * denotes an adsorption site on surface.

Figure R23. (a) *In situ* DRIFTS spectra of $\text{WO}_3\{110\}$ test in $\text{CH}_4 + \text{O}_2$ atmosphere without or (b) with H_2O addition under different light irradiation time. Here, * denotes an adsorption site on surface. The inset is the magnified W-O peak.

Our revision: The analyses of *in situ* DRIFTS spectra with H_2O addition have been added in the revised manuscript (please see the red marked parts on pages 8-11). Figure R20a, R21a, R22a and R23a have been added as Figure 3a, 4a, 3b and 4b in the revised manuscript. Figure R20b, R21b, R22b and R23b have been added as Figure S37a, S38a, S37b and S38b in the revised supplementary information.

Question 5: DFT. The calculation and simulation parts are unconvincing and not rigorous enough. The $\text{WO}_3\{110\}$ surface used is not reasonable, since there are obvious O dangling bonds, making the surface unstable. The comparison of H_2O adsorption energy and $-\text{OCH}_3$ formation energy is also not reasonable, because the formation of OCH_3 group involve many steps such as the CH_4 adsorption, C-H bond dissociation and $-\text{OCH}_3$ formation, and the energy of one of them is probably higher than that of H_2O adsorption. Therefore, only comparing H_2O adsorption energy and the final energy of $-\text{OCH}_3$ formation is not meaningful. In addition, there is only one configuration of

H₂O adsorption, other configuration should be considered such as such as the adsorption of O atom in H₂O at the W site.

Our response: We are grateful for the reviewer's insightful and critical comments. In the previous theoretical calculation models of WO₃, the purpose of adding O dangling bonds is to achieve a ratio of W atoms to O atoms as 1:3. If the O dangling bonds are removed, the exposed sites are changed to the W atoms, which are also unstable due to the unsaturated coordination. According to the high-resolution O1s XPS spectra of WO₃{001} (Figure R24a) and WO₃{110} (Figure R24b), the surface OH species are observed, which come from the protonated O dangling bonds. Thus, we have modified our models by introducing surface OH species.

Figure R24. High-resolution O1s XPS spectra of (a) WO₃{001} and (b) WO₃{110}.

Figure R25. Energy diagrams of CH_4 and H_2O adsorption as well as CH_4 activation on the surface of (a) $\text{WO}_3\{001\}$ and (b) $\text{WO}_3\{110\}$. Atomic configurations for the corresponding steps in the simulation of (c) $\text{WO}_3\{001\}$ and (d) $\text{WO}_3\{110\}$.

Figure R26. Atomic configurations of H_2O adsorption on the surface of (a) $\text{WO}_3\{001\}$ and (b) $\text{WO}_3\{110\}$.

Figure R27. EPR spectra of (a) $\text{WO}_3\{001\}$ and (b) $\text{WO}_3\{110\}$ in pure CH_4 or $\text{CH}_4 + \text{O}_2$ atmosphere at room temperature with DMPO as the radical trapping agent in aqueous solution.

Figure R28. Energy diagrams of H_2O oxidation on the surface of (a) $\text{WO}_3\{001\}$ and (b) $\text{WO}_3\{110\}$. The energy value of $\text{V}_o + 2\text{OH} (\text{free})$ and $^*\text{H} + \text{OH} (\text{free})$ on $\text{WO}_3\{001\}$ is corrected with $U_e = 2.78 \text{ eV}$. The energy values of $\text{V}_o + 2\text{OH} (\text{free})$ on $\text{WO}_3\{110\}$ is corrected with $U_e = 2.52 \text{ eV}$.

Figure R29. Atomic configurations of (a) $\text{V}_o + 2\text{OH} (\text{free})$ and (b) $^*\text{H} + \text{OH} (\text{free})$ at O_b site on the surface of $\text{WO}_3\{001\}$, and (c) $\text{V}_o + 2\text{OH} (\text{free})$ at O_b site on the surface of $\text{WO}_3\{110\}$.

To explain the distinct CH₄ oxidation process on WO₃{001} and WO₃{110}, DFT recalculations have been performed to examine their abilities responsible for CH₄ and H₂O adsorption as well as activation. Bridging O (O_b), terminal O (O_t) and W atoms are taken as the adsorption sites of CH₄, respectively. It turns out that the O_b sites (O-IN 2) in both WO₃{001} (Figure R25a and R25c) and WO₃{110} (Figure R25b and R25d) exhibit the lower CH₄ adsorption energy than O_t(O-IN 1) and W (W-IN 1) sites. This result indicates that CH₄ molecules are preferentially adsorbed at the O_b sites. To be specific to WO₃{001}, through CH₄ activation (O-IN 4), a *CH₃ group is formed and firmly adsorbed on its O_b site to generate *OCH₃ group. Such species is confirmed by the rising *OCH₃ signals in *in situ* DRIFTS spectra of WO₃{001} in CH₄ or CH₄ + O₂ atmosphere under light irradiation (Figure R20 and R21). The large positive energy of the intermediates of O-IN 3 and CH₃(free) means that the *CH₃ group is hardly desorbed from WO₃{001} surface. This deduction is proved by the EPR test of WO₃{001} in CH₄ or CH₄ + O₂ atmosphere, where no •CH₃ radical is observed (Figure R27a). Alternatively, the •OH radical is derived from the adsorption of H₂O molecules at O_b site (O-IN 5, Figure R25a and R26a) with subsequent oxidation, which is not involved in CH₄ oxidation as the corresponding signals of *in situ* DRIFTS display no change before and after H₂O addition (Figure R20 and R21). Thus, the CH₄ oxidation process of WO₃{001} for HCHO generation goes through an active site mechanism rather than a radical mechanism. To explore whether the H₂O oxidation on WO₃{001} surface affects the CH₄ oxidation process through the consumption of O_b site, the •OH formation mechanism has been investigated. According to previous reports, the formation of •OH radical via H₂O oxidation can be divided into two ways: one is that both O_b and photohole participate in H₂O oxidation to generate one oxygen vacancy and two •OH radicals (O_b + h⁺ + H₂O → V_o + 2•OH, *Nosaka, Y. et al. Chem. Rev. 117, 11302–11336 (2017); Hou, J. et al. J. Am. Chem. Soc. 134, 9978–9985 (2012); Salvador, P. et al. Prog. Surf. Sci. 86, 41–58 (2011)*); the other is simply hole oxidizing H₂O to produce one H⁺ cation and one •OH radical without O_b consumption (h⁺ + H₂O → H⁺ + •OH, *Nosaka, Y. et al. Chem. Rev. 117, 11302–11336 (2017); Jonsson, M. et al. J. Phys. Chem. C 118, 10083–10087 (2014)*). As shown in Figure R28a and R29a, the positive energy value (0.36 eV) of V_o + 2OH (free) on WO₃{001} reveals that the

- OH radical formation is not through the process of O_b + h⁺ + H₂O → V_o + 2•OH. While

the negative energy value (-0.91 eV, Figure R28a and R29b) of *H + OH (free) confirms

that the $\cdot\text{OH}$ radical is generated through $\text{h}^+ + \text{H}_2\text{O} \rightarrow \text{H}^+ + \cdot\text{OH}$ process without O_b consumption. Therefore, the H_2O oxidation on $\text{WO}_3\{001\}$ surface does not change the CH_4 oxidation process as no O_b is consumed.

As for $\text{WO}_3\{110\}$, Figure R25b and R25d uncover that after CH_4 activation at O_b site ($\text{O-IN 3} + \text{CH}_3(\text{free})$), the adsorbed $\cdot\text{CH}_3$ group is hardly formed and easily desorbed to form $\text{CH}_3(\text{free})$ leaving $\text{O}_\text{b}\text{-H}$ group. Thus, the active site mechanism of CH_4 oxidation does not occur on $\text{WO}_3\{110\}$. Due to the high surface energy of $\text{WO}_3\{110\}$ (4.13 J m^{-2}), the $\text{O}_\text{b}\text{-H}$ is also easily desorbed to form oxygen vacancy, which is similar to the H_2O oxidation to form oxygen vacancy and $\cdot\text{OH}$ radical discussed later. This is consistent with the *in situ* DRIFTS results in CH_4 atmosphere without H_2O addition, where no CH_4 oxidation is discerned only with the detected loss of W-O signal (Figure R22a). Actually, in an aqueous reaction environment, due to the large adsorption energy difference between H_2O and CH_4 molecules on the surface of $\text{WO}_3\{110\}$ (O-IN 5 , Figure R25b and R26b), O_b preferentially adsorbs H_2O molecules to block the adsorption of CH_4 . And as displayed in Figure R28b and R29c, the adsorbed H_2O molecule is easily oxidized to $\cdot\text{OH}$ radical through $\text{O}_\text{b} + \text{h}^+ + \text{H}_2\text{O} \rightarrow \text{V}_\text{o} + 2\cdot\text{OH}$

with energy of -1.24 eV ($\text{V}_\text{o} + 2\text{OH}(\text{free})$). During this process, O_b is consumed along with oxygen vacancy generation, further excluding the possibility of CH_4 oxidation at O_b active site. This is proved by the signal of W-O consumption in absence of CH_4 activation in *in situ* DRIFTS spectra of $\text{WO}_3\{110\}$ in CH_4 atmospheres with H_2O addition (Figure R22b). Thereby, only $\cdot\text{OH}$ radical without $\cdot\text{CH}_3$ radical is observed in pure CH_4 atmosphere for $\text{WO}_3\{110\}$ aqueous system (Figure R27b). In short, we confirm that the CH_4 oxidation process over $\text{WO}_3\{110\}$ for HCHO generation is not implemented through an active site mechanism. Note that the signals of $\cdot\text{CH}_3$ radicals appear in $\text{WO}_3\{110\}$ system with $\text{CH}_4 + \text{O}_2$ atmosphere, which results from oxidation of free CH_4 by substantial O_2^- ions. The large adsorption energy difference of H_2O molecules over $\text{WO}_3\{001\}$ and $\text{WO}_3\{110\}$ stems from the number of hydrogen bonds formed. On $\text{WO}_3\{001\}$ surface, H_2O molecule is adsorbed through one hydrogen bond, while on $\text{WO}_3\{110\}$ surface, two hydrogen bonds are formed after H_2O adsorption. Therefore, $\text{WO}_3\{110\}$ has a higher adsorption capacity for H_2O molecules than $\text{WO}_3\{001\}$. Minor modifications have been made on the reaction pathways of CH_4 oxidation on $\text{WO}_3\{001\}$ and $\text{WO}_3\{110\}$, which are discussed in below response to Question 6.

Computational method. The Vienna Ab Initio Simulation Package was used to carry out all the density functional theory calculations within the generalized gradient approximation using the Perdew-Burke-Ernzerh of formulation. The projected augmented wave potentials were selected to describe the ionic cores. Valence electrons were taken into account by using a plane-wave basis set with a kinetic energy cutoff of 400 eV. The electronic energy was considered self-consistent when the energy change was smaller than 10^{-5} eV. Partial occupancies of the Kohn-Sham orbitals were allowed using the Gaussian smearing method and a width of 0.02 eV. Geometry optimization was considered convergent when the energy change was smaller than 0.05 eV/Å. The Brillouin zone was sampled with $2 \times 2 \times 1$ Monkhorst mesh. A 15 Å vacuum space along the z direction was added to avoid the interaction between the two neighboring images. For the H₂O oxidation process on both WO₃{001} and WO₃{110}, since one photoelectron was left after H₂O oxidation by photohole, the formation energy of V_o + 2OH (free) and *H + OH (free) modes took account of one photoelectron energy (Watson, G. et al. *Solid State Ionics* 177, 3069–3074 (2006); Nolan, M. et al. *Chem. Phys. Lett.* 499, 126–130 (2010); Xiong, Y. et al. *Nat. Commun.* 13, 2806 (2022)). The energy of a photogenerated electron was equal to the band gap energy, which excited electron from valence band to conduction band. Thus, the energy values of V_o + 2OH (free) and *H + OH (free) on WO₃{001} and WO₃{110} were corrected with $U_e = 2.78$ eV and $U_e = 2.52$ eV, respectively.

Our revision: The analyses based on DFT calculation have been added in the revised manuscript (please see the red marked parts on pages 11-13) and the revised supplementary information (please see the red marked parts on pages S4). Figure R24a, R24b and R25 have been added as Figure 1f, 1g and 5 in the revised manuscript. Figure R26, R27, R28 and R29 have been added as Figure S41, S40, S42 and S43 in the revised supplementary information.

Question 6: Mechanism. As above discussion, if the DFT is inaccurate, the mechanism presented in this manuscript may be wrong. The valence band energy of WO₃ {001} is much higher than that of WO₃ {110}, thus, thermodynamically, strong driving force is obtained over WO₃ {001} to oxidize water to •OH radical, instead of merely oxidizing methane. In addition, the ¹⁸O isotope experiments are not rigorous, because the produced H₂¹⁸O is possibly from HCHO, but is definitely from the direct reduction of

$^{18}\text{O}_2$ to water. Also, even using the recycled $\text{WO}_3\{001\}$, the produced H_2^{18}O is probably from the direct reduction of lattice ^{18}O to water, instead of from HCHO.

Our response: We are grateful for the reviewer's insightful and critical comments. Based on the updated DFT calculations as well as *in situ* DRIFTS, EPR and XPS results, the reaction mechanisms for $\text{WO}_3\{001\}$ (Figure R30a) and $\text{WO}_3\{110\}$ (Figure R30b) have been partially modified. In the both cases of $\text{WO}_3\{001\}$ and $\text{WO}_3\{110\}$, the CH_4 molecules are preferentially attached to the lattice- O_b , which are confirmed by the DFT calculations (Figure R25a and R25b). Upon photoirradiation, the lattice- O_b of $\text{WO}_3\{001\}$ loses one electron to be activated. Afterwards, the activated lattice- O_b of $\text{WO}_3\{001\}$ is able to insert into the C-H bond of CH_4 molecule, sequentially forming the $^*\text{OCH}_3$ and $^*\text{CH}_2$ species as shown in *in situ* DRIFTS spectra of $\text{WO}_3\{001\}$ in CH_4 or $\text{CH}_4 + \text{O}_2$ atmospheres (Figure R20 and R21). Meanwhile the left H atom from CH_4 is abstracted by adjacent -OH on W site via hydrogen atom transfer (HAT) process as shown in DFT results (O-IN4, Figure R25c). Finally, the $^*\text{OCH}_2$ species is desorbed to form HCHO molecule. During this process, no $^*\text{CH}_3$ radical is produced, which is verified by the EPR (Figure R27a) and DFT calculation results. In short, the HCHO is generated through an active site mechanism rather than a radical mechanism in $\text{WO}_3\{001\}$ system. The consumed lattice- O_b becomes an oxygen vacancy, which is conducive to the adsorption of O_2 molecule as shown by the XPS results (Figure 3c in the revised manuscript). The adsorbed O_2 molecule is then reduced by photoelectrons to fix the depleted lattice- O_b , which is confirmed by the XPS spectra (Figure S39a and Table S4 in the revised supplementary information) after CH_4 oxidation in $\text{CH}_4 + \text{O}_2$ atmosphere and *in situ* DRIFTS spectra of $\text{WO}_3\{001\}$ in $\text{CH}_4 + \text{O}_2$ atmosphere (Figure R21). As for $\text{WO}_3\{110\}$ (Figure R30b), though the adsorbed CH_4 molecules can be activated by lattice- O_b , the adsorbed $^*\text{CH}_3$ group is hardly formed. Moreover, in aqueous reaction environment, the large adsorption energy of H_2O at the O_b site further inhibits the above CH_4 activation process, which is verified by the DFT result (Figure R25b). Upon H_2O adsorption, the $^*\text{OH}$ radical is generated through $\text{O}_b + \text{h}^+ + \text{H}_2\text{O} \rightarrow$

$\text{V}_o + 2^*\text{OH}$ with O_b consumption and V_o formation (Figure R28b, R29c, R27b, R22b and 3d in the revised manuscript). With O_2 addition, the adsorbed O_2 molecule at the site of oxygen vacancy can be reduced to repair the left lattice-O atom or generate O_2^- anion by the photoelectron from the conduction band of $\text{WO}_3\{110\}$. Compared with the four-electron oxygen reduction to repair lattice-O, the one-electron oxygen

reduction in O_2^- generation pathway has lower kinetic energy barrier. Therefore, for $WO_3\{110\}$, the consumed lattice-O of $WO_3\{110\}$ is only partially repaired, and O_2 is mainly involved in the formation of O_2^- anion as shown in Figure R7a. No signal of O_2^- anion is detected on $WO_3\{001\}$ (Figure S57a in the revised supplementary information) because its lower conduction band energy (0.08 V vs NHE) than O_2 reduction potential to O_2^- anion (-0.046 V vs NHE). The O_2^- anion on $WO_3\{110\}$ activates CH_4 molecule to produce $\cdot CH_3$ and $\cdot OOH$ radicals as proved by the EPR results (Figure R27b and 6b in the revised manuscript). Through combination between

- CH_3 and $\cdot OOH$ radicals, CH_3OOH is generated followed by decomposition to CH_3OH and $HCHO$. Because the Gibbs free energy (ΔG) of $CH_3OOH \rightarrow HCHO$ ($\Delta G = -145.74 \text{ kJ mol}^{-1}$) is much larger than that of $CH_3OOH \rightarrow CH_3OH$ ($\Delta G = 17.48 \text{ kJ mol}^{-1}$), $HCHO$ become the major product.

To be specific for $WO_3\{001\}$, the adsorption and oxidation of H_2O to $\cdot OH$ radical at the O_b site are the competitive reaction of CH_4 adsorption and oxidation toward photohole consumption. Based on the DFT calculation (Figure R28a), the $\cdot OH$ radical generation is through a process of $h^+ + H_2O \rightarrow H^+ + \cdot OH$ without O_b participation.

Therefore, the H_2O oxidation on $WO_3\{001\}$ surface will not change the CH_4 oxidation mechanism. Besides, the $\cdot OH$ radical is not involved in CH_4 oxidation to $HCHO$, which is verified by *in situ* DRIFTS results that no signal change happens before and after H_2O addition in both CH_4 and $CH_4 + O_2$ atmosphere (Figure R20 and R21). Therefore, the H_2O oxidation and $\cdot OH$ radical generation are not mentioned in the proposed CH_4 oxidation mechanism on $WO_3\{001\}$.

Figure R30. Proposed reaction mechanisms for photocatalytic CH_4 oxidation on (a) $WO_3\{001\}$ and (b) $WO_3\{110\}$.

Figure R31. (a) GC-MS spectra of HCHO obtained in $\text{WO}_3\{001\}$ system using ${}^{18}\text{O}_2$ as oxygen isotope and the recycled $\text{WO}_3\{001\}$ as photocatalyst without O_2 addition. (b) GC-MS spectra of HCHO obtained in $\text{WO}_3\{110\}$ system using ${}^{18}\text{O}_2$ or H_2^{18}O as oxygen isotope. (c) GC-MS spectra of HCHO obtained in $\text{WO}_3\{001\}$ system using H_2^{18}O as oxygen isotope and the recycled $\text{WO}_3\{001\}$ as photocatalyst without O_2 addition.

Figure R32. (a) Screenshot on GC-MS spectra of HCHO obtained in WO₃{001} system using ¹⁸O₂ as oxygen isotope and (b) the recycled WO₃{001} as photocatalyst without O₂ addition.

Figure R33. (a) Screenshot on GC-MS spectra of HCHO obtained in WO₃{110} system using ¹⁸O₂ or (b) H₂¹⁸O as oxygen isotope.

Figure R34. Screenshot on GC-MS spectra of HCHO obtained in $\text{WO}_3\{001\}$ system using H_2^{18}O as oxygen isotope and the recycled $\text{WO}_3\{001\}$ as photocatalyst without O_2 addition.

Actually, the most convincing way to trace the source of O in HCHO is to use oxygen isotope and perform the corresponding gas chromatography-mass spectrometry (GC-MS) tests. However, because the previous column of GC-MS was not suitable for HCHO separation, the trace test for O element of HCHO was failed. Hence, we had to extract the O-atom of HCHO into H_2O molecule for GC-MS test, which was taken as indirect evidence for the O-atom source of HCHO. Now, we fortunately find that the SH-PolarWax column from SHIMADZU can separate HCHO and a satisfactory GC-MS signals have been achieved. Test conditions: inlet temperature 180°C , splitless inlet, helium as carrier gas, linear speed of 25.5 cm s^{-1} , column temperature of 40°C with 120°C pretreatment, GC-MS ion source temperature of 200°C .

The origin of O-atoms in HCHO is then traced by isotope labeling experiments with $^{18}\text{O}_2$ and H_2^{18}O . In both $\text{WO}_3\{001\}$ (Figure R31a and R32) and $\text{WO}_3\{110\}$ (Figure R31b and R33) systems, HCH^{18}O peaks ($m/z = 32$) are found taking $^{18}\text{O}_2$ as oxygen source while H_2^{18}O makes no difference (Figure R31c, R34 and R31b, R33), indicating that the O-atom of HCHO is originated from O_2 instead of H_2O . After the CH_4 oxidation reaction in $^{18}\text{O}_2$ atmosphere or taking H_2^{18}O as solvent, the $\text{WO}_3\{001\}$ photocatalyst has been recycled and put into another CH_4 oxidation system without O_2 addition but with H_2O as solvent. Taking the recycled $\text{WO}_3\{001\}$ from $^{18}\text{O}_2$ atmosphere as

photocatalyst (Figure R31a and R32), the clear signal of HCH^{18}O confirms that the lattice-O atom of $\text{WO}_3\{001\}$ indeed participates in the formation of HCHO and can be supplemented by O_2 . On the contrary, taking the recycled $\text{WO}_3\{001\}$ from H_2^{18}O solvent as the photocatalyst (Figure R31c and R34), only HCHO without ^{18}O labelling is found, which proves that the consumed lattice-O of $\text{WO}_3\{001\}$ cannot be repaired by H_2O . Additionally, we note that no product is found in pure CH_4 atmosphere using recycled $\text{WO}_3\{110\}$ as photocatalyst without O_2 addition, which is reasonable considering that the consumed lattice-O of $\text{WO}_3\{110\}$ is hardly repaired. Based on the oxygen isotope experiments, we conclude that the O-atoms of HCHO products on $\text{WO}_3\{001\}$ and $\text{WO}_3\{110\}$ are both from O_2 rather than H_2O . Besides, for $\text{WO}_3\{001\}$ system, O_2 is involved in the formation of HCHO by repairing the lattice-O.

Our revision: The mechanism explanation has been added in the revised manuscript (please see the red marked parts on pages 15-17). The analyses of GC-MS results have been added in the revised manuscript (please see the red marked parts on pages 17-18) and the revised supplementary information (please see the red marked parts on page S4). Figure R30, R31a, R31b, R31c have been added as Figure 7, 8b, 8c, 8d in the revised manuscript. Figure R32-R34 have been added as Figure S64-S66 in the revised supplementary information.

Reviewer #2

General comment: The manuscript revealed insights into formaldehyde selectivity on WO_3 for photocatalytic methane oxidation. The research direction belongs to a current budding topic and provides new revelations which can assist in better understanding the field. There are however some issues that should be addressed as stated below: **Our response:** We truly appreciate the reviewer's encouraging comments. All of the questions raised by the reviewer have been carefully thought and answered, which greatly improves the quality of our work.

Question 1: For Figure 1, FFT is incorrectly indexed.

Our response: We are grateful for the reviewer's careful reading and professional comment. Following the reviewer's kind suggestion, the FFT pattern on $\text{WO}_3\{001\}$ has been relabeled. The diffraction spots in the FFT pattern (below Figure R35a inset) have been calculated to be (100) and (110) facets with d values as 0.70 nm and 0.37 nm, and the interior angle of 60° is in agreement with the theoretical value of the angle between the normal directions of (110) and (100) planes (Figure R35b). The presence of (100) and (110) facets indicates that the exposed crystal surface of WO_3 nanosheet is $\{001\}$ with the perpendicularly transmitted electron beam. Besides, we note that the FFT pattern of $\text{WO}_3\{110\}$ is correctly indexed (Figure R35c). The interior angle labeled in the FFT pattern between diffraction spots of (002) and (110) facets is 90° (Figure R35c inset), which is in agreement with the theoretical value of the angle between the normal directions of (110) facet and c axis (Figure R35b). Therein, the c axis is considered as the normal direction of (002) facet.

Figure R35. (a) HRTEM image of WO₃{001} with FFT image (inset). (b) Geometrical configuration of hexagonal WO₃ crystal. (c) HRTEM image of WO₃{110} with FFT image (inset).

Our revision: The FFT has been corrected in the revised manuscript (please see the red marked parts on pages 3-4). Figure R35a, R35c have been added as Figure 1d, 1e in the revised manuscript. Figure R35b has been added as Figure S4 in the revised supplementary information.

Question 2: For Figure 4, the caption “potential energy” is not terminologically correct. Based on the text, this should be more accurately depicted as adsorption energy.

Our response: We are grateful for the reviewer’s insightful comment. In order to provide more insights into the reaction mechanism, we have reconstructed the WO₃{001} and WO₃{110} models and the detailed theoretical calculation has been carried out. Following the reviewer’s kind suggestion, the caption “Potential energy” is changed to “Energy”. The corresponding model changes and computational analyses are as follows. In the previous theoretical calculation model, the purpose of adding O dangling bonds is to achieve a ratio of W atoms to O atoms as 1:3 in WO₃ model. However, the O dangling bonds will make the surface unstable. If the O dangling bonds are removed, the exposed sites are changed to the W atoms, which are also unstable due to the unsaturated coordination. According to the high-resolution O1s XPS spectra of WO₃{001} (Figure R36a) and WO₃{110} (Figure R36b), the surface OH species are observed, which come from the protonated O dangling bonds. Thus, we have modified our models by introducing surface OH species.

Figure R36. High-resolution O1s XPS spectra of (a) $\text{WO}_3\{001\}$ and (b) $\text{WO}_3\{110\}$.

Figure R37. (a) Energy diagrams of CH_4 and H_2O adsorption as well as CH_4 activation on the surface of $\text{WO}_3\{001\}$ and (b) $\text{WO}_3\{110\}$. (c) Atomic configurations for the corresponding steps in the simulation of $\text{WO}_3\{001\}$ and (d) $\text{WO}_3\{110\}$.

Figure R38. (a) Atomic configurations of H₂O adsorption on WO₃{001} and (b) WO₃{110} surface.

Figure R39. (a) EPR spectra of WO₃{001} and (b) WO₃{110} in the pure CH₄ or CH₄ + O₂ atmosphere at room temperature with DMPO as the radical trapping agent in aqueous solution.

Figure R40. Energy diagrams of H₂O oxidation on the surface of (a) WO₃{001} and (b) WO₃{110}. The energy value of V_o + 2OH (free) and *H + OH (free) on WO₃{001} is corrected with $U_e = 2.78$ eV. The energy values of V_o + 2OH (free) on WO₃{110} is corrected with $U_e = 2.52$ eV.

Figure R41. Atomic configurations of (a) $V_o + 2OH$ (free) and (b) $*H + OH$ (free) at O_b site on the surface of $WO_3\{001\}$, and (c) $V_o + 2OH$ (free) at O_b site on the surface of $WO_3\{110\}$.

To explain the distinct CH_4 oxidation process on $WO_3\{001\}$ and $WO_3\{110\}$, DFT recalculations have been performed to examine their abilities responsible for CH_4 and H_2O adsorption as well as activation. Bridging O (O_b), terminal O (O_t) and W atoms are taken as the adsorption sites of CH_4 , respectively. It turns out that the O_b sites (O-IN 2) in both $WO_3\{001\}$ (Figure R37a and R37c) and $WO_3\{110\}$ (Figure R37b and R37d) exhibit the lower CH_4 adsorption energy than O_t (O-IN 1) and W (W-IN 1) sites. This result indicates that CH_4 molecules are preferentially adsorbed at the O_b sites. To be specific to $WO_3\{001\}$, through CH_4 activation (O-IN 4), a $*CH_3$ group is formed and firmly adsorbed on its O_b site to generate $*OCH_3$ group. Such species is confirmed by the rising $*OCH_3$ signals in *in situ* DRIFTS spectra of $WO_3\{001\}$ in CH_4 or $CH_4 + O_2$ atmosphere under light irradiation (Figure 3a and 4a in the revised manuscript, Figure S37a and S38a in the revised supplementary information). The large positive energy of the intermediates of O-IN 3 and CH_3 (free) means that the $*CH_3$ group is hardly desorbed from $WO_3\{001\}$ surface. This deduction is proved by the EPR test of $WO_3\{001\}$ in CH_4 or $CH_4 + O_2$ atmosphere, where no $*CH_3$ radical is observed (Figure R39a). Alternatively, the $*OH$ radical is derived from the adsorption of H_2O molecules at O_b site (O-IN 5, Figure R37a and R38a) with subsequent oxidation, which is not involved in CH_4 oxidation as the corresponding signals of *in situ* DRIFTS display no change before and after H_2O addition (Figure 3a and 4a in the revised manuscript, Figure S37a and S38a in the revised supplementary information). Thus, the CH_4 oxidation process of $WO_3\{001\}$ for HCHO generation goes through an active site mechanism rather than a radical mechanism. To explore whether the H_2O oxidation on $WO_3\{001\}$ surface affects the CH_4 oxidation process through the consumption of O_b

site, the $\cdot\text{OH}$ formation mechanism has been investigated. According to previous reports, the formation of $\cdot\text{OH}$ radical via H_2O oxidation can be divided into two ways: one is that both O_b and photohole participate in H_2O oxidation to generate one oxygen vacancy and two $\cdot\text{OH}$ radicals ($\text{O}_b + h^+ + \text{H}_2\text{O} \rightarrow \text{V}_o + 2\cdot\text{OH}$, *Nosaka, Y. et al. Chem. Rev. 117, 11302–11336 (2017)*; *Hou, J. et al. J. Am. Chem. Soc. 134, 9978–9985 (2012)*; *Salvador, P. et al. Prog. Surf. Sci. 86, 41–58 (2011)*); the other is simply photohole oxidizing H_2O to produce one H^+ cation and one $\cdot\text{OH}$ radical without O_b consumption ($h^+ + \text{H}_2\text{O} \rightarrow \text{H}^+ + \cdot\text{OH}$, *Nosaka, Y. et al. Chem. Rev. 117, 11302–11336 (2017)*; *Jonsson, M. et al. J. Phys. Chem. C 118, 10083–10087 (2014)*). As shown in Figure R40a and R41a, the positive energy value (0.36 eV) of $\text{V}_o + 2\text{OH}$ (free) on $\text{WO}_3\{001\}$ reveals that the $\cdot\text{OH}$ radical formation is not through the process of $\text{O}_b + h^+ + \text{H}_2\text{O} \rightarrow \text{V}_o + 2\cdot\text{OH}$. While the negative energy value (-0.91 eV, Figure R40a and R41b) of $h^+ + \text{OH}$ (free) confirms that the $\cdot\text{OH}$ radical is generated through $h^+ + \text{H}_2\text{O} \rightarrow \text{H}^+ + \cdot\text{OH}$ process without O_b consumption. Therefore, the H_2O oxidation on $\text{WO}_3\{001\}$ surface does not change the CH_4 oxidation process as no O_b is consumed.

As for $\text{WO}_3\{110\}$, Figure R37b and R37d uncover that after CH_4 activation at O_b site ($\text{O-IN 3} + \text{CH}_3(\text{free})$), the adsorbed $h^+ + \text{CH}_3$ group is hardly formed and easily desorbed to form $\text{CH}_3(\text{free})$ leaving $\text{O}_b\text{-H}$ group. Thus, the active site mechanism of CH_4 oxidation does not occur on $\text{WO}_3\{110\}$. Due to the high surface energy of $\text{WO}_3\{110\}$ (4.13 J m^{-2}), the $\text{O}_b\text{-H}$ is also easily desorbed to form oxygen vacancy, which is similar to the H_2O oxidation to form oxygen vacancy and $\cdot\text{OH}$ radical discussed later. This is consistent with the *in situ* DRIFTS results in CH_4 atmosphere without H_2O addition, where no CH_4 oxidation is discerned only with the detected loss of W-O signal (Figure 3b in the revised manuscript). Actually, in an aqueous reaction environment, due to the large adsorption energy difference between H_2O and CH_4 molecules on the surface of $\text{WO}_3\{110\}$ (O-IN 5 , Figure R37b and R38b), O_b preferentially adsorbs H_2O molecules to block the adsorption of CH_4 . And as displayed in Figure R40b and R41c, the adsorbed H_2O molecule is easily oxidized to $\cdot\text{OH}$ radical through $\text{O}_b + h^+ + \text{H}_2\text{O} \rightarrow \text{V}_o + 2\cdot\text{OH}$ with energy of -1.24 eV ($\text{V}_o + 2\text{OH}$ (free)). During this process, O_b is consumed along with V_o generation, further excluding the possibility of CH_4 oxidation at O_b active site. This is proved by the signal of W-O consumption in absence of CH_4 activation in *in situ* DRIFTS spectra of $\text{WO}_3\{110\}$ in CH_4 atmospheres with H_2O addition (Figure S37b in revised supporting information). Thereby, only $\cdot\text{OH}$

radical without $\cdot\text{CH}_3$ radical is observed in pure CH_4 atmosphere for $\text{WO}_3\{110\}$ aqueous system (Figure R39b). In short, we confirm that the CH_4 oxidation process over $\text{WO}_3\{110\}$ for HCHO generation is not implemented through an active site mechanism. Note that the signals of $\cdot\text{CH}_3$ radicals appear in $\text{WO}_3\{110\}$ system with $\text{CH}_4 + \text{O}_2$ atmosphere, which results from oxidation of free CH_4 by substantial O_2^- ions. The large adsorption energy difference of H_2O molecule between $\text{WO}_3\{001\}$ and $\text{WO}_3\{110\}$ stems from the number of hydrogen bonds formed. On $\text{WO}_3\{001\}$ surface, H_2O molecule is adsorbed through one hydrogen bond, while on $\text{WO}_3\{110\}$ surface, two hydrogen bonds are formed after H_2O adsorption. Therefore, $\text{WO}_3\{110\}$ has a higher adsorption capacity for H_2O molecules than $\text{WO}_3\{001\}$.

Computational method. The Vienna Ab Initio Simulation Package was used to carry out all the density functional theory calculations within the generalized gradient approximation using the Perdew-Burke-Ernzerh of formulation. The projected augmented wave potentials were selected to describe the ionic cores. Valence electrons were taken into account by using a plane-wave basis set with a kinetic energy cutoff of 400 eV. The electronic energy was considered self-consistent when the energy change was smaller than 10^{-5} eV. Partial occupancies of the Kohn-Sham orbitals were allowed using the Gaussian smearing method and a width of 0.02 eV. Geometry optimization was considered convergent when the energy change was smaller than 0.05 eV/Å. The Brillouin zone was sampled with $2 \times 2 \times 1$ Monkhorst mesh. A 15 Å vacuum space along the z direction was added to avoid the interaction between the two neighboring images. For H_2O oxidation process on both $\text{WO}_3\{001\}$ and $\text{WO}_3\{110\}$, since one photoelectron was left after H_2O oxidation by photohole, the formation energy of $\text{V}_o + 2\text{OH}$ (free) and $\cdot\text{H} + \text{OH}$ (free) modes took account of one photoelectron energy (Watson, G. et al. Solid State Ionics 177, 3069–3074 (2006); Nolan, M. et al. Chem. Phys. Lett. 499, 126–130 (2010); Xiong, Y. et al. Nat. Commun. 13, 2806 (2022)). The energy of a photogenerated electron was equal to the band gap energy, which excited electron from valence band to conduction band. Thus, the energy values of $\text{V}_o + 2\text{OH}$ (free) and $\cdot\text{H} + \text{OH}$ (free) on $\text{WO}_3\{001\}$ and $\text{WO}_3\{110\}$ were corrected with $U_e = 2.78$ eV and $U_e = 2.52$ eV, respectively.

Our revision: The analyses of DFT calculation have been added in the revised manuscript (please see the red marked parts on pages 11-13) and the revised supplementary information (please see the red marked parts on page S4). Figure R36a,

R36b, R37 have been added as Figure 1f, 1g, 5 in the revised manuscript. Figure R38, R39, R40 and R41 have been added as Figure S41, S40, S42 and S43 in the revised supplementary information.

Question 3: Quantum efficiency is in the range $\sim 10^{-4}\%$ which are relatively lower than reported literature values (J. Am. Chem. Soc. 2019, 141, 20507-20515; Nat. Commun. 2021, 12, 1-20) or past works by the authors (Nat. Sustain. 2021, 4, 509-515; Appl. Catal. B 2021, 283, 119661), even using a similar WO_3 material.

Our response: We are grateful for the reviewer's careful reading and kind reminding. After data recalculation and verification, we have found that the order of magnitude of the data in our previous manuscript was wrong. The recalculated Q.E. results (Table R1) are 0.33% (365 nm), 0.02% (420 nm), 0% (470 nm), 0% (535 nm), 0% (630 nm) for $\text{WO}_3\{001\}$ and 0.25% (365 nm), 0.04% (420 nm), 0% (470 nm), 0% (535 nm), 0% (630 nm) for $\text{WO}_3\{110\}$. We also notice that these Q.E. values are still low.

Table R1. The recalculated Q.E. values of $\text{WO}_3\{001\}$ and $\text{WO}_3\{110\}$ irradiated by different monochromatic light of 365, 420, 470, 535 and 630 nm, respectively.

Catalyst	Wavelength (nm)	Light intensity (mW cm ⁻²)	E _λ (J)	Number of products (μmol)		Q.E. (%)
				HCHO	CO ₂	
$\text{WO}_3\{001\}$	365	6.26	5.4×10^{-19}	2.16	0	0.33
	420	22.50	4.7×10^{-19}	0.61	0	0.02
	470	24.56	4.2×10^{-19}	0	0	0
	535	13.81	3.7×10^{-19}	0	0	0
	630	13.82	3.2×10^{-19}	0	0	0
$\text{WO}_3\{110\}$	365	6.26	5.4×10^{-19}	1.65	0	0.25
	420	22.50	4.7×10^{-19}	0.94	0	0.04
	470	24.56	4.2×10^{-19}	0	0	0
	535	13.81	3.7×10^{-19}	0	0	0
	630	13.82	3.2×10^{-19}	0	0	0

Fortunately, the Q.E. value of photocatalytic CH_4 oxidation is defined as the ratio of the number of electrons participating in CH_4 oxidation to the incident photon of a given wavelength. Through increasing the mass of catalyst, the gas pressure and the

amount of solvent aqueous solution, we anticipate to improve the Q.E. values.

For WO₃{001} system, 200 mg of photocatalyst was added into 150 mL of H₂O, meanwhile 11 bar O₂ + 20 bar CH₄ were injected into high pressure reactor. Under monochromatic light of 365, 420, 470, 535 and 630 nm irradiation for 3 h at 25°C, the productivities of HCHO (Figure R42a, R43a and Table R2) and CO₂ (Figure R42b-R42f, R43b-R43f and Table R2) were tested. Then, Q.E. values are calculated according to the following equation.:

$$\text{Q.E.} = \frac{R(\text{electron}) N_A}{IS/tE\lambda} \times 100\%$$

where I , S , t and N_A stand for the light intensity irradiated on the sample, irradiation area (12.56 cm²), reaction time (3 h) and Avogadro's constant (6.02 × 10²³ mol⁻¹), respectively. The value of $E\lambda$ is calculated through the formula of $E\lambda = hc / \lambda$. Taking $\lambda = 365$ nm as an example, the value of $E\lambda$ is calculated to be 5.4 × 10⁻¹⁹ J. $R(\text{electron})$ represents the number of electrons involved in the product formation.

WO₃{001}/ WO₃{110}

As for WO₃{001}, the production of HCHO and CO₂ from CH₄ and O₂ involves 4 and 8 electrons. Taking 365 nm light irradiation as an example ($I = 6.26$ mW cm⁻²), the yields of HCHO and CO₂ are 38.70 and 2.00 μmol (Table R2). Then, Q.E. = 6.02 × 10²³ mol⁻¹ × (38.70 μmol × 4 + 2.00 μmol × 8) × 100% / (6.26 mW/cm² × 12.56 cm² × 3 × 3600 s / 5.4 × 10⁻¹⁹ J) = 6.54%. Likewise, the obtained Q.E. values at the wavelength of 420, 470, 535 and 630 nm are 0.53%, 0%, 0% and 0% (Figure R44a and Table R2), respectively.

Likewise, the Q.E. values of WO₃{110} at the wavelength of 365, 420, 470, 535 and 630 nm are 4.96%, 0.41%, 0% and 0% (Figure S34b and Table S1), which were also obtained with the similar reaction conditions except 14 bar O₂ + 17 bar CH₄.

Figure R42. (a) UV-visible absorption spectra of HCHO product on $\text{WO}_3\{001\}$ with different wavelength of monochromatic light irradiation. (b-f) GC spectra of gas product from CH_4 oxidation on $\text{WO}_3\{001\}$ with different wavelength of monochromatic light irradiation. Peaks at 9.21 and 19.11 min are attributed to residual CH_4 and produced CO_2 , respectively. Reaction condition: 200 mg catalyst, 11 bar O_2 , 20 bar CH_4 , 3 h reaction time, reaction temperature 25°C.

Figure R43. (a) UV-visible absorption spectra of HCHO product on WO₃{110} with different wavelength of monochromatic light irradiation. (b-f) GC spectra of gas product from CH₄ oxidation on WO₃{110} with different wavelength of monochromatic light irradiation. Peaks at 9.21 and 19.11 min are attributed to residual CH₄ and produced CO₂, respectively. Reaction condition: 200 mg catalyst, 14 bar O₂, 17 bar CH₄, 3 h reaction time, reaction temperature 25°C.

Figure R44. Q.E. values with different wavelength of monochromatic light irradiation at 25°C reaction temperature, 365 nm 6.26 mW cm⁻², 420 nm 22.50 mW cm⁻², 470 nm 24.56 mW cm⁻², 535 nm 13.81 mW cm⁻² and 630 nm 13.82 mW cm⁻², along with the diffuse reflectance spectra of (a) WO₃{001} and (b) WO₃{110}.

Table R2. Q.E. values of WO₃{001} and WO₃{110} irradiated by different monochromatic light of 365, 420, 470, 535 and 630 nm, respectively.

Catalyst	Wavelength (nm)	Light intensity (mW cm ⁻²)	E _λ (J)	Number of products (μmol)		Q.E. (%)
				HCHO	CO ₂	
WO ₃ {001}	365	6.26	5.4 × 10 ⁻¹⁹	38.70	2.00	6.54
	420	22.50	4.7 × 10 ⁻¹⁹	13.47	0.36	0.53
	470	24.56	4.2 × 10 ⁻¹⁹	0	0	0
	535	13.81	3.7 × 10 ⁻¹⁹	0	0	0
	630	13.82	3.2 × 10 ⁻¹⁹	0	0	0
WO ₃ {110}	365	6.26	5.4 × 10 ⁻¹⁹	32.16	0.11	4.96
	420	22.50	4.7 × 10 ⁻¹⁹	11.084	0	0.41
	470	24.56	4.2 × 10 ⁻¹⁹	0	0	0
	535	13.81	3.7 × 10 ⁻¹⁹	0	0	0
	630	13.82	3.2 × 10 ⁻¹⁹	0	0	0

Our revision: The specific calculation of Q.E. has been added in the revised manuscript (please see page 8) and revised supplementary information (please see page S39-S40). Figure R42, R43, R44 and Table R2 have been added as Figure S35, S36, S34 and Table S1 in the revised supplementary information.

Question 4: For Figure 3a-b and Figure 5a-b, DRIFT studies are rarely interpreted on the basis of negative axes/baseline.

Our response: We are grateful for the reviewer's careful reading and insightful comment. We would like to politely point out that many *in situ* DRIFT studies of lattice-O have been reported on the basis of negative axes/baseline. As shown in Figure R45a (*Jangjou, J. et al. ACS Catal. 8, 1325–1337 (2018)*), for selective catalytic reduction of NO_x with NH₃ using Cu-SSZ-13 as catalysts, the curve at 0 min (Figure R45a) in absence of obvious signals was taken as the baseline. Along with reaction time, negative IR features at 900 and 950 cm⁻¹ appeared and gradually increased, suggesting solvation of Z₂Cu and ZCuOH on Cu-SSZ-13 by breaking the Cu-O-Cu bond. Similar result was also observed on Cu-SSZ-39 in selective catalytic reduction of NO_x with NH₃ (Figure R45b, *Du, J. et al. Catal. Sci. Technol. 10, 1256–1263 (2020)*). Taking the black curve (Fresh) as baseline, the negative peaks at 900 and 940 cm⁻¹ originated from the breakage of Cu-O bond. In photocatalytic H₂O oxidation on NaTaO₃-based photocatalysts (Figure R45c, *Ding, Q. et al. J. Mater. Chem. A 8, 6812–6821 (2020)*, *Fang, Z. et al. J. Semicond. 4, 43 (2022)*), the intensity of broad band region less than 1060 cm⁻¹ generally decreased with the UV irradiation time. This indicated that lattice-O was consumed and participated in H₂O oxidation to produce O₂ using Ta(VI)=O and Ta(O₂) peroxy species as intermediates. To be specific for CH₄ oxidation reaction, we notice that though lattice-O consumption has not been clearly assigned in the *in situ* DRIFT spectra, the signal of CH₄ consumption on the basis of negative axes/baseline has been frequently observed (*Sun, X. et al. Nat. Commun. 13, 6677 (2022)*, *Luo, L. et al. Nat. Commun. 12, 1218 (2021)*, *Sun, S. et al. Catal. Sci. Technol. 12, 3727-3736 (2022)*, *He, J. et al. J. Am. Chem. Soc. 142, 17119-17130 (2020)*).

Figure R45. (a) *In situ* DRIFTS spectra of selective catalytic reduction of NO_x with NH₃ using Cu-SSZ-13 or (b) Cu-SSZ-39 as catalysts. (c) *In situ* DRIFTS spectra of photocatalytic H₂O oxidation on NaTaO₃-based photocatalyst.

Question 4.1: Particularly, findings pertaining to the generation of oxygen vacancies raises few questions and require clarifications, such as

- The mechanism behind the generation of oxygen vacancies is not explicitly discussed. Is it the same for $\text{WO}_3\{001\}$ and $\text{WO}_3\{110\}$?

Our response: We are grateful for the reviewer's insightful question. In regard of oxygen vacancies of $\text{WO}_3\{001\}$ and $\text{WO}_3\{110\}$, their formation modes and played roles are different in photocatalytic CH_4 oxidation reactions.

Figure R46. Proposed reaction mechanisms for photocatalytic oxidation of CH_4 on (a) $\text{WO}_3\{001\}$ and (b) $\text{WO}_3\{110\}$.

Figure R47. *In situ* DRIFTS spectra of $\text{WO}_3\{001\}$ in (a) CH_4 or (b) $\text{CH}_4 + \text{O}_2$ atmosphere under different light irradiation time. Here, * denotes an adsorption site on surface. The insets show the magnified $\text{W}-\text{O}$ peak.

Figure R48. (a) High-resolution O1s XPS spectra of $\text{WO}_3\{001\}$ before and (b) after reaction in CH_4 atmosphere or (c) $\text{CH}_4 + \text{O}_2$ atmosphere.

During the CH_4 oxidation process in $\text{WO}_3\{001\}$ system, the formation of oxygen vacancy is caused by the consumption of lattice-O in the formation of HCHO (Path 1, Figure R46a). The role of oxygen vacancy is to adsorb O_2 molecules and reduce them by photoelectrons to repair the consumed lattice-O (Path 2, Figure R46a), which can ensure the sustainable production of HCHO. The elaborate explanation is as follows. As shown by the reaction mechanism of $\text{WO}_3\{001\}$ (Figure R46a) and DFT calculations (Figure R37a), CH_4 molecules are preferentially adsorbed at the bridging O sites (lattice- O_b). Upon photoirradiation, the lattice- O_b is activated through transferring one electron to the adjacent W atom. Afterwards, the activated lattice- O_b of $\text{WO}_3\{001\}$ is able to insert into the C-H bond of CH_4 molecule sequentially forming the $^*\text{OCH}_3$ and $^*\text{CH}_2$ species, as shown in *in situ* DRIFTS spectra of $\text{WO}_3\{001\}$ in CH_4 atmosphere (Figure R47a). Meanwhile the left H atoms from CH_4 is abstracted by adjacent -OH on W site through hydrogen atom transfer (HAT) process, as shown in DFT results (O-IN4 in Figure R37c). The $^*\text{OCH}_2$ species are desorbed to form HCHO molecules, where the consumed lattice- O_b becomes an oxygen vacancy. Note that unlike the $\text{WO}_3\{110\}$ system, the H_2O oxidation in $\text{WO}_3\{001\}$ system to generate $^*\text{OH}$ radical is through the process of $\text{h}^+ + \text{H}_2\text{O} \rightarrow \text{H}^+ + ^*\text{OH}$ without lattice-O consumption

(Figure R40a). Thus, no oxygen vacancy is formed during H_2O oxidation in $\text{WO}_3\{001\}$ system. The role of oxygen vacancy is to adsorb O_2 molecules as shown by the XPS results (Figure R48a and R48b). The adsorbed O_2 molecules by oxygen vacancy are then reduced by photoelectrons to fix the depleted lattice- O_b , which is confirmed by the repaired lattice-O peak in XPS spectra after CH_4 oxidation in $\text{CH}_4 + \text{O}_2$ atmosphere (Figure R48c) and positive W-O peak in *in situ* DRIFTS spectra of $\text{WO}_3\{001\}$ in $\text{CH}_4 + \text{O}_2$ atmosphere (Figure R47b inset). The repaired lattice- O_b sequentially participate in the CH_4 oxidation to HCHO formation.

Figure R49. (a) *In situ* DRIFTS spectra of $\text{WO}_3\{110\}$ in CH_4 atmosphere with H_2O addition under different light irradiation time. Here, * denotes an adsorption site on surface. (b) High-resolution O1s XPS spectra of $\text{WO}_3\{110\}$ before and (c) after reaction in CH_4 atmosphere.

Figure R50. EPR spectra of $\text{WO}_3\{110\}$ in O_2 atmosphere at 77 K temperature.

As for $\text{WO}_3\{110\}$, the formation of oxygen vacancy is mainly through the H_2O oxidation by surface lattice-O with high surface energy (Path 1, Figure R46b). As shown by the DFT calculation results (Figure R37b), due to the large adsorption energy difference between H_2O and CH_4 molecules at O_b site of $\text{WO}_3\{110\}$, H_2O molecules are preferentially adsorbed inhibiting the adsorption of CH_4 molecules. And the adsorbed H_2O molecule is easily oxidized to $\cdot\text{OH}$ radical through $\text{O}_b + \text{h}^+ + \text{H}_2\text{O} \rightarrow \text{V}_o + 2\cdot\text{OH}$ with energy of -1.24 eV ($\text{V}_o + 2\text{OH}$ (free), Figure R40b and R41c) according to the DFT result. During this process, O_b is consumed with V_o generation, which is

verified by the negative W-O peak in *in situ* DRIFTS spectra (Figure R49a) and the 35.1% consumption of lattice-O of $\text{WO}_3\{110\}$ after CH_4 oxidation (Figure R49b and R49c) in H_2O solution. The role of oxygen vacancy is to adsorb O_2 molecule (Figure R49c), which can engage into the repairment of detached lattice-O or be reduced to O_2^- group by photoelectron (Path 2, Figure R46b). However, compared with the four-electron oxygen reduction to repair lattice-O, the one-electron oxygen reduction for O_2^- generation pathway has favorable kinetic properties. Therefore, for $\text{WO}_3\{110\}$, the consumed lattice-O of $\text{WO}_3\{110\}$ can only be partially repaired, and the adsorbed O_2 is mainly involved in the formation of O_2^- anion (Figure R50). The O_2^- anion activates CH_4 molecule to produce $\cdot\text{CH}_3$ and $\cdot\text{OOH}$ radicals. Through combination between $\cdot\text{CH}_3$ and $\cdot\text{OOH}$ radicals, CH_3OOH is generated followed by decomposition to CH_3OH and HCHO .

Our revision: The elucidation of generation and role of oxygen vacancies has been added in the revised manuscript (please see the red marked parts on pages 15-17). Figure R46, R47a, R47b, R48a, R48b, R49b, R49c, R50 have been added as Figure 7, 3a, 4a, 1f, 3c, 1g, 3d, 6a in the revised manuscript. Figure R48c and R49a have been added as Figure S39a and S37b in the revised supplementary information.

Question 4.2: Since, authors claim that oxygen source originates from lattice-O from $\text{WO}_3\{001\}$, shouldn't this directly connote that there is a higher lattice loss in $\text{WO}_3\{001\}$ than in $\text{WO}_3\{110\}$ after the reaction under pure CH_4 atmosphere? But this is not the case as revealed from the DRIFT/XPS results.

Our response: We are grateful for the reviewer's insightful question. As shown by the DRIFT/XPS results, a higher lattice-O loss is found on $\text{WO}_3\{110\}$ than $\text{WO}_3\{001\}$. This is because in aqueous solutions, H_2O molecules are easily adsorbed on lattice-O of $\text{WO}_3\{110\}$ and oxidized to produce $\cdot\text{OH}$ radicals ($\text{O}_b + h^+ + \text{H}_2\text{O} \rightarrow \text{V}_o + 2\cdot\text{OH}$) with lattice-O participation, which is verified by the DFT calculations (Figure R40b and R41c). During this process, the lattice-O of $\text{WO}_3\{110\}$ is largely consumed and oxygen vacancy is formed. Therefore, the lattice-O consumption of $\text{WO}_3\{110\}$ should be equal to half of the $\cdot\text{OH}$ radical production without O_2 addition (consumed lattice-O of $\text{WO}_3\{110\} = 0.5 \times C(\cdot\text{OH})$). In $\text{WO}_3\{001\}$ system, since the lattice-O participates into HCHO formation, the amount of consumed lattice-O is equal to that of HCHO produced without O_2 addition (consumed lattice-O of $\text{WO}_3\{001\} = C(\text{HCHO})$). Based

on the XPS results, after reactions in CH₄ atmosphere, the amount of lattice-O consumed on WO₃{110} should be 1.5 times than that of WO₃{001}, that is $0.5 \times C(\cdot\text{OH}) = 1.5 \times C(\text{HCHO}) \rightarrow C(\cdot\text{OH}) = 3 \times C(\text{HCHO})$. However, through coumarin fluorescence test, the $\cdot\text{OH}$ radical generated over WO₃{110} after reaction in CH₄ atmosphere (or Ar atmosphere) is quantified to be only $\sim 0.03 \mu\text{mol L}^{-1}$ for 3 h reaction time in 5 mL H₂O (Figure R51a and R51b), which is far less than the generated HCHO ($17.8 \mu\text{mol L}^{-1}$, that is $0.567 \mu\text{mol m}^{-2}$, Figure S11 in revised supplementary information) over WO₃{001} under the same reaction condition. Moreover, no H₂O₂ is generated (Figure R51c). Thus, we hypothesize that the $\cdot\text{OH}$ must be quenched soon after its formation. Seen from the DFT calculation on WO₃{110}, the terminal OH group or the H atom is always spontaneously dissociated from the surface and likely combined to form H₂O molecule due to the high surface energy (Figure R52a). And the number of dissociated terminal H atoms is greater than that of dissociated terminal OH groups, which is revealed by the acidic nature of solution (pH = 6.06) after reaction in 20 bar CH₄ atmosphere with H₂O as solvent (Figure R52b). Given this, we deduce that the $\cdot\text{OH}$ radical generated by lattice-O is involved in this process and then quenched. As a result, the detected number of $\cdot\text{OH}$ radical is far less than the consumed amount of lattice-O over WO₃{110}. To verify that lattice-O consumption is indeed caused by H₂O oxidation, we use D₂O aqueous solution containing coumarin to carry out free radical trapping experiments. After light irradiation for 3 h in 20 bar CH₄, weak fluorescent signal is observed, indicating that trace amount of $\cdot\text{OD}$ is generated ($4.5 \times 10^{-3} \mu\text{mol L}^{-1}$, Figure R53a). This value is much lower than the generated $\cdot\text{OH}$ radical amount ($0.03 \mu\text{mol L}^{-1}$) with H₂O as solvent, which is attributed to the low kinetic energy of deuterium isotope effect. The pH value of the reaction solution is tested to be 5.79 (Figure R52c), which is lower than that of H₂O solution (pH = 6.06). The lower pH value is reasonable considering that the consumption of H⁺ is less with the smaller amount of $\cdot\text{OD}$ radical produced in D₂O. The O 1s XPS spectra of the recycled WO₃{110} further show that only 6.22% of the lattice-O is consumed using D₂O as solvent (Figure R53b and R53c). Altogether, we suppose that the lattice-O of WO₃{110} is indeed participated and consumed in H₂O oxidation to form $\cdot\text{OH}$ radical, which is easily quenched by combining with the dissociated H atom from the surface OH group of WO₃{110} instead of involvement of C₁ oxygenates formation. Moreover, due to the higher surface energy of WO₃{110} facets (4.13 J m^{-2}) than WO₃{001} facets (2.46 J m^{-2})

m⁻²), the detachment of lattice-O on WO₃{110} is accelerated (Wu, R. *et al. Appl. Surf. Sci.* 481, 1154–1159 (2019); Ma, J. *et al. Adv. Funct. Mater.* 28, 1705268 (2018); Wang, X. *et al. Comp. Mater. Sci.* 68, 218–221 (2013)). Thus, the WO₃{110} is more likely to lose lattice-O as revealed from *in situ* DRIFT and XPS results.

Figure R51. (a) Fluorescence spectra of 1 mM coumarin solution after reaction over WO₃{110}. (b) Calculated concentration of •OH radicals. (c) H₂O₂ monitoring by real-time UV-vis absorption spectra through color-developing method. Reaction condition: 10 mg WO₃{110}, 20 bar CH₄ or 20 bar Ar, 5 mL H₂O, 50°C reaction temperature, 3 h reaction time, Xenon light 150 mW cm⁻².

Figure R52. (a) Spontaneous dissociation or combination of terminal OH group or terminal H atom on WO₃{110} surface. (b) PH value of solution after reaction in H₂O or (c) D₂O solution with 20 bar CH₄ addition for 3 h at 50°C.

Figure R53. (a) Fluorescence spectrum of 1 mM coumarin solution taking WO₃{110} as photocatalyst with 20 bar CH₄ and 5 mL D₂O as solvent. (b) High-resolution O1s XPS spectra of WO₃{110} after reaction in CH₄ atmosphere with 5 mL D₂O as solvent.

(c) Pristine high-resolution O1s XPS spectra of WO₃{110}. Reaction condition: 10 mg WO₃{110}, 20 bar CH₄, 5 mL D₂O, 50°C reaction temperature, 3 h reaction time, Xenon light 150 mW cm⁻².

Inspired by the reviewer's insightful comment, we have more thoughts and answers about oxygen source when using WO₃{110}. As shown in Figure R39b, after O₂ is added to CH₄ atmosphere, both •OH radicals and •CH₃ radicals are detected. Do the •OH radicals all come from the H₂O decomposition? Is H₂O involved in CH₃OH formation through the combination between •CH₃ radical and •OH radical?

To answer the raised questions, the •OH radical is quantified with O₂ addition. Both in 9 bar O₂ + 11 bar Ar and 9 bar O₂ + 11 bar CH₄ atmosphere under light irradiation for 3 h at 50°C, the generated •OH radical amount is ~0.33 $\mu\text{mol L}^{-1}$ over WO₃{110} (Figure R54a and R54b). According to previous reports (Nosaka, Y. *et al. Chem. Rev.* **117**, 11302–11336 (2017); Jiang, Y. *et al. J. Am. Chem. Soc.* **145**, 2698–2707 (2023)), the •OH radical is generated through two electron reduction process of O₂ using H₂O₂ as intermediate ($\text{O}_2 + 2\text{H}^+ + 2\text{e}^- = \text{H}_2\text{O}_2 = 2\cdot\text{OH}$, $E(\text{O}_2, 2\text{H}^+/\text{H}_2\text{O}_2) = 0.695 \text{ V}_{\text{vs NHE}}$). Yet no signal of H₂O₂ is detected, indicating that all the H₂O₂ is decomposed (Figure R54c). Compared to the •OH radical (0.03 $\mu\text{mol L}^{-1}$) in absence of O₂, the concentration of •OH radical upon O₂ addition (0.33 $\mu\text{mol L}^{-1}$) is increased 11-fold. This result manifests that the •OH radical upon O₂ addition is mainly from O₂ reduction rather H₂O decomposition. Although the •OH radical is increased after O₂ addition, compared to the generated CH₃OH concentration (12.59 $\mu\text{mol L}^{-1}$, Figure 2e in the revised manuscript), the •OH radical (0.33 $\mu\text{mol L}^{-1}$) is still very low, suggesting that CH₃OH mainly originates from CH₃OOH decomposition rather than the combination between

- CH₃ radical and •OH radical. Besides, taking coumarin as the trapping agent for •OH radical, the type and yield of oxygenates are not changed, indicating that •OH radical is not involved in CH₄ oxidation process (Figure R55). Isotopically labeled D₂O and H₂¹⁸O as solvents are added to track whether H₂O is involved in oxygenates formation and the role of oxygen vacancy. Taking WO₃{110} as photocatalyst in 5 mL D₂O and 11 bar CH₄ + 9 bar O₂ atmosphere (Figure R56 and R57a, R57c, R57e), the yield of oxygenates is reduced compared with that using H₂O as solvent (Figure 2e in revised manuscript). This is because using WO₃{110} as catalyst to oxidize CH₄ with O₂, first lattice-O is needed to decompose H₂O to produce oxygen vacancy, and then oxygen vacancy adsorbs O₂ to participate in CH₄ oxidation. Using D₂O as a solvent, the

decomposition process of D₂O by lattice-O is difficult, and thus the number of oxygen vacancies and adsorbed O₂ is reduced (Figure R53b and R49c). Notably, taking the recycled WO₃{110} after reaction in 20 bar CH₄ and 5 mL H₂O as photocatalyst and being re-fed into 5 mL D₂O and 11 bar CH₄ + 9 bar O₂ atmosphere, the yield of oxygenates is found to be similar to that using H₂O as solvent (Figure R56 and R57b, R57d, R57f). This is because the recycled WO₃{110} is already equipped with sufficient oxygen vacancy, ensuring the O₂ adsorption and CH₄ oxidation (Figure R49c). After reaction, the HCHO signal generated in D₂O solvent demonstrates that the D atom of D₂O cannot participate in CH₄ oxidation into HCHO (Figure R58a and R59a), and the H atom of HCHO does not undergo H-D exchange with D₂O (Figure R58a and R59b). Despite CH₃OD signal is detected (Figure R58b and R60a), it may be caused by H-D exchange between CH₃OH and D₂O (Figure R58b and R60b).

Using ¹⁸O₂ + H₂O or O₂ + H₂¹⁸O as reactants and solvents, we find that only CH₃¹⁸OH (Figure R61 and R62a) and HCH¹⁸O (Figure R66b and R68a) or CH₃¹⁶OH (Figure R61 and R62b) and HCH¹⁶O (Figure R66b and R68b) are generated, further validating that H₂O cannot participate into CH₃OH and HCHO formation through [•]OH radical.

Figure R54. (a) Fluorescence spectra of 1 mM coumarin solution taking WO₃{110} as photocatalyst in 11 bar CH₄ + 9 bar O₂ or 11 bar Ar + 9 bar O₂ atmosphere. (b) Calculated concentration of [•]OH radicals. (c) H₂O₂ monitoring by real-time UV-vis absorption spectra through color-developing method. Reaction condition: 10 mg WO₃{110}, 11 bar CH₄ + 9 bar O₂ or 11 bar Ar + 9 bar O₂, 5 mL H₂O, 50°C reaction temperature, 3 h reaction time, Xenon light 150 mW cm⁻².

Figure R55. Photocatalytic CH_4 oxidation performance over $\text{WO}_3\{110\}$ using coumarin as $\cdot\text{OH}$ radical trapping agent. (a) UV-visible absorption spectrum of HCHO product. (b) ^1H NMR spectrum of oxygenates. (c) GC spectrum of gas product. (d) Yield of oxygenates from CH_4 oxidation. Reaction condition: 10 mg $\text{WO}_3\{110\}$, 5 mL 1mM coumarin solution, 11 bar CH_4 + 9 bar O_2 , 3 h reaction time, Xenon light 150 mW cm^{-2} .

Figure R56. Photocatalytic CH_4 oxidation performance on $\text{WO}_3\{110\}$ and recycled $\text{WO}_3\{110\}$ with D_2O as solvent. Recycled $\text{WO}_3\{110\}$ is obtained after CH_4 oxidation without O_2 addition and with H_2O solvent. Reaction condition: 10 mg catalyst, 5 mL D_2O , 11 bar CH_4 + 9 bar O_2 = 20 bar, 3 h reaction time, Xenon light 150 mW cm^{-2} .

Figure R57. Photocatalytic CH_4 oxidation performance on $\text{WO}_3\{110\}$ and the recycled $\text{WO}_3\{110\}$ with D_2O as solvent. (a) UV-visible absorption spectra of HCHO product on $\text{WO}_3\{110\}$ and (b) the recycled $\text{WO}_3\{110\}$. (c) ^1H NMR spectra of oxygenates on $\text{WO}_3\{110\}$ and (d) the recycled $\text{WO}_3\{110\}$. (e) GC spectra of gas product on $\text{WO}_3\{110\}$ and (f) the recycled $\text{WO}_3\{110\}$. The recycled $\text{WO}_3\{110\}$ is obtained after CH_4 oxidation without O_2 addition and with H_2O as solvent. Reaction condition: 10 mg catalyst, 5 mL D_2O , 11 bar CH_4 + 9 bar O_2 = 20 bar, 3 h reaction time, Xenon light 150 mW cm^{-2} .

Figure R58. (a) GC-MS spectra of HCHO obtained from CH_4 oxidation using D_2O as solvent over $\text{WO}_3\{110\}$, or from 20 mM HCHO D_2O solution. The 20 mM HCHO D_2O solution is prepared by mixing 16 μL 37% HCHO solution with 10 mL D_2O under stirring for 3 h. (b) GC-MS spectra of CH_3OH obtained from CH_4 oxidation using D_2O as solvent over $\text{WO}_3\{110\}$, or from 20 mM CH_3OH D_2O solution. The 20 mM CH_3OH D_2O solution is prepared by mixing 11 μL 98% CH_3OH solution with 10 mL D_2O under stirring for 3 h.

Figure R59. (a) Screenshot on GC-MS spectra of HCHO obtained from CH_4 oxidation using D_2O as solvent over $\text{WO}_3\{110\}$, (b) or from 20 mM HCHO D_2O solution. The 20 mM HCHO D_2O solution is prepared by mixing 16 μL 37% HCHO solution with 10 mL D_2O under stirring for 3 h.

Figure R60. (a) Screenshot on GC-MS spectra of CH₃OH obtained from CH₄ oxidation using D₂O as solvent over WO₃{110}, (b) or from 20 mM CH₃OH D₂O solution. The 20 mM CH₃OH D₂O solution is prepared by mixing 11 μL 98% CH₃OH solution with 10 mL D₂O under stirring for 3 h.

Figure R61. (a) GC-MS spectra of CH₃OH obtained from CH₄ oxidation using ¹⁸O₂ or H₂¹⁸O over WO₃{110}. Reaction condition: 10 mg catalyst, 11 bar CH₄, 9 bar ¹⁸O₂ + 5 mL H₂O or 9 bar O₂ + 5 mL H₂¹⁸O, 3 h reaction time, Xenon light 150 mW cm⁻².

Figure R62. (a) Screenshot on GC-MS spectra of CH₃OH obtained from CH₄ oxidation using ¹⁸O₂ or (b) H₂¹⁸O over WO₃{110}. Reaction condition: 10 mg catalyst, 11 bar CH₄, 9 bar ¹⁸O₂ + 5 mL H₂O or 9 bar O₂ + 5 mL H₂¹⁸O, 3 h reaction time, Xenon light 150 mW cm⁻².

Our revision: The reason of more lattice-O loss on WO₃{110} than WO₃{001} has been added in the revised manuscript (please see the red marked parts on page 9 and 12-13) and the revised supplementary information (please see the red marked parts on page S54-S57). Figure R51-R62 have been added as Figure S45-S56 in the revised supplementary information.

Question 4.3: To further confirm whether WO₃{110} Ov fails to be replenished in a mixed CH₄ + O₂ atmosphere, quantitative XPS data should also be provided as per Table S3.

Our response: We are grateful for the reviewer's kind reminding and constructive comment. The quantitative O1s XPS data of both WO₃{001} and WO₃{110} before and after photocatalytic reaction in a mixed CH₄ + O₂ atmosphere has been provided as shown in Table R3. Before the reaction, the peak intensity of lattice-O in the O1s XPS spectra of WO₃{110} is 6.26×10^4 . As comparison, after the reaction in CH₄ + O₂ atmosphere, the lattice-O intensity becomes 5.39×10^4 . This indicates that even in the O₂ atmosphere, 13.9% of the lost lattice-O in the WO₃{110} system has not been repaired. On the contrary, nearly 100% of lost lattice-O in the WO₃{001} system has

been fully restored.

Table R3. XPS peak intensities of O1s before and after reactions in CH₄ + O₂ atmosphere on WO₃{001} and WO₃{110}.

Condition	Types of O1s	WO ₃ {001} (intensity × 10 ⁴)	WO ₃ {110} (intensity × 10 ⁴)
Before reaction	Lattice-O	5.39	6.26
	OH	1.41	2.04
	Adsorbed O ₂	0	0
	C=O	0	0
After reaction	Lattice-O	5.33	5.39
	OH	1.41	1.72
	Adsorbed O ₂	1.32	1.72
	C=O	1.33	1.59

Our revision: The quantitative XPS data explanation has been added in the revised manuscript (please see the red marked parts on pages 11) and the revised supplementary information (please see the red marked parts on page S80). Table R3 has been added as Table S4 in the revised supplementary information.

Question 5: Although the utilization routes for photogenerated electron-hole pairs are clearer for WO₃{110} which proceed through the radical process (e⁻ for superoxide generation and h⁺ for hydroxyl ions), it is unclear for WO₃{001} which proceed through active site routes. Where do the e⁻ and h⁺ pairs flow to for WO₃{001}?

Our response: We are grateful for the reviewer's insightful question. The utilization routes for photogenerated electron-hole pairs on WO₃{001} can be elucidated by the reaction mechanism (Figure R46a). It is known that the valence band maximum of WO₃ is mainly composed of O2p orbitals while the conduction band minimum is mainly constituted by the W5d orbitals (*Li, W. et al. ACS Nano 8, 11770–11777 (2014); Wei, S. et al. Appl. Catal. B Environ. 283, 119661 (2021)*). Under light irradiation, the photoelectrons from valence band of WO₃ are excited to the conduction band, that is,

from the O2p orbitals to the W5d orbitals. This makes the valence state of O_b change from O²⁻ to O⁻ and W atom from W⁶⁺ to W⁵⁺ (Besnardiere, J. et al. *Nat. Commun.* 10, 327 (2019); Mi, Q. et al. *J. Am. Chem. Soc.* 134, 18318–18324 (2012); Li, W. et al. *ACS Nano* 8, 11770–11777 (2014)). Therein, the O⁻ is the photohole (h⁺) and the W₅₊ is photoelectron (e⁻). After the CH₄ adsorption at the O_b site, the O⁻ (h⁺) of WO₃{001} is able to insert into the C-H bond of CH₄ molecule sequentially forming the *OCH₃ and *CH₂ species, as shown by the *in situ* DRIFTS spectra of WO₃{001} in CH₄ and CH₄ + O₂ atmospheres (Figure R47) and DFT calculation (O-IN 4, Figure R37a). Meanwhile the left H atom from CH₄ is abstracted by adjacent -OH on W site through hydrogen atom transfer (HAT) process, as verified by the DFT calculation (Figure R37a). The *OCH₂ species are desorbed to form HCHO molecules. During this process, the O⁻ (h⁺) has been consumed leaving e⁻ to remain as negatively charged W⁵⁺. The consumed O_b atom becomes an oxygen vacancy, which is conducive to O₂ adsorption on the surface of WO₃{001} as shown by the XPS result (Figure R48b, Table S3 in revised supplementary information). The adsorbed O₂ molecules are then reduced by photoelectrons (e⁻) from W⁵⁺ to fix the depleted O_b atoms, which is proved by the XPS result (Figure R48c, Table R3). During this process, the e⁻ is consumed. The above process is the utilization routes for photogenerated electron-hole pairs for WO₃{001}. **Our revision:** The explanation on the utilization routes for photogenerated electron-hole pairs of WO₃{001} has been added in the revised manuscript (please see the red marked parts on pages 15-16).

Question 6: Any reasons as to why ¹³C NMR was used for ¹³C isotope labeling experiments and not GC-MS like in the case with ¹⁸O₂?

Our response: We are grateful for the reviewer's careful reading and insightful question. Due to the small molecular weight and low-boiling point (-19.5°C), the peak velocity of HCHO is too fast to be accurately measured by GC-MS in most columns. Therefore, in the previous oxygen isotope test (¹⁸O₂), we extracted the O-atom of HCHO molecule into H₂O molecule through acetylacetone color-developing for GC-MS test (Figure R63). If the ¹³C isotope labeling experiment is adopted for GC-MS characterization, the 3,5-diacetyl-1,4-dihydrolutidine molecule will be the test target. However, owing to the large molecular weight and complex molecular structure, it is difficult to find a suitable column to separate it and as-obtained GC-MS spectrum is too

complicated to be analyzed.

Figure R63. Reaction mechanism on acetylacetone color-developing of HCHO.

Figure R64. (a) GC-MS spectra of HCHO obtained in $\text{WO}_3\{001\}$ and $\text{WO}_3\{110\}$ using $^{13}\text{CH}_4$ as carbon isotope.

Figure R65. Screenshot on GC-MS spectra of HCHO obtained in $\text{WO}_3\{001\}$ and $\text{WO}_3\{110\}$ using $^{13}\text{CH}_4$ as carbon isotope.

Fortunately, we find that the SH-Polar Wax column from SHIMADZU can separate HCHO and a satisfactory GC-MS signal can be achieved. Then, we have switched to directly measure HCHO by GC-MS spectroscopy. Test conditions: inlet temperature 180°C, splitless inlet, helium as carrier gas, linear speed of 25.5 cm s⁻¹, column temperature of 40°C with 120°C pretreatment, GC-MS ion source temperature of 200°C. The origin of C-atoms in HCHO can be traced by isotope labeling experiments with ¹³CH₄. In both WO₃{001} and WO₃{110} (Figure R64 and R65) systems, H¹³CHO peaks (m/z = 31) are found taking ¹³CH₄ as carbon source, revealing that the C-atom of HCHO is originated from CH₄.

Our revision: The GC-MS results have been added in the revised manuscript (please see the red marked parts on pages 17-18) and the revised supplementary information (please see the red marked parts on page S4). Figure R64 has been added as Figure 8a in the revised manuscript. Figure R65 has been added as Figure S63 in the revised supplementary information.

Question 7: There should be discussion on the role of H₂O in the reaction system. Is it possible that H₂O can act as an O source as well? This possibility was neglected in the entire work.

Our response: We are grateful for the reviewer's critical comment. The origin of O-atoms in HCHO can be traced by isotope labeling experiments with ¹⁸O₂ and H₂¹⁸O. In both WO₃{001} (Figure R66a, R67) and WO₃{110} (Figure R66b, R68) systems, HCH¹⁸O peaks (m/z = 32) are found taking ¹⁸O₂ as oxygen source while H₂¹⁸O makes no difference (Figure R66c, R69 and R66b, R68), indicating that the O-atom of HCHO is originated from O₂ rather than H₂O. After the CH₄ oxidation reaction in ¹⁸O₂ atmosphere or taking H₂¹⁸O as solvent, the WO₃{001} photocatalyst has been recycled and discharged into another CH₄ oxidation system without O₂ addition and with H₂O as solvent. Taking the recycled WO₃{001} from ¹⁸O₂ atmosphere as photocatalyst, the signal of HCH¹⁸O with is observed, confirming that the lattice-O atom of WO₃{001} is indeed able to participate in the formation of HCHO and can be supplemented by O₂. However, only HCHO without ¹⁸O labelling is found taking the recycled WO₃{001} from H₂¹⁸O solvent as photocatalyst, which uncovers that the consumed lattice-O of WO₃{001} cannot be repaired by H₂O molecule. Additionally, we note that no product is produced in pure CH₄ atmosphere using recycled WO₃{110} as photocatalyst without

O₂ addition, which is reasonable considering that the consumed lattice-O of WO₃{110} is hardly repaired. Based on the oxygen isotope experiments, we get that the O-atoms of HCHO products from WO₃{001} and WO₃{110} are both from O₂ rather than H₂O. Besides, for WO₃{001} system, O₂ is involved in the formation of HCHO by repairing the lattice-O.

Figure R66. (a) GC-MS spectra of HCHO obtained in WO₃{001} system using ¹⁸O₂ as oxygen isotope and the recycled WO₃{001} as photocatalyst without O₂ addition. (b) GC-MS spectra of HCHO obtained in WO₃{110} system using ¹⁸O₂ or H₂¹⁸O as oxygen isotope. (c) GC-MS spectra of HCHO obtained in WO₃{001} system using H₂¹⁸O as oxygen isotope and the recycled WO₃{001} as photocatalyst without O₂ addition.

Figure R67. (a) Screenshot on GC-MS spectra of HCHO obtained in WO₃{001} system using ¹⁸O₂ as oxygen isotope and (b) the recycled WO₃{001} as photocatalyst without O₂ addition.

Figure R68. (a) Screenshot on GC-MS spectra of HCHO obtained in WO₃{110} system using ¹⁸O₂ or (b) H₂¹⁸O as oxygen isotope.

Figure R69. Screenshot on GC-MS spectra of HCHO obtained in $\text{WO}_3\{001\}$ system using H_2^{18}O as oxygen isotope and the recycled $\text{WO}_3\{001\}$ as photocatalyst without O_2 addition.

Our revision: The GC-MS results have been added in the revised manuscript (please see the red marked parts on pages 17-18) and the revised supplementary information (please see the red marked parts on page S4). Figure R66a-R66c have been added as Figure 8b-8d in the revised manuscript. Figure R67-R69 have been added as Figure S64-S66 in the revised supplementary information.

Question 8: There are several grammatical and typological errors which are present in the manuscript which include and are not limited to

- Figure 1 caption should provide descriptions for inset
- In Figure S3, ‘calculation’ should be ‘calcination’
- On Page S15, ‘carbohydrate’ should be carbon oxygenates

Our response: We are grateful for the reviewer’s careful reading. The descriptions for inset have been added in Figure 1 caption. In Figure S3, ‘calculation’ has been corrected to ‘calcination’. On Page S15, ‘carbohydrate’ has been corrected to carbon oxygenates. Besides, the whole manuscript has been carefully proofread to ensure the accuracy of grammar and word spelling.

Our revision: The whole manuscript has been carefully proofread to ensure the accuracy of grammar and word spelling.

REVIEWER COMMENTS

Reviewer #1 (Remarks to the Author):

1. Reaction mechanism. The authors propose that CH₄ is activated by O₂⁻ anion from O₂ photoreduction to form •CH₃ over WO₃ {110}, while CH₄ is activated by photo-induced lattice-O of WO₃ over WO₃ {001}. The authors performed control experiments to rule out the reaction pathway of CH₄ activation by •OH. However, according to very recently reported work published on Nature Communications ((2023)14:2690) and the authors' own work (J. Mater. Chem. A, 2020, 8, 13277; Nature Sustainability 4, 509–515 (2021)), CH₄ is exclusively activated by •OH radicals on WO₃ and other photocatalysts such as BiVO₄ and ZnO. ESR results in this work also show that •OH radicals are formed on both WO₃ {110} and {001}. It would be great for the readers if the author could compare and explain the different reaction mechanism between this work and the reported studies mentioned above in the manuscript.
2. Comparing Figure 2 f and Figure 2h, the trend of HCHO production and selectivity on WO₃{001} is quite similar to that of WO₃ {110}. When using 150 mL water, the selectivity of HCHO produced on WO₃ {001} and {110} are nearly 100% with similar amount. Does this mean that the reaction mechanism is not only affected by the crystal facets but also by the reaction conditions.
3. GC-MS results show that the O atoms in CH₃OH is from O₂, which is different from the authors' own work (J. Mater. Chem. A, 2020, 8, 13277). It would be great if the authors could explain the difference. In Figure 8 c and 8d, m/z=28 of HCHO is missing.
4. Page 3, line 17, WO₃ {001} should be {110}.

Reviewer #2 (Remarks to the Author):

The authors have conducted a more comprehensive investigation into this work, addressing a significant portion of the concerns raised in the previous evaluation. However, there are certain issues that must be attended to before the manuscript can be considered for publication.

1. The authors stated that “hydrothermal process with polyvinylpyrrolidone as capping agent and ammonium ion as directing agent for facet formation”. Are there any supporting statements or investigation conducted in this regard? If yes, please provide.
2. Reference peak used for XPS calibration is not stated in the manuscript.

3. The authors concluded the similar specific surface area of $\text{WO}_3\{001\}/\text{WO}_3\{110\}$ from BET investigation. However, the presented isotherm appears to lack substantiating evidence. Please provide the linear fit for specific surface area evaluation.

4. The authors claimed that no H_2 produced during CH_4 oxidation on $\text{WO}_3\{001\}$ from Figure S13. From the experimental details provided, the GC was equipped with He carrier gas. This raises concern as the use of He carrier gas in GC would lead to incorrect analysis of H_2 product, particularly for the low concentration generated from photocatalysis.

5. Confidence level on 100% selectivity. Based on the initial GC raw data provided in Figure S27-S35, the CO_2 baseline is not flat and it overlaps with the CH_4 peak. This could lead to some degrees of incorrectness towards data analysis and interpretation. Thus, how confident are the authors in claiming 100% selectivity towards CH_4 ?

6. The authors found out that the lattice-O in $\text{WO}_3\{110\}$ is unable to produce HCHO in pure CH_4 environment, but is capable in producing significant HCHO yield in $\text{CH}_4\text{-O}_2$ mixed atmosphere. Please provide a more detailed explanation.

7. The authors proposed the oxygen-replenishment mechanism in $\text{WO}_3\{001\}$ from surrounding O_2 . It is advised to provide cyclic and long-term stability test for the photocatalyst to showcase the as-claimed "100% replenishment".

8. The authors claimed that massive lattice-O is lost from $\text{WO}_3\{110\}$ in Figure 3b, without any observable carboxyl peaks nor formation of HCHO. Does it indicate that $\text{WO}_3\{110\}$ is prone to severe photocorrosion? Stability investigation shall be supplemented.

9. Insufficient details provided for DFT calculations. For example, the authors must state clearly on any correction factors adopted as well as the formula used to evaluate the "energy". Moreover, does the "energy" mentioned in the manuscript refer to binding energy, adsorption energy or Gibbs energy? It is apparent that these three terms are used interchangeably throughout the manuscript; such scenario is not deemed appropriate. Careful discussion is required.

10. The authors must check carefully and balance any chemical equations provided. For example, P14, L3: $\text{CH}_3\text{OOH} \rightarrow \text{HCHO}$ is not stoichiometrically balanced.

11. The authors are advised to illustrate the redox potential energy for the intermediate formation at least in the supplementary information, to better demonstrate the mechanism.

12. A meaningful unit must be provided for the vertical axis of photocurrent response in Figure S60, instead of an arbitrary unit.

Response to Reviewers

Dear Reviewers,

Thank you for your great effort and the insightful and constructive comments concerning on our manuscript entitled “Insight into selectivity of photocatalytic methane oxidation to formaldehyde on tungsten trioxide” (NCOMMS-22-40994A-Z). All the comments raised are valuable and very helpful for us to improve the quality of the manuscript, as well as the important guiding significance to our researches. We have studied all the comments carefully and made comprehensive corrections that we hope will meet your approval. All the revised portions are marked in “red” in the revised manuscript and the revised supplementary information. The point-to-point responses to the comments are listed as following:

Reviewer #1

Question 1: Reaction mechanism. The authors propose that CH₄ is activated by O₂⁻ anion from O₂ photoreduction to form •CH₃ over WO₃ {110}, while CH₄ is activated by photo-induced lattice-O of WO₃ over WO₃ {001}. The authors performed control experiments to rule out the reaction pathway of CH₄ activation by •OH. However, according to very recently reported work published on Nature Communications ((2023)14:2690) and the authors' own work (J. Mater. Chem. A, 2020, 8, 13277; Nature Sustainability 4, 509–515 (2021)), CH₄ is exclusively activated by •OH radicals on WO₃ and other photocatalysts such as BiVO₄ and ZnO. ESR results in this work also show that •OH radicals are formed on both WO₃ {110} and {001}. It would be great for the readers if the author could compare and explain the different reaction mechanism between this work and the reported studies mentioned above in the manuscript.

Our response: We appreciate the reviewer’s comprehensive analysis and are grateful for the insightful comment. The different reaction mechanism between this work and the reported studies mentioned above is mainly attributed to the follows.

The different reaction mechanism is primarily associated with different reactive surface environment. Since the publication of *Luo, L. et al. Nat. Commun. 14, 2690 (2023)* is also focused on using WO₃ for CH₄ oxidation, the comparison of reactive surface environment between their and our work can best reveal the distinct reaction mechanism. Through careful reading, we find that the photocatalyst in *Luo, L. et al. Nat. Commun. 14, 2690 (2023)* is single-atom Cu and W^{δ+} co-modified WO₃ (Cu_{0.029}-

def-WO₃) rather than pristine WO₃ semiconductor. Therein, single-atom Cu exists in the form of Cu²⁺ bonding to two lattice-O adjacent to W, and serves as the electron acceptor (below Figure R1a). W^{δ+} is the reduced state of W⁶⁺ in WO₃, which is induced by the oxygen vacancy (O_v). According to the previous reports (*Pastor, E. et al. Nat. Rev. Mater.* 7, 503–521 (2022); *Wang, Z. et al. J. Phys. Chem. Lett.* 3, 102–106 (2012); *Esch, F. Science* 309, 752-755 (2005)), O₂ is easily adsorbed at W^{δ+} and reduced to repair O_v or generate reactive oxygen species (such as O₂⁻). However, for the Cu_{0.029}-def-WO₃ photocatalyst in the report of *Luo, L. et al. Nat. Commun.* 14, 2690 (2023), since the adjacent Cu²⁺ acts as the electron acceptor, it abstracts the electron from W^{δ+} that becomes a hole acceptor to generate W^{(δ+1)+}. As-formed W^{(δ+1)+} then acts as the oxidation site for H₂O to generate [•]OH and H⁺ rather than the O₂ reduction site (Figure R1b). Therefore, a large number of adsorbed [•]OH is formed and becomes the main reactive species for CH₄ activation and oxidation to form HCHO. Meanwhile, the simulation result reveals that as-reduced Cu⁺ can adsorb O₂ and the bond length of adsorbed O₂ is largely stretched to 1.41 Å in the molecular form (Figure R2a and R2c). Due to the low oxidation states of Cu⁺, this stretched O₂ is further readily protonated by the adsorbed H₂O on W^{(δ+1)+} active site (Figure R2b), which is reflected by the DMPO-[•]OOH signal with six prominent characteristic signals (purple curve in Figure R2d). Noteworthily, without single-atom Cu modification, O₂ is mainly reduced to DMPO-[•]O₂⁻ with four-fold peaks on def-WO₃ and WO₃ (green and orange curves in Figure R2d), which is consistent with our result. Therefore, no steady O₂⁻ over Cu_{0.029}-def-WO₃ is formed to activate and oxidize CH₄. The above reaction mechanism is also applicable to Au modified ZnO sample and q-BiVO₄ nanoparticles in our own works (*Fan, Y. et al. J. Mater. Chem. A* 8, 13277–13284 (2020); *Fan, Y. et al. Nat. Sustain.* 4, 509–515 (2021)), where Au acts as an electron acceptor to abstract the photoelectrons from Zn atom of ZnO (Figure R3a), and V element of q-BiVO₄ can share the loading of photoelectrons from Bi element (Figure R3b).

Figure R1. (a) The optimized configuration of atomic Cu at O_v site of def- WO_3 over the (002) surface in the top view ($\text{Cu}_{0.029}$ -def- WO_3). Therein, the Cu^{2+} species serves as the electron acceptor, while the adjacent $\text{W}^{\delta+}$ species acts as the hole acceptor. (b) $\text{W}^{(\delta+1)+}$ acting as the oxidation site for H_2O to generate $\cdot\text{OH}$ and H^+ rather than the O_2 reduction site (Luo, L. et al. Nat. Commun. 14, 2690 (2023))

Figure R2. (a) The adsorbed O_2 molecule with a stretched bond length of 1.41 Å at Cu^+ site, and (b, c) protonation of the stretched O_2 by H_2O adsorbed at $\text{W}^{(\delta+1)+}$ site. (d) $\text{DMPO}\cdot\text{OOH}$ with six-fold signals on $\text{Cu}_{0.029}$ -def- WO_3 and $\text{DMPO}\cdot\text{O}_2^-$ with four-fold peaks on def- WO_3 and WO_3 . (Luo, L. et al. Nat. Commun. 14, 2690 (2023))

Figure R3. (a) Quenching the photoluminescence of ZnO by Au electron acceptor (*Fan, Y. et al. J. Mater. Chem. A 8, 13277–13284 (2020)*). (b) Photoelectrons transfer from Bi element to V element in q-BiVO₄ (*Fan, Y. et al. Nat. Sustain. 4, 509–515 (2021)*).

Figure R4. (a) Gradual enrichment of O_v defects on WO₃{110} with W atom as the electron acceptor. (b) EPR signals of DMPO-O₂⁻ and DMPO-[•]OOH without and with CH₄ addition in WO₃{110} system. (c) The terminal -OH groups on the surface of WO₃{110}. (d) Quenching [•]OH radical by the terminated OH group on WO₃{110}

surface.

Figure R5. Preferred absorption of free $\cdot\text{OH}$ radicals at (a) Ti_{5c} sites of TiO_2 (*Tan, S. et al. J. Am. Chem. Soc. 134, 9978–9985 (2012)*), (b) $\text{W}^{\delta+}$ sites of W -doped $\text{Ni}(\text{OH})_2$ (*Yan, J. et al. Nat. Commun. 10, 2149 (2019)*) and (c) $\text{Bi}^{\delta+}$ sites of BiOCl (*Li, H. et al. Environ. Sci. Technol. 51, 5685–5694 (2017)*), where the structure of adsorbed $\cdot\text{OH}$ radical is equal to the terminal $-\text{OH}$ group.

While in our manuscript, because no external electron acceptor is added, the W atom is the electron acceptor as the conduction band minimum of WO_3 is mainly constituted by $\text{W}5d$ orbitals. For pristine $\text{WO}_3\{110\}$, as described in our manuscript, H_2O molecules are easily adsorbed on lattice- O of $\text{WO}_3\{110\}$ and oxidized to produce $\cdot\text{OH}$ radicals with the generation of O_v and reduced W atom (W^{5+}) (Figure R4a, INT 4). Therefore, during the photocatalytic process, $\text{WO}_3\{110\}$ is gradually enriched with O_v defects, which is similar to def- WO_3 in *Luo, L. et al. Nat. Commun. 14, 2690 (2023)*. Subsequently, the O_v site adsorbs O_2 and undergoes one-electron reduction to O_2^- or 4-electron reduction to repair lattice- O . Owing to the slow dynamic characteristics of 4-electron reduction, O_2 is more easily reduced to O_2^- by one electron with a minor

reduction potential (-0.046 V *vs* NHE). Besides, similar to def-WO₃ in *Luo, L. et al. Nat. Commun. 14, 2690 (2023)*, the generated O₂⁻ will not be directly protonated by adjacent surface -OH group and thus the evident signal of DMPO-[•]O₂⁻ is found in absence of CH₄ (Figure R4b). This guarantees its reactivity for CH₄ activation and oxidation, which is proved by the signal of DMPO-[•]OOH with CH₄ addition. Even if we assume that [•]OH can activate CH₄ in our WO₃{110} system, it must be the adsorbed [•]OH according to the reports from *Luo, L. et al. Nat. Commun. 14, 2690 (2023)* and our own works (*Fan, Y. et al. J. Mater. Chem. A 8, 13277–13284 (2020)*; *Fan, Y. et al. Nat. Sustain. 4, 509–515 (2021)*). Based on the previous reports (*Tan, S. et al. J. Am. Chem. Soc. 134, 9978–9985 (2012)*; *Yan, J. et al. Nat. Commun. 10, 2149 (2019)*; *Li, H. et al. Environ. Sci. Technol. 51, 5685–5694 (2017)*), the free [•]OH radicals are inclined to be adsorbed at the cation sites of metal oxide such as Ti_{5c} sites of TiO₂ (Figure R5a), W^{δ+} sites of W-doped Ni(OH)₂ (Figure R5b) and Bi^{δ+} sites of BiOCl (Figure R5c), where the structure of adsorbed [•]OH radical is equal to the terminal -OH group. For the pristine WO₃{110} surface, the terminal -OH already exists (Figure R4c), so additional adsorption of [•]OH radical results in the presence of two -OH groups at a W site. Two adjacent -OH groups inevitably combine and shed one H₂O molecule, i.e. [•]OH radical is quenched by the surface -OH group (Figure R4d), which has already been explained in our previous version of “Responses to Comments”. In brief, unlike the previous works of *Luo, L. et al. Nat. Commun. 14, 2690 (2023)*, *Fan, Y. et al. J. Mater. Chem. A 8, 13277–13284 (2020)* and *Fan, Y. et al. Nat. Sustain. 4, 509–515 (2021)*, the different surface structure of WO₃{110} makes [•]OH radicals unable to activate and oxidize CH₄ molecule. Analogously, the terminal -OH groups also exist on the surface of WO₃{001}, thus the [•]OH radicals cannot activate and oxidize CH₄ in WO₃{001} systems. Furthermore, in our previous version of “Responses to Comments” and “manuscript”, we have verified that the lattice-O of WO₃{001} can oxidize CH₄ directly to produce HCHO, which is not affected by the [•]OH radical formation process.

Our revision: The analysis about different reaction mechanism between this work and the previous works (*Luo, L. et al. Nat. Commun. 14, 2690 (2023)*; *Fan, Y. et al. J. Mater. Chem. A 8, 13277–13284 (2020)*; *Fan, Y. et al. Nat. Sustain. 4, 509–515 (2021)*) has been added in the revised manuscript on page of 17 and the revised supplementary information in Part II on pages of S5-S6.

Question 2: Comparing Figure 2f and Figure 2h, the trend of HCHO production and selectivity on $\text{WO}_3\{001\}$ is quite similar to that of $\text{WO}_3\{110\}$. When using 150 mL water, the selectivity of HCHO produced on $\text{WO}_3\{001\}$ and $\{110\}$ are nearly 100% with similar amount. Does this mean that the reaction mechanism is not only affected by the crystal facets but also by the reaction conditions?

Our response: We are grateful for the reviewer's professional comment. We totally agree that the reaction conditions can affect the reaction mechanism, but it usually happens under the extreme conditions. For example (Luo, L. et al. *Nat. Commun.* 14, 2690 (2023)), only CO_2 was produced without H_2O addition for CH_4 oxidation under O_2 atmosphere by $\text{Cu}_{0.029}\text{-def-WO}_3$; whereas, after H_2O addition, HCHO appears as a product. This is because H_2O can dissolve HCHO and promptly desorb it from the catalyst surface, preventing its overoxidation to CO_2 . The desorption process of HCHO by H_2O also involves the breaking of adsorption bonds, namely, a new reaction pathway is opened up to cause a mechanistic change.

Figure R6. (a) No change happens on the reaction environment with H_2O volume increasing from 5 to 150 mL. (b) Decreased concentration of both HCHO and photocatalyst with H_2O volume increasing from 5 to 150 mL, severely inhibiting the CO_2 generation. (c) Photocatalytic CH_4 oxidation performance on $\text{WO}_3\{001\}$ at 50 °C

with variation of H₂O amount. (d) Corresponding decreased concentration with the increase of H₂O amount at 50 °C reaction time. Reaction condition: 10 mg catalyst, 3 h reaction time, Xenon light 150 mW cm⁻², 7 bar O₂ + 13 bar CH₄.

Figure R7. (a) Gibbs free energy corresponding to the formation of CH₃OOH, CH₃OH, HCHO and CO₂ from CH₄ oxidation at 298 K. (b) The productivity changes of oxygenates generated over WO₃{110} with 150 mL H₂O, 9 bar O₂ + 11 bar CH₄ at 50 °C, before and (c) after light irradiation treatment under air or N₂ atmosphere. (d) The corresponding UV-visible absorption spectra of HCHO product and (e-g) ¹H NMR spectra before and after light irradiation treatment under air or N₂ atmosphere.

Irradiation treatment: the product solution was centrifuged to remove the photocatalyst, and then exposed to Xenon light irradiation under air or N₂ atmosphere for 20 min.

On the contrary, the HCHO selectivity promotion along with H₂O volume increasing from 5 to 150 mL for both WO₃{001} and WO₃{110} in our work does not involve the change of reaction mechanism. This is because 5 mL H₂O is sufficient to disperse 10 mg WO₃{001} or WO₃{110} nanoparticles, and such adequate dispersion of photocatalyst in aqueous solution is kept along with the volume of H₂O increasing to 150 mL (Figure R6a). Therefore, the change in H₂O volume can only change the concentration of reactants and products, rather than change the reaction pathway (mechanism). The reason for the HCHO selectivity enhancement over WO₃{001} and WO₃{110} along with H₂O volume increase is analyzed as follows.

For WO₃{001} at both 25°C and 50°C reaction temperature, CH₄ can be oxidized directly to HCHO, which has been firmly confirmed in our previous version of “Responses to Comments” and “manuscript”. Due to the small volume of 5 mL H₂O, the produced HCHO and photocatalyst of relatively high concentration possess the high possibility to collide with each other, leading to the further oxidation of HCHO to CO₂ (Figure R6b). When the H₂O volume increases to 150 mL, despite the total amount of HCHO increasing (Figure R6c), the concentration of both HCHO and photocatalyst decreases, severely inhibiting their collision and CO₂ generation (Figure R6d). Consequently, the selectivity of HCHO nearly reaches 100 % in 150 mL H₂O. Additionally, we note that the CO₂ signal disappearance is not equal to no any CO₂ product, because all the GC instruments have a detection limit. In short, the inhibition of CO₂ generation does not mean that the reaction mechanism is changed.

For WO₃{110}, owing to the existence of CH₃OOH and CH₃OH as intermediates, the selectivity enhancement of HCHO along with H₂O volume increasing should be separately discussed at 25°C and 50°C. The CH₄ oxidation pathway on WO₃{110} is CH₄ → CH₃OOH → HCHO (CH₃OH) → CO₂, where the production of HCHO may be accompanied by CH₃OOH and CH₃OH as intermediates. The signal disappearance and appearance of CH₃OOH and CH₃OH at 25°C and 50°C, respectively, are attributed to their rate coefficients (*k**) of CH₄ oxidation, which has been confirmed in the previous version of “Responses to Comments” and “supplementary information”. To offer a complete answer here, the brief explanation is given as below. The *k** of WO₃{110} with radical mechanism exhibits a non-Arrhenius dependence and increases

with the reaction temperature (Equation R1).

$$k^* = A^*T^n \exp(-E_a^*/RT) \text{ or } k^* = A^*T^6 \exp(-E_a^*/RT) \quad (\text{Equation R1})$$

where k^* represents the rate constant, A^* is the preexponential factor, T is the reaction temperature, E_a^* is the activation energy and calculated to be 16.71 kJ mol⁻¹, R is the molar gas constant, and n is as high as 6.

At low temperature of 25°C, the high E_a^* value leads to slow k^* and low productivity of oxygenates (HCHO, 6.19 μmol m⁻² within 5 mL H₂O). Amongst such low yield of oxygenates, it is reasonable to miss the intermediate signals of CH₃OOH and CH₃OH. Similar to WO₃{001} system, along with H₂O volume increasing, the concentration of oxygenates become even lower, thus the intermediate signals of CH₃OOH and CH₃OH are more unlikely to appear. Moreover, the reduced concentration of HCHO avoids its overoxidation. Ultimately, the selectivity of HCHO on WO₃{110} reaches 100% at 25°C.

When the reaction temperature rises to 50°C, the k^* of WO₃{110} is considerably promoted with the productivity of oxygenates reaching 16.88 μmol m⁻² within 5 mL H₂O. The promoted reaction rate k^* of WO₃{110} at 50°C also accelerates the formation of intermediate products, contributing to the appearance of CH₃OOH and CH₃OH signals. Along with H₂O volume increasing from 5 to 150 mL, all productivities of CH₃OOH, HCHO and CH₃OH are boosted. Meanwhile, the HCHO selectivity is improved from 58.01% (5 mL H₂O) to 76.88% (150 mL H₂O), indicating that HCHO is always the primary product. This is because, based on the Gibbs free energy, the conversion of CH₃OOH to HCHO is thermodynamically favorable (Figure R7a). Furthermore, it is found that CH₃OOH spontaneously and rapidly decomposes to HCHO under light irradiation without photocatalyst no matter in air or N₂ atmosphere. As shown in Figure R7b, R7d and R7e (at 50°C), the sum of productivity of CH₃OOH + HCHO is ~ 68.54 μmol m⁻². After light irradiation treatment without photocatalyst no matter in air or N₂ atmosphere (Figure R7c, R7d, R7f and R7g), the signal of CH₃OOH disappears while the productivity of HCHO increases to ~ 68.41 μmol m⁻² (the curves in Figure R7d demonstrate the increased productivity of HCHO no matter in air or N₂ atmosphere). Thus, we can deduce that CH₃OOH is readily decomposed to HCHO, which is thermodynamically and kinetically favorable. Whereas, CH₃OOH → CH₃OH is an electron reduction process, which is relatively unfavorable in thermodynamics and dynamics. Therefore, along with H₂O volume increasing, the

increase in the productivity of HCHO at 50°C is greater than CH₃OOH and CH₃OH. The reason for the CO₂ signal disappearance in WO₃{110} system is the same as that in WO₃{001} system. Altogether, with the increase of H₂O amount at 50°C, the productivity and selectivity of HCHO is obviously improved.

Our revision: The corresponding explanation on HCHO selectivity change along with H₂O volume increase has been added in the revised manuscript (please see the red marked parts on page 8) and the revised supplementary information (please see the red marked parts on page S45-S48). Figure R6c and R7b have been added as Figure 2g and 2i in the revised manuscript. Figure R6a, R6b, R6d, R7c, R7d, R7e, R7f and R7g has been added as Figure S36a, S36b, S36c, S37a, S32+ S37b, S33c, S37c and S37d in the revised supplementary information.

Question 3: GC-MS results show that the O atoms in CH₃OH is from O₂, which is different from the authors' own work (*J. Mater. Chem. A*, 2020, 8, 13277). It would be great if the authors could explain the difference. In Figure 8c and 8d, m/z = 28 of HCHO is missing.

Our response: We really appreciate the reviewer's careful reading and the insightful comment. The different oxygen source of CH₃OH between this work and the previous work (*Fan, Y. et al. J. Mater. Chem. A* 8, 13277–13284 (2020)) is mainly attributed to their different active sites for CH₄ oxidation.

In the work of *Fan, Y. et al. J. Mater. Chem. A* 8, 13277–13284 (2020), Au nanoparticles are the active sites, which inhibits the O₂⁻ formation from O₂ reduction. Thus, CH₃OH is not synthesized using CH₃OOH as intermediates, but from the combination between [•]CH₃ and [•]OH. Therefore, the O atoms in CH₃OH is mainly from H₂O with a small part resulting from O₂. The detailed analysis is as follows. In the work of *Fan, Y. et al. J. Mater. Chem. A* 8, 13277–13284 (2020), the photocatalyst is the Au nanoparticles modified ZnO, such as Au_{0.75}/ZnO, where the 0.75 is the mass ratio (0.75 wt%) between Au and ZnO nanoparticles. The previous literatures have revealed that Au nanoparticles act as the active sites for CH₄ oxidation to form CH₃OH through the combination between [•]CH₃ and [•]OH (or [•]OOH) (Figure R8. *Jiang, Y. et al. J. Am. Chem. Soc.* 145, 2698–2707 (2023); *Luo, L. et al. Nat. Commun.* 12, 1218 (2021); *Song, H. et al. J. Am. Chem. Soc.* 141, 20507-20515 (2019); *Agarwal N. et al. Science* 358, 223–227 (2017)). In our work of *Fan, Y. et al. J. Mater. Chem. A* 8, 13277–13284

(2020), $\cdot\text{OH}$ radical is largely produced with 80% from H_2O oxidation and 20% from O_2 reduction (Figure R9a). Noteworthy, no $\cdot\text{OOH}$ signal is detected, indicating that O_2 cannot be reduced to O_2^- (Figure R9b). This is because O_2 is mainly adsorbed in the side-on form (Figure R9c) on the surface of Au nanoparticles, which is conducive to breaking O-O bond of O_2 forming $\cdot\text{OH}$ radicals rather than O_2^- (*Jiang, Y. et al. J. Am. Chem. Soc. 145, 2698–2707 (2023)*). Thus, on the surface of $\text{Au}_{0.75}/\text{ZnO}$, CH_3OH is generated by the combination of $\cdot\text{CH}_3$ and $\cdot\text{OH}$ radicals, and the $\cdot\text{OH}$ radical is mainly from H_2O oxidation with a small part resulting from O_2 (Figure R9d, R9e). Nevertheless, for $\text{WO}_3\{110\}$ in our current work, the generated amount of $\cdot\text{OH}$ radical is small and O_2 can be reduced to O_2^- , which oxidizes CH_4 to CH_3OOH . The CH_3OH in $\text{WO}_3\{110\}$ system is then generated by CH_3OOH reduction, so the O-source is from O_2 . The reason why CH_4 is not oxidized by $\cdot\text{OH}$ radical has been explained in detail in Question 4.2 for Reviewer #2 in the previous version of “Response to Editors and Reviewers”.

entry	catalyst	amount of product (μmol)				
		CH ₃ OOH	CH ₃ OH	HCHO	CO	CO ₂
1	ZnO	0	0	25.2	0.04	1.2
2	TiO ₂ (P25)	0	0	43.0	0.06	6.5
3 ^c	0.1 wt % Pt/ZnO	65.6	44.5	101.1	0.6	20.7
4 ^c	0.1 wt % Pd/ZnO	41.3	60.7	110.3	0.3	19.5
5 ^c	0.1 wt % Au/ZnO	123.4	41.2	86.3	0.4	11.6

Figure R8. Au nanoparticles acting as the active sites for CH₄ oxidation through the combination between •CH₃ and •OH (and •OOH) to produce CH₃OH.

Figure R9. (a) Fluorescence spectra of 1 mM coumarin solution over Au_{0.75}/ZnO with or without O₂ incorporation. The •OH radical is largely produced with 80% from H₂O oxidation and 20% from O₂ reduction. (b) No •OOH was generated over Au_{0.75}/ZnO with O₂ addition from EPR spectrum. (c) O₂ adsorption in side-on form on the surface of Au nanoparticles. (d) CH₃OH production by combination of •CH₃ and •OH radicals. (e) GC-MS spectra of CH₃OH as the product from CH₄ oxidation using Au_{0.75}/ZnO as the photocatalyst with H₂¹⁸O + ¹⁶O₂ or H₂¹⁶O + ¹⁸O₂, respectively.

Figure R10. (a) Standard HCHO MS spectrum. (b) GCMS column type used in our work. (c) Two impurity peaks at $m/z = 28$ ($[\text{Si}]^+$) and 32 ($[\text{O}_2]^+$) observed in Wax column.

Figure R11. (a) GCMS spectra of commercial HCHO solution (0.3 mmol L⁻¹) before and (b) after impurity deduction. (c) GCMS spectra of commercial HCHO solution (0.1 mmol L⁻¹) before and (d) after impurity deduction. (e) GCMS spectra of commercial HCHO solution (0.05 mmol L⁻¹) before and (f) after impurity deduction.

Figure R12. (a) Similarity retrieval for 0.3 mmol L⁻¹, (b) 0.1 mmol L⁻¹, (c) 0.05 mmol L⁻¹ of commercial HCHO solution, respectively.

Figure R13. (a) GCMS spectra of HCHO obtained in WO₃{110} system using ¹⁸O₂ as oxygen isotope before and (b) after impurity deduction. (c) GCMS spectra of HCHO obtained in WO₃{110} system using H₂¹⁸O as oxygen isotope before and (d) after impurity deduction.

Figure R14. (a) GCMS spectra of HCHO obtained in $\text{WO}_3\{001\}$ system using H_2^{18}O as oxygen isotope before and (b) after impurity deduction. (c) GCMS spectra of HCHO obtained in $\text{WO}_3\{001\}$ system using the recycled $\text{WO}_3\{001\}$ as photocatalyst without O_2 addition before and (d) after impurity deduction.

Figure R15. (a) Similarity retrieval of HCHO obtained in $\text{WO}_3\{110\}$ system using H_2^{18}O as oxygen isotope. (b) Similarity retrieval of HCHO obtained in $\text{WO}_3\{001\}$ system using H_2^{18}O as oxygen isotope. (c) Similarity retrieval of HCHO obtained in $\text{WO}_3\{001\}$ system using the recycled $\text{WO}_3\{001\}$ as photocatalyst without O_2 addition.

According to the standard GCMS spectrum of HCHO, a small peak should exist at $m/z = 28$ (Figure R10a). Nevertheless, in our real test, only high concentration instead of low concentration of HCHO displays the peak of $m/z = 28$. This is because the GCMS column for HCHO aqueous solution test is the Wax column (for example SH-PolarWax in our work, Figure R10b) containing SiO_2 component, which inevitably brings two impurity peaks at $m/z = 28$ ($[\text{Si}]^+$) and 32 ($[\text{O}_2]^+$, Figure R10c). Therefore, the impurity peak of $[\text{Si}]^+$ ($m/z = 28$) overlaps that of $[\text{CO}]^+$ ($m/z = 28$) fragment of HCHO product. To obtain the convincing GCMS result, the impurity peak must be deducted during the GCMS spectroscopy observation, which will weaken or even erase the $[\text{CO}]^+$ peak ($m/z = 28$) in low concentration HCHO solution. For examples, as shown in Figure R11a and R11c, for high HCHO concentrations (0.3 mmol L^{-1} and 0.1 mmol L^{-1}) before SiO_2 impurity peaks deduction, their GCMS spectra are almost identical to the standard spectra of HCHO (Figure R10a). However, through careful

observation, we can find that compared with 0.3 mmol L⁻¹ HCHO solution ([H₂CO]⁺, *I* = 92%; [CO]⁺ + [Si]⁺, *I* = 30%; [O₂]⁺, *I* = 2%), the peak of 0.1 mmol L⁻¹ HCHO solution at *m/z* = 30 ([H₂CO]⁺, *I* = 86%) is lower whereas the peak at *m/z* = 28 ([CO]⁺ + [Si]⁺, *I* = 40%) and the peak at *m/z* = 32 ([O₂]⁺, *I* = 5%) become more obvious. Therein, the *I* is the percentage of peak intensity. This is because the lower concentration of 0.1 mmol L⁻¹ HCHO solution leads to a decrease in the proportion of HCHO ion fragments and an increase in the proportion of impurity ion fragments. After impurity deduction, the peak intensity of *m/z* = 28 ([Si]⁺, Figure R11b) and *m/z* = 32 ([O₂]⁺, Figure R11d) decreases and disappears, respectively. The corrected HCHO GCMS spectrum shows 95% (Figure R12a) and 94% (Figure R12b) similarity with the standard HCHO spectrum. While, for the HCHO solution of low concentration (0.05 mmol L⁻¹, Figure R11e), its ion fragment abundance would be lower than the impurity abundance ([HCO]⁺, *I* = 47%; [H₂CO]⁺, *I* = 30%; [CO]⁺ + [Si]⁺, *I* = 50%; [O₂]⁺, *I* = 15%). Through the impurity deduction, the impurity ion fragments of [Si]⁺ and [O₂]⁺ are removed, and the *I* value of [HCO]⁺ and [H₂CO]⁺ is improved to 100% and 90%, respectively. We also note that because the concentration of HCHO is lower than that of the impurity abundance and its [CO]⁺ has the same *m/z* value with [Si]⁺, no peak of [CO]⁺ at *m/z* = 28 is observed for 0.05 mmol L⁻¹ HCHO after the impurity deduction (Figure R11f). The absence of *m/z* = 28 does not negate the qualitative accuracy of HCHO, which is proved by its 86% similarity with the standard HCHO spectrum (Figure R12c). Similarly, for Figure 8c and 8d in our work, the missing of *m/z* = 28 in low HCHO concentration test is reasonable after impurity deduction (Figure R13a-R13d and Figure R14a-R14d) with the high similarity (Figure R15a-c).

Our revision: The different oxygen source analysis of CH₃OH between this work and the previous work (*Fan, Y. et al. J. Mater. Chem. A* 8, 13277–13284 (2020)) has been added in the revised manuscript on page of 19 and the revised supplementary information in Part II on pages of S6-S7. The explanation of the missing peak signal at *m/z* = 28 has been added in the revised manuscript on page of 19 and the revised supplementary information on pages of S96-S102. Figure R10-R15 has been added as Figure S80-S85 in the revised supplementary information.

Question 4: Page 3, line 17, WO₃ {001} should be {110}.

Our response: We are grateful for the reviewer's careful reading. The $\text{WO}_3\{001\}$ has been corrected to $\text{WO}_3\{110\}$.

Our revision: The $\text{WO}_3\{001\}$ has been corrected to $\text{WO}_3\{110\}$ as shown in the revised manuscript on page of 3.

Reviewer #2

General comment: The authors have conducted a more comprehensive investigation into this work, addressing a significant portion of the concerns raised in the previous evaluation. However, there are certain issues that must be attended to before the manuscript can be considered for publication.

Our response: We truly appreciate the reviewer's encouraging comments. All of the insightful questions raised by the reviewer have been carefully thought and answered, which greatly improves the quality of our work.

Question 1: The authors stated that "hydrothermal process with polyvinylpyrrolidone as capping agent and ammonium ion as directing agent for facet formation". Are there any supporting statements or investigation conducted in this regard? If yes, please provide.

Our response: Thanks a lot for reviewer's kind reminding and constructive comment. For the crystal facet regulation of $\text{WO}_3\{001\}$ and $\text{WO}_3\{110\}$ with analysis, our references are *Li, Y. et al. J. Am. Chem. Soc. 140, 9078–9082 (2018)*; *Zhu, J. et al. Chem. Commun. 47, 4403–4405 (2011)*; *Safo, I. et al. Nanoscale Adv. 1, 3095-3106 (2019)*; *Wang, X. et al. Comp. Mater. Sci. 68, 218-221 (2013)*. The specific synthesis mechanism is analyzed as follows.

A simple hydrothermal method was used for $\text{WO}_3\{001\}$ and $\text{WO}_3\{110\}$ photocatalysts preparation (Figure 1a in the revised manuscript). Adjusting the pH value of Na_2WO_4 solution by acetic acid facilitated the decomposition of tungsten source into WO_3 substance. As a capping agent, polyvinylpyrrolidone (PVP) could coordinate to W^{6+} ion of $\text{WO}_3(001)$ plane through its oxygen atom (O) of pyrrolidone group at high temperature^{1,2}, directing the anisotropic growth of WO_3 nanosheet perpendicular to the $\langle 001 \rangle$ crystal axis. Rinsing thoroughly with water, the $\text{WO}_3\{001\}$ nanosheet was obtained. According to the previous reports^{1, 3, 4}, the NH_4^+ cation was inclined to stabilize the surface of (110) probably by interaction with terminal O^{2-} ion, thus $\text{WO}_3\{110\}$ with terminal crystal oxygen atoms was prepared. Finally, the calcination at 300°C was applied for both samples to fully remove the chemical adsorbed O_2 and the capping polyvinylpyrrolidone or ammonium from the catalyst surfaces.

1. Lin, R. et al. Quantitative Study of Charge Carrier Dynamics in Well-Defined WO_3

Nanowires and Nanosheets: Insight into the Crystal Facet Effect in Photocatalysis. *J. Am. Chem. Soc.* **140**, 9078–9082 (2018).

2. Safo, I., Werheid, M., Dosche, C., Oezaslan, M. The role of Polyvinylpyrrolidone (PVP) as Capping and Structure-Directing Agent in the Formation of Pt Nanocubes. *Nanoscale Adv.* **1**, 3095-3106 (2019)

3. Wang, X. et al. Surface stabilization of hexagonal WO₃ by non-metallic atoms: A DFT study. *Comp. Mater. Sci.* **68**, 218-221 (2013).

4. Zhu, J.; Wang, S.; Xie, S.; Li, H., Hexagonal single crystal growth of WO₃ nanorods along a [110] axis with enhanced adsorption capacity. *Chem. Commun.* **47**, 4403–4405 (2011).

Our revision: The principle for photocatalyst preparation with references has been added in the revised supplementary information on pages of S2 and Reference section.

Question 2: Reference peak used for XPS calibration is not stated in the manuscript.

Our response: Thanks a lot for reviewer’s careful reading and kind reminding. In “Characterization” section, the XPS test has been described as “X-ray photoelectron spectroscopy (XPS) experiments were carried out using an X-ray photoelectron spectrometer (EscaLab 250Xi, Thermo Scientific) and the spectra were calibrated with the C 1s peak at 284.8 eV.”

Our revision: The reference peak used for XPS calibration has been added in the revised manuscript on page of 20.

Question 3: The authors concluded the similar specific surface area of WO₃{001}/WO₃{110} from BET investigation. However, the presented isotherm appears to lack substantiating evidence. Please provide the linear fit for specific surface area evaluation.

Our response: Thanks a lot for the reviewer’s constructive comments. The calculation process of BET surface area including the linear fitting is as follows. BET surface area is known to be derived from N₂ adsorption/desorption isotherm and calculated by the BET isothermal equation in general form:

$$\frac{P}{V_{\text{ads}}(P_0 - P)} = \frac{C - 1}{V_m C} \left(\frac{P}{P_0} \right) + \frac{1}{V_m C}$$

where V_{ads} is the volume of gas adsorbed at P ; V_m is the monolayer volume of N₂

molecules on the surface of catalyst; P is the pressure of N_2 (at equilibrium); P_0 is the saturation pressure of N_2 ; C is the adsorption coefficient. Using P/P_0 as the horizontal coordinate and $P/[V_{ads}(P_0-P)]$ as the vertical coordinate, the V_m and C can be obtained. Then, surface area can be calculated by V_m .

As shown in below Figure R16a and R17a, during N_2 adsorption and desorption tests, despite a wide range of P/P_0 from 0.00 to 1.00 has been tested, only low-pressure region is used in the BET surface area calculation. This is because the low-pressure of P/P_0 ensures the monolayer adsorption of N_2 , guaranteeing the accurate calculation of photocatalyst surface area. Whereas, the high-pressure of P/P_0 would cause the N_2 multilayer adsorption, which cannot be used in photocatalyst surface area calculation. To determine the low-pressure region, the curves of $V_{ads}(1-P/P_0)$ vs P/P_0 are drawn (Figure R16b and R17b). For both $WO_3\{001\}$ and $WO_3\{110\}$, we can observe that the term $V_{ads}(1-P/P_0)$ continuously increases with P/P_0 in the range of 0 to 0.22. This means that in this low-pressure range, N_2 is adsorbed in form of monolayer (V_m) and the C is positive. Thus, in the range (P/P_0) of 0 to 0.22, the linear fit is performed between $P/[V_{ads}(P_0-P)]$ (i.e. $1/[Q(P_0/P-1)]$, $Q = V_{ads}/P$) and P/P_0 (Figure R16c and R17c). Through the intercept and slope values of the fitted lines, the V_m value is calculated to be $3.60 \text{ cm}^3 \text{ g}^{-1} \text{ STP}$ ($WO_3\{001\}$) and $3.82 \text{ cm}^3 \text{ g}^{-1} \text{ STP}$ ($WO_3\{110\}$) while C value is calculated to be 159.62 ($WO_3\{001\}$) and 169.38 ($WO_3\{110\}$). Therein, STP is the condition of standard temperature and pressure. Under the STP condition, each cm^3 of N_2 molecules paves into a monolayer and occupies an area of 4.354 m^2 . Thus, the BET surface area of $WO_3\{001\} = 3.60 \text{ cm}^3 \text{ g}^{-1} * 4.354 \text{ m}^2 \text{ cm}^{-3} = 15.70 \text{ m}^2 \text{ g}^{-1}$ and the BET surface area of $WO_3\{110\} = 3.82 \text{ cm}^3 \text{ g}^{-1} * 4.354 \text{ m}^2 \text{ cm}^{-3} = 16.63 \text{ m}^2 \text{ g}^{-1}$. This is the reason why $WO_3\{001\}$ and $WO_3\{110\}$ are equipped with different N_2 absorption and desorption curves, but possess similar specific surface area.

Figure R16. (a) N_2 adsorption–desorption isotherm curves of $\text{WO}_3\{001\}$, (b) the corresponding curve of $V_{\text{ads}}(1-P/P_0)$ vs P/P_0 to determine the low-pressure region. (c) Linear fit of $P/[V_{\text{ads}}(P_0-P)]$ (i.e. $1/[Q(P_0/P-1)]$, $Q = V_{\text{ads}}/P$) against P/P_0 under low-pressure region.

Figure R17. (a) N_2 adsorption–desorption isotherm curves of $\text{WO}_3\{110\}$, (b) the corresponding curve of $V_{\text{ads}}(1-P/P_0)$ vs P/P_0 to determine the low-pressure region. (c) Linear fit of $P/[V_{\text{ads}}(P_0-P)]$ (i.e. $1/[Q(P_0/P-1)]$, $Q = V_{\text{ads}}/P$) against P/P_0 under low-pressure region.

Our revision: The calculation process of BET surface area has been added in the revised manuscript on the page of 5 and the revised supplementary information on pages of S14-S15. Figure R16a-R16c and R17a-R17c has been added as Figure S6a-S6f.

Question 4: The authors claimed that no H_2 produced during CH_4 oxidation on $\text{WO}_3\{001\}$ from Figure S13. From the experimental details provided, the GC was equipped with He carrier gas. This raises concern as the use of He carrier gas in GC would lead to incorrect analysis of H_2 product, particularly for the low concentration generated from photocatalysis.

Our response: Thanks a lot for the reviewer’s constructive comments and sorry for the misleading information. The GC carrier gas for detecting H_2 signal is N_2 rather than He

using TCD detector. The carrier gas of He is used for GCMS test, such as the GCMS signals of HCHO and CH₃OH. The respective test conditions are described as follows.

Gaseous products were qualitatively and quantitatively determined by gas chromatography (GC) tests with flame ionization detector (FID) and thermal conductivity detector (TCD). Test condition of GC: inlet temperature 100°C, nitrogen as carrier gas with 0.1 MPa, column temperature of 60°C, FID temperature of 100°C, TCD bridge current of 60 mA.

The GC-mass spectrum (GC-MS) was performed on SHIMADZU with the SH-PolarWax column. Test condition of GCMS: inlet temperature 180°C, splitless inlet, helium as carrier gas, linear speed of 25.5 cm s⁻¹, column temperature of 40°C with 120°C pretreatment, GC-MS ion source temperature of 200°C.

Our revision: The test condition of GC has been added in the revised manuscript on page of 21-22.

Question 5: Confidence level on 100% selectivity. Based on the initial GC raw data provided in Figure S27-S35, the CO₂ baseline is not flat and it overlaps with the CH₄ peak. This could lead to some degrees of incorrectness towards data analysis and interpretation. Thus, how confident are the authors in claiming 100% selectivity towards CH₄?

Our response: Thanks a lot for the reviewer's critical comments. Exactly as the reviewer commented, the GC baseline for CO₂ testing in our laboratory is not perfectly flat and a small overlap happens between the peaks of CH₄ and CO₂, which may lead to some deviation in CO₂ result (Figure R18a). Actually, this deviation has been considered in the error bar as shown in Figure R19. To further verify this result, the gas products on WO₃{001} at 50°C reaction temperature with 7 bar O₂ + 13 bar CH₄ and 5 mL H₂O (Figure R18a) have been retested in Shiyanjia Lab (testing company, www.shiyanjia.com). As shown in Figure R18b, the baseline is perfectly flat and the peaks between CH₄ and CO₂ are completely separated. Through quantitative analysis, the productivities of CO₂ are 1.47 μmol m⁻² (GC in our lab) and 1.62 μmol m⁻² (GC in testing company), respectively, which is within the productivity range from 1.29 to 1.68 μmol m⁻² with error bar (7 bar O₂, Figure R19). Thus, the CO₂ data is reasonable in our manuscript.

The 100% selectivity towards CH₄ oxidation is obtained with WO₃{001} and

WO₃{110} in 150 mL H₂O, where the GC signal of CO₂ disappears (Figure S27-S35 as the reviewer mentioned). To determine whether the CO₂ signal truly disappears or is masked by the CH₄ peak, the gas products obtained with 150 mL H₂O have been detected in Shiyanjia Lab. As shown in Figure R20, no CO₂ signal is observed, indicating that no CO₂ is generated. Therefore, the HCHO selectivity over WO₃{001} (25°C and 50°C) and WO₃{110} (25°C) with 150 mL H₂O can be regarded as 100%.

Figure R18. (a) GC spectra of gas product from CH₄ oxidation on WO₃{001} tested in our lab and (b) in testing company. Peaks at 9.21 min (our lab), 13.02 min (testing company) and 19.11 min (our lab), 5.37 min (testing company) are attributed to residual CH₄ and produced CO₂, respectively. Reaction condition: 10 mg catalyst, 7 bar O₂, 13 bar CH₄, 3 h reaction time, Xenon light 150 mW cm⁻², reaction temperature 50°C.

Figure R19. Photocatalytic CH₄ oxidation performance on WO₃{001} at 50°C with variation of O₂ amounts. Reaction condition: 10 mg catalyst, 3 h reaction time, Xenon light 150 mW cm⁻², 5 mL H₂O, pressures of CH₄ + O₂ = 20 bar. **Error bars indicate standard deviations.**

Figure R20. (a) GC spectra of gas product from CH₄ oxidation on WO₃{001} with 150 mL H₂O at 25°C and (b) 50°C. (c) GC spectra of gas product from CH₄ oxidation on WO₃{110} with 150 mL H₂O at 25°C and (d) 50°C. Reaction condition: 10 mg catalyst, 3 h reaction time, Xenon light 150 mW cm⁻², (a, b) 7 bar O₂ + 13 bar CH₄, (c, d) 9 bar O₂ + 11 bar CH₄.

Our revision: The retest GC results have been added in the revised manuscript on the pages of 6, 7 and the revised supplementary information on the pages of S29, S44. Figure R19 is the Figure 2c in the revised manuscript. Figure R18a, R18b and R20 has been added as Figure S19, S20 and S35 in the revised supplementary information.

Question 6: The authors found out that the lattice-O in WO₃{110} is unable to produce HCHO in pure CH₄ environment, but is capable in producing significant HCHO yield in CH₄-O₂ mixed atmosphere. Please provide a more detailed explanation.

Our response: Thanks a lot for the reviewer's critical comments. To provide the more detailed and comprehensible explanation, the photocatalytic reactions on WO₃{110} surface in pure CH₄ atmosphere or mixed CH₄ + O₂ atmosphere have been separately described as follows.

Figure R21. Proposed reaction mechanism for photocatalytic oxidation of CH₄ on WO₃{110}, which has been divided into pure CH₄ atmosphere and mixed CH₄ + O₂ atmosphere.

Figure R22. (a) Energy diagrams of CH₄ and H₂O adsorption as well as CH₄ activation on the surface of WO₃{110}. (b) EPR spectra of WO₃{110} in pure CH₄ or CH₄ + O₂ atmosphere at room temperature with DMPO as the radical trapping agent in aqueous solution. (c) *In situ* DRIFTS spectra of WO₃{110} in pure CH₄ atmosphere with H₂O addition under different light irradiation time.

Figure R23. (a) Energy diagram of H₂O oxidation on the surface of WO₃{110}. The energy value of V_o + 2OH(g) on WO₃{110} is corrected with $-eU_g = -2.52$ eV. (b) Spontaneous dissociation or combination of terminal OH group or terminal H atom on WO₃{110} surface. (c) The pH value of solution after reaction in H₂O solution with 20 bar CH₄ addition for 3 h at 50°C.

In pure CH₄ atmosphere, only H₂O rather than CH₄ can be oxidized (Figure R21, the blue line region). Under light irradiation, since the conduction band and valence band of WO₃ are mainly constituted by W5d orbitals and O2p orbitals, respectively (*Li, W. et al. ACS Nano 8, 11770-11777 (2014); Wei, S. et al. Appl. Catal. B: Environ. 283, 119661 (2021)*), photoelectrons and photoholes generated on WO₃{110} are separately combined with W⁶⁺ and O_b²⁻ to form W⁵⁺ and O_b⁻. Therein, the O_b is the bridging O, which is the lattice-O. According to previous reports (*Luo, L. et al. J. Am. Chem. Soc. 144, 740-750 (2022); Luo, L. et al. Nat. Commun. 13, 2930 (2022); Jiang, Y. et al. J. Am. Chem. Soc. 144, 15977-15987 (2022)*), H₂O and CH₄ can be competitively oxidized by photoholes (O⁻). Thus, the DFT calculations have been carried out. The results reveal that in an aqueous environment, due to the large difference in the energy of H₂O and CH₄ molecules adsorption on the surface of WO₃{110} (Figure R22a), O_b preferentially adsorbs H₂O molecules to block the adsorption of CH₄ (O_b⁻*CH₄). Thus, only H₂O rather than CH₄ can be oxidized by O_b⁻. This result is experimentally verified by the generation of [•]OH from H₂O oxidation (Figure R22b), while no carboxyl peaks of CH₄ oxidation found in *in situ* DRIFTS test on WO₃{110} in pure CH₄ atmosphere with H₂O addition (Figure R22c). Moreover, as displayed in Figure R23a, the adsorbed H₂O molecule is easily oxidized to [•]OH radical through O_b + h⁺ + H₂O → V_o + 2[•]OH with the energy of -1.24 eV (V_o + 2OH(g)). During this process, O_b is consumed along with oxygen vacancy generation, further excluding the possibility of CH₄ oxidation at O_b active site. This result is proved by the signal of W-O consumption in absence of

CH₄ activation in *in situ* DRIFTS spectra of WO₃{110} in CH₄ atmospheres with H₂O addition (Figure R22c). Nevertheless, the [•]OH is quenched soon after its formation. This is because on the surface of WO₃{110}, the terminal OH group or the H atom is always spontaneously dissociated from the surface and likely combined to form H₂O molecule due to the high surface energy, which is proved by the DFT calculation (Figure R23b). And the number of dissociated terminal H atoms is greater than that of dissociated terminal OH groups, which is revealed by the acidic nature of solution (pH = 6.06) after reaction in 20 bar CH₄ atmosphere with H₂O as solvent (Figure R23c). Given this, the generated [•]OH is easily quenched by combination with H⁺ rather than participate in CH₄ activation. This result is confirmed by the fact that the detected number of [•]OH radical is far less than the consumed amount of lattice-O over WO₃{110} and no [•]CH₃ radical is observed in pure CH₄ atmosphere. Thus, in pure CH₄ environment, WO₃{110} is not capable to produce HCHO.

Figure R24. (a) Band energy diagram of WO₃{110}. (b) EPR spectra of WO₃{110} in O₂ atmosphere at 77 K temperature. The WO₃{110} is the recycled sample after CH₄ oxidation reaction without O₂. (c) EPR spectrum of WO₃{110} under light irradiation for 80 s with CH₄ and O₂ dissolved in methanol. DMPO is added to the reaction mixture as the radical trapping agent. The WO₃{110} is the recycled sample after CH₄ oxidation reaction without O₂.

In the mixed CH₄ + O₂ atmosphere, CH₄ can be oxidized by O₂ to produce HCHO (Figure R21, red line region). The energy band structure of WO₃{110} (Figure R24a) is established with the valence band energy of 2.46 V vs normal hydrogen electrode (NHE) and the conduction band energy of -0.06 V vs NHE, respectively. The formation

potential of O_2^- anion from O_2 reduction is reported to be -0.046 V vs NHE , which is lower than -0.06 V of $\text{WO}_3\{110\}$. Therefore, $\text{WO}_3\{110\}$ favors the formation of O_2^- anion, which is proved by its EPR signal (Figure R24b). As shown in Figure R22b and R24c, the O_2^- anion activates CH_4 molecule to produce $\cdot\text{CH}_3$ and $\cdot\text{OOH}$ radicals. Through combination between $\cdot\text{CH}_3$ and $\cdot\text{OOH}$ radicals, CH_3OOH is generated followed by decomposition to CH_3OH and HCHO .

Our revision: Figure R21 and R22a have been added as Figure 7b and 5b in the revised manuscript. Figure R22b, R22c, R23a, R23b and R23c have been added as Figure S52b, S49b, S54b, S58a and S58b in the revised supplementary information.

Question 7: The authors proposed the oxygen-replenishment mechanism in $\text{WO}_3\{001\}$ from surrounding O_2 . It is advised to provide cyclic and long-term stability test for the photocatalyst to showcase the as-claimed “100% replenishment”.

Our response: Thanks a lot for the reviewer’s constructive comments. Following the reviewer’s kind suggestion, the stability of photocatalyst has been evaluated to prove the 100% replenishment of lattice-O for $\text{WO}_3\{001\}$ from surrounding O_2 . After five photocatalytic cycles, the initial activity of $\text{WO}_3\{001\}$ is almost fully preserved (Figure R25 and R26). Besides, by extending the reaction time to 24 h, the productivity of oxygenates is increased by ~ 8.8 times (HCHO , $275.85\ \mu\text{mol m}^{-2}$; CO_2 , $2.3\ \mu\text{mol m}^{-2}$, Figure R27) compared with the reaction for 3 h (HCHO , $31.59\ \mu\text{mol m}^{-2}$; CO_2 , $0\ \mu\text{mol m}^{-2}$). Both cyclic test and long-term reaction reveal the satisfied performance stability of $\text{WO}_3\{001\}$ and again verifies the 100% replenishment of lattice-O on $\text{WO}_3\{001\}$ from surrounding O_2 .

Figure R25. Stability of $\text{WO}_3\{001\}$ for CH_4 oxidation. Reaction condition: 10 mg catalyst, 150 mL H_2O , 7 bar O_2 + 13 bar CH_4 , 3 h reaction time of each cycle, Xenon light 150 mW cm^{-2} , 50°C .

Figure R26. (a) UV-visible absorption spectra of HCHO product on WO₃{001} for each cycle reaction. (b-f) GC spectra of gas product from CH₄ oxidation on WO₃{001} for each cycle reaction. Reaction condition: 10 mg catalyst, 150 mL H₂O, 7 bar O₂ + 13 bar CH₄, 3 h reaction time of each cycle, Xenon light 150 mW cm⁻², 50°C.

Figure R27. (a) Photocatalytic CH₄ oxidation performance on WO₃{001} for 24 h reaction time. (b) The corresponding UV-visible absorption spectrum of HCHO product and (c) GC spectrum of gas product. Reaction condition: 10 mg catalyst, 150 mL H₂O, 7 bar O₂ + 13 bar CH₄, 24 h reaction time, Xenon light 150 mW cm⁻², 50°C.

Our revision: Both cyclic test and long-term reaction have been added in revised manuscript on the page of 8 and revised supplementary information on the pages of S49-S51. Figure R25-R27 has been added as Figure S38-S40 in the revised supplementary information.

Question 8: The authors claimed that massive lattice-O is lost from WO₃{110} in Figure 3b, without any observable carboxyl peaks nor formation of HCHO. Does it indicate that WO₃{110} is prone to severe photocorrosion? Stability investigation shall be supplemented.

Our response: Thanks a lot for the reviewer's insightful comments. To rule out lattice-O loss from photocorrosion, the *in situ* DRIFTS spectra of WO₃{110} have been carried out in pure Ar atmosphere with different irradiation time. As shown in Figure R28, no negative peak is observed upon the light irradiation from 0 to 25 min. Thus, the lattice-O loss of WO₃{110} in CH₄ atmosphere without H₂O or with H₂O is attributed to its reaction with CH₄ or H₂O rather than the photocorrosion. As the reviewer knows, the reactions between lattice-O of WO₃{110} with CH₄ or H₂O have already been clarified by the DFT calculations (Figure 5 in revised manuscript and S53 in revised

supplementary information), thus the corresponding explanation is not repeated here.

The performance stability of $\text{WO}_3\{110\}$ has been evaluated by the cyclic test and long-term reaction. After five photocatalytic cycles, the initial activity of $\text{WO}_3\{110\}$ is almost fully preserved (Figure R29-R31). Besides, by extending the reaction time to 24 h, the productivity of oxygenates is increased by ~ 7.8 times (CH_3OOH , $95.21 \mu\text{mol m}^{-2}$; CH_3OH , $40.02 \mu\text{mol m}^{-2}$; HCHO , $451.20 \mu\text{mol m}^{-2}$; CO_2 , $2.3 \mu\text{mol m}^{-2}$, Figure R32) compared with the reaction for 3 h (CH_3OOH , $11.91 \mu\text{mol m}^{-2}$; CH_3OH , $4.99 \mu\text{mol m}^{-2}$; HCHO , $58.19 \mu\text{mol m}^{-2}$; CO_2 , $0 \mu\text{mol m}^{-2}$). Both cyclic test and long-term reaction reveal the satisfied performance stability of $\text{WO}_3\{110\}$.

Figure R28. *In situ* DRIFTS spectra of $\text{WO}_3\{110\}$ in pure Ar atmosphere under different light irradiation time without H_2O addition.

Figure R29. Stability of $\text{WO}_3\{110\}$ for CH_4 oxidation. Reaction condition: 10 mg catalyst, 150 mL H_2O , 9 bar O_2 + 11 bar CH_4 , 3 h reaction time of each cycle, Xenon light 150 mW cm^{-2} , 50°C .

Figure R30. ^1H NMR spectra of products on $\text{WO}_3\{110\}$ for each cycle reaction. Reaction condition: 10 mg catalyst , $150\text{ mL H}_2\text{O}$, $9\text{ bar O}_2 + 11\text{ bar CH}_4$, 3 h reaction time of each cycle, $\text{Xenon light } 150\text{ mW cm}^{-2}$, 50°C .

Figure R31. (a) UV-visible absorption spectra of HCHO product on WO₃{110} for each cycle reaction. (b-f) GC spectra of gas product from CH₄ oxidation on WO₃{110} for each cycle reaction. Reaction condition: 10 mg catalyst, 150 mL H₂O, 9 bar O₂ + 11 bar CH₄, 3 h reaction time of each cycle, Xenon light 150 mW cm⁻², 50 °C.

Figure R32. (a) Photocatalytic CH_4 oxidation performance on $\text{WO}_3\{110\}$ for 24 h reaction time. (b) ^1H NMR spectrum of product. (c) The corresponding UV-visible absorption spectrum of HCHO product and (d) GC spectrum of gas product. Reaction condition: 10 mg catalyst, 150 mL H_2O , 9 bar O_2 + 11 bar CH_4 , 24 h reaction time, Xenon light 150 mW cm^{-2} , 50°C .

Our revision: Both cyclic test and long-term reaction with *in situ* DRIFTS spectra have been added in the revised manuscript on the page of 8 and the revised supplementary information on the pages of S52-S56. Figure R28-R32 has been added as Figure S41-S45 in the revised supplementary information.

Question 9: Insufficient details provided for DFT calculations. For example, the authors must state clearly on any correction factors adopted as well as the formula used to evaluate the “energy”. Moreover, does the “energy” mentioned in the manuscript refer to binding energy, adsorption energy or Gibbs energy? It is apparent that these three terms are used interchangeably throughout the manuscript; such scenario is not deemed appropriate. Careful discussion is required.

Our response: Thanks a lot for the reviewer’s careful reading and kind reminding. The corresponding correction factors and the formula for “Energy” (ΔE) are taken from the

following equations based on the previous reports (*Ji, Y. et al. J. Am. Chem. Soc. 138, 15896–15902 (2016)*; *Nolan, M. et al. Chem. Phys. Lett. 499, 126–130 (2010)*; *Castleton, C. et al. J. Chem. Phys. 127, 244704 (2007)*; *Fabris, S. et al. J. Phys. Chem. B, 109, 48 (2005)*; *He, Y. et al. J. Am. Chem. Soc. 142, 17119–17130 (2020)*; *Luo, L. et al. Nat. Commun. 13, 2930 (2022)*; *Nolan, M. et al. Solid State Ionics 177, 3069–3074 (2006)*).

$$\Delta E = \Delta E_{\text{DFT}} - neU_{\text{g}}$$

$$\Delta E_{\text{DFT}} = E_{\text{im/sub}} - E_0 = E_{\text{im/sub}} - (E_{\text{slab}} + E_{\text{gas}})$$

Where ΔE_{DFT} is the computed reaction energy obtained from VASP, $E_{\text{im/sub}}$ is the computed energy of the optimized intermediate/substrate system, and $E_0 = E_{\text{slab}} + E_{\text{gas}}$ is the computed energy of the initial state that is set to 0. n is the number of photoelectrons left on the model by oxygen vacancy (V_{o}) formation, e is the elementary charge, and U_{g} is the bandgap of the semiconductor. The latter term $-neU_{\text{g}}$ in the formula is introduced to compensate the energy of V_{o} formation. This is because upon formation of a neutral V_{o} , one photoelectron is supposed to undergo trapping (*Nolan, M. et al. Chem. Phys. Lett. 499, 126–130 (2010)*; *Ji, Y. et al. J. Am. Chem. Soc. 138, 15896–15902 (2016)*). The trapped photoelectron cannot be described properly by GGA because of the self-interaction error in DFT (*Ji, Y. et al. J. Am. Chem. Soc. 138, 15896–15902 (2016)*; *Mori-Sanchez, P. et al. Phys. Rev. Lett. 100, 146401 (2008)*). To solve this problem, the $-neU_{\text{g}}$ correction should be applied (*Ji, Y. et al. J. Am. Chem. Soc. 138, 15896–15902 (2016)*; *Nolan, M. et al. Chem. Phys. Lett. 499, 126–130 (2010)*; *Castleton, C. et al. J. Chem. Phys. 127, 244704 (2007)*; *Fabris, S. et al. J. Phys. Chem. B, 109, 48 (2005)*). Other computational details have been included in the “Computational details” section of the revised supplementary information

All the words that refer to “energy” in the revised manuscript and the revised supplementary information have been carefully considered and used appropriately. For examples, “binding energy” is used in the XPS spectra analysis; the original “adsorption energy” in DFT calculation section has been changed to “the energy for adsorption” in order to be consistent with the title of y-axis coordinate of energy diagram (such as Figure 5a and 5b in the revised manuscript); “Gibbs energy” has been

deleted and the relevant expressions have been changed as the “Our response” to “Question 10” below.

Our revision: The corresponding correction factors and the formula for “Energy” evaluation has been added in the revised supplementary information on the pages of S3. Careful description on theoretical calculation has been provided in the revised manuscript on the pages of 12-15.

Question 10: The authors must check carefully and balance any chemical equations provided. For example, P14, L3: $\text{CH}_3\text{OOH} \rightarrow \text{HCHO}$ is not stoichiometrically balanced.

Our response: Thanks a lot for the reviewer’s careful reading and constructive comments. The corresponding sentence has been changed to “As previous reports (*Feng, N. et al. Nat. Commun. 12, 4652 (2021)*; *Agarwal, N. et al. Science 358, 223–227 (2017)*; *Anglada, J. M. et al. Phys. Chem. Chem. Phys. 19, 12331–12342 (2017)*; *Jiang, Y. et al. J. Am. Chem. Soc. 145, 2698–2707 (2023)*), CH_3OOH can be spontaneously decomposed into HCHO, whereas the conversion of CH_3OOH into CH_3OH is an electron reduction process. Thus, equation 9 becomes the major pathway of HCHO formation”.

Our revision: The corresponding part has been added in revised manuscript on the page of 14-15.

Question 11: The authors are advised to illustrate the redox potential energy for the intermediate formation at least in the supplementary information, to better demonstrate the mechanism.

Our response: Thanks a lot for the reviewer’s insightful comments. According to equation 1-9 for CH_4 oxidation on $\text{WO}_3\{110\}$ in the revised manuscript, the CH_4 oxidation to HCHO and CH_3OH involves many intermediates including $\cdot\text{OH}$ from H_2O oxidation ($\cdot\text{OH}/\text{H}_2\text{O} = 2.38 \text{ eV vs NHE}$, *Nosaka, Y. et al. Chem. Rev. 117, 11302–11336 (2017)*; *Jiang, Y. et al. J. Am. Chem. Soc. 145, 2698–2707 (2023)*), $\cdot\text{OOH}$ from O_2 reduction ($\text{O}_2/\cdot\text{OOH} = -0.046 \text{ eV vs NHE}$, *Nosaka, Y. et al. Chem. Rev. 117, 11302–11336 (2017)*; *Jiang, Y. et al. J. Am. Chem. Soc. 145, 2698–2707 (2023)*), and $\cdot\text{CH}_3$ from CH_4 activation ($\cdot\text{CH}_3/\text{CH}_4 = 2.06 \text{ eV vs NHE}$, *Reed, J. J. Res. Natl. Inst. Stand. Technol. 125, 125007 (2020)*; *Amano, F. et al. ACS Energy Lett. 4, 502–507 (2019)*; *Jiang, Y. et al. J. Am. Chem. Soc. 145, 2698–2707 (2023)*), which all locate

within the band energy potential of $\text{WO}_3\{110\}$.

Our revision: The redox potential energy for the intermediate formation has been provided in the revised supplementary information on page of S8.

Question 12: A meaningful unit must be provided for the vertical axis of photocurrent response in Figure S60, instead of an arbitrary unit.

Our response: Thanks a lot for the reviewer's kind reminding. Following the reviewer's kind suggestion, the information for the vertical axis of photocurrent response has been added, as shown in Figure R33.

Figure R33. Photocurrent of (a) $\text{WO}_3\{001\}$ and (b) $\text{WO}_3\{110\}$.

Our revision: Figure R38 has been added as Figure S72 in the revised supplementary information.

REVIEWERS' COMMENTS

Reviewer #1 (Remarks to the Author):

My concerns are addressed in this response. This article is suitable for publication.

Reviewer #2 (Remarks to the Author):

The authors have adequately addressed previously raised concerns, and the observed reaction trends are now satisfactorily explained. The proposed reaction mechanisms for respective $WO_3\{100\}$ and $WO_3\{110\}$ have been validated and neatly reconciled with an extensive list of supporting theoretical and experimental studies. However, there are still some minor errors that can be spotted in the manuscript, such as:

- 1) Page 4, line 1: it should be lattice-O (ca. 530.4 eV)
- 2) Page S68, line 21: 'sing' should be 'single'
- 3) Page 19: "Discussion" should be "Conclusion"

I would recommend the publication of the manuscript after the authors carefully undertake final proofreading.

Response to Reviewers

Reviewer #1

Question 1: My concerns are addressed in this response. This article is suitable for publication.

Our response: We truly appreciate the reviewer's encouraging comments. And we hope that this work will open the avenue towards optimizing the performance of photocatalytic CH₄ oxidation reaction.

Reviewer #2

Question 1: The authors have adequately addressed previously raised concerns, and the observed reaction trends are now satisfactorily explained. The proposed reaction mechanisms for respective WO₃{100} and WO₃{110} have been validated and neatly reconciled with an extensive list of supporting theoretical and experimental studies. However, there are still some minor errors that can be spotted in the manuscript, such as:

- 1) Page 4, line 1: it should be lattice-O (ca. 530.4 eV)
- 2) Page S68, line 21: 'sing' should be 'single'
- 3) Page 19: "Discussion" should be "Conclusion"

I would recommend the publication of the manuscript after the authors carefully undertake final proofreading.

Our response: We truly appreciate the reviewer's encouraging comments. A careful proofreading has been made.

- 1) Page 4, line 1: the lattice-O (ca. 530.4 eV) has been modified.
- 2) Page S68, line 21: 'sing' has been modified to 'single'.
- 3) According to the article structure of Nature communications, no "Conclusion" section should be involved, but a "Discussion" section. In order to make the article structure more reasonable, the "Discussion" part is removed, meanwhile the "Results" section is changed to "Results and Discussion".

Besides, the whole part of the article has been carefully proofread.